# DELTA: DEGRADATION-FREE FULLY TEST-TIME ADAPTATION[*]

**Bowen Zhao**[1,2]**, Chen Chen**[3,✉]**, Shu-Tao Xia**[1,4,✉]
[1]Tsinghua University, [2]Tencent TEG AI, [3]OPPO research institute, [4]Peng Cheng Laboratory
zbw18@mails.tsinghua.edu.cn, chen1634chen@gmail.com, xiast@sz.tsinghua.edu.cn

## ABSTRACT

Fully test-time adaptation aims at adapting a pre-trained model to the test stream during real-time inference, which is urgently required when the test distribution differs from the training distribution. Several efforts have been devoted to improving adaptation performance. However, we find that two unfavorable defects are concealed in the prevalent adaptation methodologies like test-time batch normalization (BN) and self-learning. First, we reveal that the normalization statistics in test-time BN are completely affected by the currently received test samples, resulting in inaccurate estimates. Second, we show that during test-time adaptation, the parameter update is biased towards some dominant classes. In addition to the extensively studied test stream with independent and class-balanced samples, we further observe that the defects can be exacerbated in more complicated test environments, such as (time) dependent or class-imbalanced data. We observe that previous approaches work well in certain scenarios while show performance degradation in others due to their faults. In this paper, we provide a plug-in solution called DELTA for Degradation-freE fuLly Test-time Adaptation, which consists of two components: (i) Test-time Batch Renormalization (TBR), introduced to improve the estimated normalization statistics. (ii) Dynamic Online re-weighTing (DOT), designed to address the class bias within optimization. We investigate various test-time adaptation methods on three commonly used datasets with four scenarios, and a newly introduced real-world dataset. DELTA can help them deal with all scenarios simultaneously, leading to SOTA performance.

## 1 INTRODUCTION

Models suffer from performance decrease when test and training distributions are mismatched (Quinonero-Candela et al., 2008). Numerous studies have been conducted to narrow the performance gap based on a variety of hypotheses/settings. Unsupervised domain adaptation methods (Ganin et al., 2016) necessitate simultaneous access to labeled training data and unlabeled target data, limiting their applications. Source-free domain adaptation approaches (Liang et al., 2020) only need a trained model and do not require original training data when performing adaptation. Nonetheless, in a more difficult and realistic setting, known as fully test-time adaptation (Wang et al., 2021), the model must perform online adaptation to the test stream in real-time inference. The model is adapted in a single pass on the test stream using a pre-trained model and continuously arriving test data (rather than a prepared target set). Offline iterative training or extra heavy computational burdens beyond normal inference do not meet the requirements.

There have been several studies aimed at fully test-time adaptation. Test-time BN (Nado et al., 2020) / BN adapt (Schneider et al., 2020) directly uses the normalization statistics derived from test samples instead of those inherited from the training data, which is found to be beneficial in reducing the performance gap. Entropy-minimization-based methods, such as TENT (Wang et al., 2021), further optimize model parameters during inference. Contrastive learning (Chen et al., 2022), data augmentation (Wang et al., 2022a) and uncertainty-aware optimization (Niu et al., 2022) have been introduced to enhance adaptation performance. Efforts have also been made to address test-time adaptation in more complex test environments, like LAME (Boudiaf et al., 2022).

---

[*]work done by Bowen Zhao (during internship) and Chen Chen at Tencent.

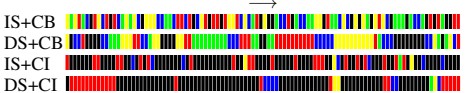

Figure 1: IS+CB / DS+CB: the test stream which is independently / dependently sampled from a class-balanced test distribution; IS+CI/ DS+CI: independently / dependently drawn from a class-imbalanced test distribution. Each bar represents a sample, each color represents a category.

Table 1: Comparison of fully test-time adaptation methods against the pre-trained model on CIFAR100-C. DELTA achieves improvement in all scenarios.

| Scenario | TENT | LAME | DELTA (Ours) |
|---|---|---|---|
| IS+CB | ▲ | ▼ | ▲ |
| DS+CB | ▼ | ▲ | ▲ |
| IS+CI | ▲ | ▼ | ▲ |
| DS+CI | ▼ | ▲ | ▲ |

Despite the achieved progress, we find that there are non-negligible defects hidden in the popular methods. First, we take a closer look at the normalization statistics within inference (Section 3.2). We observe that the statistics used in BN adapt is inaccurate in per batch compared to the actual population statistics. Second, we reveal that the prevalent test-time model updating is biased towards some dominant categories (Section 3.3). We notice that the model predictions are extremely imbalanced on out-of-distribution data, which can be exacerbated by the self-learning-based adaptation methods. Besides the most common independent and class-balanced test samples considered in existing studies, following Boudiaf et al. (2022), we investigate other three test scenarios as illustrated in Figure 1 (please see details in Section 3.1) and find when facing the more intricate test streams, like dependent samples or class-imbalanced data, the prevalent methods would suffer from severe performance degradation, which limits the usefulness of these test-time adaptation strategies.

To address the aforementioned issues, we propose two powerful tools. Specifically, to handle the inaccurate normalization statistics, we introduce test-time batch renormalization (TBR) (Section 3.2), which uses the test-time moving averaged statistics to rectify the normalized features and considers normalization during gradient optimization. By taking advantage of the observed test samples, the calibrated normalization is more accurate. We further propose dynamic online re-weighting (DOT) (Section 3.3) to tackle the biased optimization, which is derived from cost-sensitive learning. To balance adaptation, DOT assigns low/high weights to the frequent/infrequent categories. The weight mapping function is based on a momentum-updated class-frequency vector that takes into account multiple sources of category bias, including the pre-trained model, the test stream, and the adaptation methods (the methods usually do not have an intrinsic bias towards certain classes, but can accentuate existing bias). TBR can be applied directly to the common BN-based pre-trained models and does not interfere with the training process (corresponding to the fully test-time adaptation setting), and DOT can be easily combined with other adaptation approaches as well.

Table 1 compares our method to others on CIFAR100-C across various scenarios. The existing test-time adaptation methods behave differently across the four scenarios and show performance degradation in some scenarios. While our tools perform well in all four scenarios simultaneously without any prior knowledge of the test data, which is important for real-world applications. Thus, the whole method is named DELTA (Degradation-freE fuLly Test-time Adaptation).

The major contributions of our work are as follows. **(i)** We expose the defects in commonly used test-time adaptation methods, which ultimately harm adaptation performance. **(ii)** We demonstrate that the defects will be even more severe in complex test environments, causing performance degradation. **(iii)** To achieve degradation-free fully test-time adaptation, we propose DELTA which comprises two components: TBR and DOT, to improve the normalization statistics estimates and mitigate the bias within optimization. **(iv)** We evaluate DELTA on three common datasets with four scenarios and a newly introduced real-world dataset, and find that it can consistently improve the popular test-time adaptation methods on all scenarios, yielding new state-of-the-art results.

## 2 RELATED WORK

**Unsupervised domain adaptation (UDA).** In reality, test distribution is frequently inconsistent with the training distribution, resulting in poor performance. UDA aims to alleviate the phenomenon with the collected unlabeled samples from the target distribution. One popular approach is to align the statistical moments across different distributions (Gretton et al., 2006; Zellinger et al., 2017; Long et al., 2017). Another line of studies adopts adversarial training to achieve adaptation (Ganin et al., 2016;

Long et al., 2018). UDA has been developed for many tasks including object classification (Saito et al., 2017)/detection (Li et al., 2021) and semantic segmentation (Hoffman et al., 2018).

**Source-free domain adaptation (SFDA).** SFDA deals with domain gap with only the trained model and the prepared unlabeled target data. To be more widely used, SFDA methods should be built on a common source model trained by a standard pipeline. SHOT (Liang et al., 2020) freezes the source model's classifier and optimizes the feature extractor via entropy minimization, diversity regularization, and pseudo-labeling. SHOT incorporates weight normalization, 1D BN, and label-smoothing into backbones and training, which do not exist in most off-the-shelf trained models, but its other ideas can be used. USFDA (Kundu et al., 2020) utilizes synthesized samples to achieve compact decision boundaries. NRC (Yang et al., 2021b) encourages label consistency among local target features with the same network architecture as SHOT. GSFDA (Yang et al., 2021a) further expects the adapted model performs well not only on target data but also on source data.

**Fully test-time adaptation (FTTA).** FTTA is a more difficult and realistic setting. In the same way that SFDA does not provide the source training data, only the trained model is provided. Unlike SFDA, FTTA cannot access the entire target dataset; however, the methods should be capable of doing online adaptation on the test stream and providing instant predictions for the arrived test samples. BN adapt (Nado et al., 2020; Schneider et al., 2020) replaces the normalization statistics estimated during training with those derived from the test mini-batch. On top of it, TENT (Wang et al., 2021) optimizes the affine parameters in BN through entropy minimization during test. EATA (Niu et al., 2022) and CoTTA (Wang et al., 2022a) study long-term test-time adaptation in continually changing environments. ETA (Niu et al., 2022) excludes unreliable and redundant samples from the optimization. AdaContrast (Chen et al., 2022) resorts to contrastive learning to promote feature learning along with a pseudo label refinement mechanism. Both AdaContrast and CoTTA utilize heavy data augmentation during test, which will increase inference latency. Besides, AdaContrast modifies the model architecture as in SHOT. Different from them, LAME (Boudiaf et al., 2022) does not rectify the model's parameters but only the model's output probabilities via the introduced unsupervised objective laplacian adjusted maximum-likelihood estimation.

**Class-imbalanced learning.** Training with class-imbalanced data has attracted widespread attention (Liu et al., 2019). Cost-sensitive learning (Elkan, 2001) and resampling (Wang et al., 2020) are the classical strategies to handle this problem. Ren et al. (2018) designs a meta-learning paradigm to assign weights to samples. Class-balanced loss (Cui et al., 2019) uses the effective number of samples when performing re-weighting. Decoupled training (Kang et al., 2020b) learns the feature extractor and the classifier separately. Menon et al. (2021) propose logit adjustment from a statistical perspective. Other techniques such as weight balancing (Alshammari et al., 2022; Zhao et al., 2020), contrastive learning (Kang et al., 2020a), knowledge distillation (He et al., 2021), *etc.* have also been applied to solve this problem.

## 3 DELTA: DEGRADATION-FREE FULLY TEST-TIME ADAPTATION

### 3.1 PROBLEM DEFINITION

Assume that we have the training data $\mathcal{D}^{\text{train}} = \{(x_i, y_i)\}_{i=1}^{N^{\text{train}}} \sim P^{\text{train}}(x, y)$, where $x \in \mathcal{X}$ is the input and $y \in \mathcal{Y} = \{1, 2, \cdots, K\}$ is the target label; $f_{\{\theta_0, a_0\}}$ denotes the model with parameters $\theta_0$ and normalization statistics $a_0$ learned or estimated on $\mathcal{D}^{\text{train}}$. Without loss of generality, we denote the test stream as $\mathcal{D}^{\text{test}} = \{(x_j, y_j)\}_{j=1}^{N^{\text{test}}} \sim P^{\text{test}}(x, y)$, where $\{y_j\}$ are not available actually, the subscript $j$ also indicates the sample position within the test stream. When $P^{\text{test}}(x, y) \neq P^{\text{train}}(x, y)$ (the input/output space $\mathcal{X}/\mathcal{Y}$ is consistent between training and test data), $f_{\{\theta_0, a_0\}}$ may perform poorly on $\mathcal{D}^{\text{test}}$. Under fully test-time adaptation scheme (Wang et al., 2021), during inference step $t \geq 1$, the model $f_{\{\theta_{t-1}, a_{t-1}\}}$ receives a mini-batch of test data $\{x_{m_t+b}\}_{b=1}^{B}$ with $B$ batch size ($m_t$ is the number of test samples observed before inference step $t$), and then elevates itself to $f_{\{\theta_t, a_t\}}$ based on current test mini-batch and outputs the real-time predictions $\{p_{m_t+b}\}_{b=1}^{B}$ ($p \in \mathbb{R}^K$). Finally, the evaluation metric is calculated based on the online predictions from each inference step. Fully test-time adaptation emphasizes performing adaptation during real-time inference entirely, *i.e.*, the training process cannot be interrupted, the training data is no longer available during test, and the adaptation should be accomplished in a single pass over the test stream.

The most common hypothesis is that $\mathcal{D}^{\text{test}}$ is independently sampled from $P^{\text{test}}(x, y)$. However, in real environment, the assumption does not always hold, *e.g.*, samples of some classes may appear more frequently in a certain period of time, leading to another hypothesis: the test samples are dependently sampled. Most studies only considered the scenario with class-balanced test samples, while in real-world, the test stream can be class-imbalanced[1]. We investigate fully test-time adaptation under the four scenarios below, considering the latent sampling strategies and the test class distribution. For convenience, we denote the scenario where test samples are independently/dependently sampled from a class-balanced test distribution as IS+CB / DS+CB; denote the scenario where test samples are independently/dependently sampled from a class-imbalanced test distribution as IS+CI/DS+CI, as shown in Figure 1. Among them, IS+CB is the most common scenario within FTTA studies, and the other three scenarios also frequently appear in real-world applications.

### 3.2 A CLOSER LOOK AT NORMALIZATION STATISTICS

We revisit BN (Ioffe & Szegedy, 2015) briefly. Let $v \in \mathbb{R}^{B \times C \times S \times S'}$ be a mini-batch of features with $C$ channels, height $S$ and width $S'$. BN normalizes $v$ with the normalization statistics $\mu, \sigma \in \mathbb{R}^C$: $v^* = \frac{v - \mu}{\sigma}, v^\star = \gamma \cdot v^* + \beta$, where $\gamma, \beta \in \mathbb{R}^C$ are the learnable affine parameters, $\{\gamma, \beta\} \subset \theta$. We mainly focus on the first part $v \to v^*$ (all the discussed normalization methods adopt the affine parameters). In BN, during training, $\mu, \sigma$ are set to the empirical mean $\mu^{\text{batch}}$ and standard deviation $\sigma^{\text{batch}}$ calculated for each channel $c$: $\mu^{\text{batch}}[c] = \frac{1}{BSS'} \sum_{b,s,s'} v[b, c, s, s']$, $\sigma^{\text{batch}}[c] = \sqrt{\frac{1}{BSS'} \sum_{b,s,s'} (v[b, c, s, s'] - \mu^{\text{batch}}[c])^2 + \epsilon}$, where $\epsilon$ is a small value to avoid division by zero. During inference, $\mu, \sigma$ are set to $\mu^{\text{ema}}, \sigma^{\text{ema}}$ which are the exponential-moving-average (EMA) estimates over training process ($a_0$ is formed by the EMA statistics of all BN modules). However, when $P^{\text{test}}(x, y) \neq P^{\text{train}}(x, y)$, studies found that replacing $\mu^{\text{ema}}, \sigma^{\text{ema}}$ with the statistics of the test mini-batch: $\hat{\mu}^{\text{batch}}, \hat{\sigma}^{\text{batch}}$ can improve model accuracy (Nado et al., 2020) (for clarify, statistics estimated on test samples are denoted with '^'). The method is also marked as "BN adapt" (Schneider et al., 2020).

**Diagnosis I: Normalization statistics are inaccurate within each test mini-batch.** We conduct experiments on CIFAR100-C. From Figure 2 we can see that the statistics $\hat{\mu}^{\text{batch}}, \hat{\sigma}^{\text{batch}}$ used in BN adapt fluctuate dramatically during adaptation, and are inaccurate in most test mini-batches. It should be noted that for BN adapt, predictions are made online based on real-time statistics, so poor estimates can have a negative impact on performance. More seriously, the estimates in the DS+CB scenario are worse. In Table 2, though BN adapt and TENT can improve accuracy compared to Source (test with the fixed pre-trained model $f_{\{\theta_0, a_0\}}$) in IS+CB scenario, they suffer from degradation in the DS+CB cases. Overall, we can see that the poor statistics severely impede test-time adaptation because they are derived solely from the current small mini-batch.

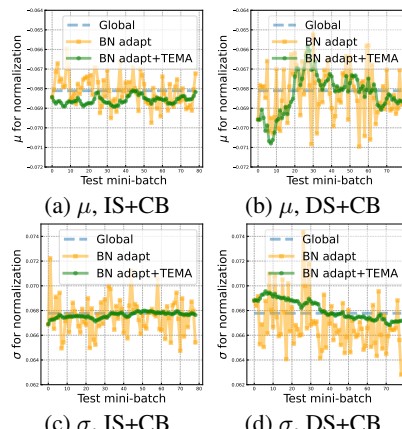

(a) $\mu$, IS+CB      (b) $\mu$, DS+CB

(c) $\sigma$, IS+CB      (d) $\sigma$, DS+CB

Figure 2: Normalization statistics in different scenarios on CIFAR100-C.

**Treatment I: Test-time batch renormalization (TBR) is a simple and powerful tool to improve the normalization.** It is natural to simply employ the test-time moving averages $\hat{\mu}^{\text{ema}}, \hat{\sigma}^{\text{ema}}$ to perform normalization during adaptation, referring to as TEMA, where $\hat{\mu}_t^{\text{ema}} = \alpha \cdot \hat{\mu}_{t-1}^{\text{ema}} + (1 - \alpha) \cdot sg(\hat{\mu}_t^{\text{batch}}), \hat{\sigma}_t^{\text{ema}} = \alpha \cdot \hat{\sigma}_{t-1}^{\text{ema}} + (1 - \alpha) \cdot sg(\hat{\sigma}_t^{\text{batch}})$, $sg(\cdot)$ stands for the operation of stopping gradient, *e.g.*, the Tensor.detach() function in PyTorch, $\alpha$ is a smoothing coef-

Table 2: Average accuracy (%) of 15 corrupted sets on CIFAR100-C.

| Method | IS+CB | DS+CB |
|---|---|---|
| Source | $53.5_{\pm 0.00}$ | $53.5_{\pm 0.00}$ |
| BN adapt | $64.3_{\pm 0.05}$ | $27.3_{\pm 1.12}$ |
| BN adapt+TEMA | $64.8_{\pm 0.04}$ | $63.5_{\pm 0.51}$ |
| TENT | $68.5_{\pm 0.13}$ | $23.7_{\pm 1.04}$ |
| TENT+TEMA | $21.8_{\pm 0.84}$ | $26.2_{\pm 1.27}$ |
| TENT+TBR | $\mathbf{68.8}_{\pm 0.13}$ | $\mathbf{64.1}_{\pm 0.57}$ |

---

[1]Regarding training class distribution, in experiments, we primarily use models learned on balanced training data following the benchmark of previous studies. Furthermore, when $P^{\text{train}}(y)$ is skewed, some techniques are commonly used to bring the model closer to the one trained on balanced data, such as on YTBB-sub (Section 4), where the trained model is learned with logit adjustment on class-imbalanced training data.

ficient. TEMA can consistently improve BN adapt: the normalization statistics in Figure 2 become more stable and accurate, and the test accuracy in Table 2 is improved as well.

However, for TENT which involves parameters update, TEMA can destroy the trained model as shown in Table 2. As discussed in Ioffe & Szegedy (2015), simply employing the moving averages would neutralize the effects of gradient optimization and normalization, as the gradient descent optimization does not consider the normalization, leading to unlimited growth of model parameters. Thus, we introduce batch renormalization (Ioffe, 2017) into test-time adaptation, leading to TBR, which is formulated by

$$v^* = \frac{v - \hat{\mu}^{\text{batch}}}{\hat{\sigma}^{\text{batch}}} \cdot r + d, \quad \text{where} \quad r = \frac{sg(\hat{\sigma}^{\text{batch}})}{\hat{\sigma}^{\text{ema}}}, \quad d = \frac{sg(\hat{\mu}^{\text{batch}}) - \hat{\mu}^{\text{ema}}}{\hat{\sigma}^{\text{ema}}}, \tag{1}$$

We present a detailed algorithm description in Appendix A.2. Different from BN adapt, we use the test-time moving averages to rectify the normalization (through $r$ and $d$). Different from the TEMA, TBR is well compatible with gradient-based adaptation methods (*e.g.*, TENT) and can improve them as summarised in Table 2. For BN adapt, TEMA is equal to TBR. Different from the original batch renormalization used in the training phase, TBR is employed in the inference phase which uses the statistics and moving averages derived from test batches. Besides, as the adaptation starts with a trained model $f_{\{\theta_0, a_0\}}$, TBR discards the warm-up and truncation operation to $r$ and $d$, thus does not introduce additional hyper-parameters. TBR can be applied directly to a common pre-trained model with BN without requiring the model to be trained with such calibrated normalization.

### 3.3 A CLOSER LOOK AT TEST-TIME PARAMETER OPTIMIZATION

Building on BN adapt, TENT (Wang et al., 2021) further optimizes the affine parameters $\gamma, \beta$ through entropy minimization and shows that test-time parameter optimization can yield better results compared to employing BN adapt alone. We further take a closer look at this procedure.

**Diagnosis II: the test-time optimization is biased towards dominant classes.** We evaluate the model on IS+CB and DS+CB gaussian-noise-corrupted test data (Gauss) of CIFAR100-C. We also test the model on the original clean test set of CIFAR100 for comparison. Figure 3 depicts the per-class number

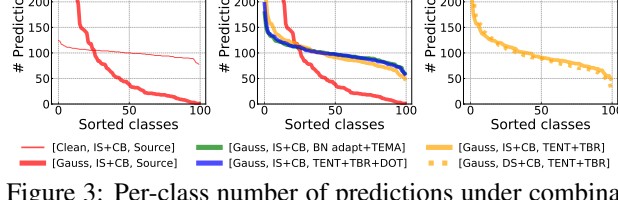

Figure 3: Per-class number of predictions under combinations of [*data, scenario, method*].

Table 3: Standard Deviation (STD), Range (R) of per-class number of predictions and accuracy (Acc, %) on Gauss data.

| Method | IS+CB | | | DS+CB | | |
|---|---|---|---|---|---|---|
| | STD | R | Acc | STD | R | Acc |
| Source | $158.3_{\pm 0.0}$ | $956.0_{\pm 0.0}$ | $27.0_{\pm 0.0}$ | $158.3_{\pm 0.0}$ | $956.0_{\pm 0.0}$ | $27.0_{\pm 0.0}$ |
| BN adapt+TEMA | $18.4_{\pm 0.2}$ | $121.6_{\pm 3.7}$ | $58.0_{\pm 0.2}$ | $19.8_{\pm 1.1}$ | $130.0_{\pm 13.6}$ | $56.7_{\pm 0.5}$ |
| TENT+TBR | $35.8_{\pm 2.9}$ | $269.8_{\pm 44.0}$ | $62.2_{\pm 0.4}$ | $52.4_{\pm 9.1}$ | $469.2_{\pm 104.2}$ | $57.1_{\pm 0.8}$ |
| TENT+TBR+DOT | $20.4_{\pm 1.1}$ | $122.0_{\pm 15.2}$ | $\mathbf{63.9}_{\pm 0.2}$ | $25.5_{\pm 2.1}$ | $164.6_{\pm 43.0}$ | $\mathbf{60.4}_{\pm 0.5}$ |

of predictions, while Table 3 shows the corresponding standard deviation, range (maximum subtract minimum), and accuracy. We draw the following five conclusions.

- Predictions are imbalanced, even for a model trained on class-balanced training data and tested on a class-balanced test set with $P^{\text{test}}(x, y) = P^{\text{train}}(x, y)$: the "clean" curve in Figure 3 (left) with standard deviation 8.3 and range 46. This phenomenon is also studied in Wang et al. (2022b).
- Predictions becomes more imbalanced when $P^{\text{test}}(x, y) \neq P^{\text{train}}(x, y)$ as shown in Figure 3 (left): the ranges are 46 and 956 on the clean and corrupted test set respectively.
- BN adapt+TEMA improves accuracy (from 27.0% to 58.0%) and alleviates the prediction imbalance at the same time (the range dropped from 956 to 121.6).
- Though accuracy is further improved with TENT+TBR (from 58.0% to 62.2%), the predictions become more imbalanced inversely (the range changed from 121.6 to 269.8). The entropy minimization loss focuses on data with low entropy, while samples of some classes may have relatively lower entropy owing to the trained model, thus TENT would aggravate the prediction imbalance.
- On dependent test streams, not only the model accuracy drops, but also the predictions become more imbalanced (range 269.8 / range 469.2 on independent/dependent samples for TENT+TBR), as the model may be absolutely dominated by some classes over a period of time in DS+CB scenario.

---

**Algorithm 1: D**ynamic **O**nline reweigh**T**ing (DOT)

---

**Input:** inference step $t := 0$; test stream samples $\{x_j\}$; pre-trained model $f_{\{\theta_0,a_0\}}$; class-frequency vector $z_0$; loss function $\mathcal{L}$; smooth coefficient $\lambda$.

1 **while** *the test mini-batch* $\{x_{m_t+b}\}_{b=1}^{B}$ *arrives* **do**
2      $t = t + 1$
3      $\{p_{m_t+b}\}_{b=1}^{B}, f_{\{\theta_{t-1},a_t\}} \leftarrow \text{Forward}(\{x_{m_t+b}\}_{b=1}^{B}, f_{\{\theta_{t-1},a_{t-1}\}})$ // `output predictions`
4      **for** $b = 1$ **to** $B$ **do**
5          $k^*_{m_t+b} = \arg\max_{k \in [1,K]} p_{m_t+b}[k]$ // `predicted label`
6          $w_{m_t+b} = 1/(z_{t-1}[k^*_{m_t+b}]+\epsilon)$ // `assign sample weight`
7      $\bar{w}_{m_t+b} = B \cdot w_{m_t+b}/\sum_{b'=1}^{B} w_{m_t+b'},\ b = 1, 2, \cdots, B$ // `normalize sample weight`
8      $l = \frac{1}{B}\sum_{b=1}^{B} \bar{w}_{m_t+b} \cdot \mathcal{L}(p_{m_t+b})$ // `combine sample weight with loss`
9      $f_{\{\theta_t,a_t\}} \leftarrow \text{Backward \& Update}(l, f_{\{\theta_{t-1},a_t\}})$ // `update θ`
10      $z_t \leftarrow \lambda z_{t-1} + \frac{(1-\lambda)}{B}\sum_{b=1}^{B} p_{m_t+b}$ // `update z`

---

The imbalanced data is harmful during the normal training phase, resulting in biased models and poor overall accuracy (Liu et al., 2019; Menon et al., 2021). Our main motivation is that the test-time adaptation methods also involve gradient-based optimization which is built on the model predictions; however, the predictions are actually imbalanced, particularly for dependent or class-imbalanced streams and the low-entropy-emphasized adaptation methods. Therefore, we argue that the test-time optimization is biased towards some dominant classes actually, resulting in inferior performance. A vicious circle is formed by skewed optimization and imbalanced predictions.

**Treatment II: Dynamic online re-weighting (DOT) can alleviate the biased optimization.** Many methods have been developed to deal with class imbalance during the training phase, but they face several challenges when it comes to fully test-time adaptation: (i) Network architectures are immutable. (ii) Because test sample class frequencies are dynamic and agnostic, the common constraint of making the output distribution uniform (Liang et al., 2020) is no longer reasonable. (iii) Inference and adaptation must occur in real-time when test mini-batch arrived (only a single pass through test data, no iterative learning).

Given these constraints, we propose DOT as presented in Algorithm 1. DOT is mainly derived from class-wise re-weighting (Cui et al., 2019). To tackle the dynamically changing and unknown class frequencies, we use a momentum-updated class-frequency vector $z \in \mathbb{R}^K$ instead (Line 10 of Algorithm 1), which is initiated with $z[k] = \frac{1}{K}$, $k = 1, 2, \cdots, K$. For each inference step, we assign weights to each test sample based on its pseudo label and the current $z$ (Line 5,6 of Algorithm 1). Specifically, when $z[k]$ is relatively large, during the subsequent adaptation, DOT will reduce the contributions of the $k^{\text{th}}$ class samples (pseudo label) and emphasize others. It is worth noting that DOT can alleviate the biased optimization caused by the pre-trained model (*e.g.*, inter-class similarity), test stream (*e.g.*, class-imbalanced scenario) simultaneously.

DOT is a general idea to tackle the biased optimization, some parts in Algorithm 1 have multiple options, so it can be combined with different existing test-time adaptation techniques. For the "Forward (·)" function (Line 3 of Algorithm 1), the discussed BN adapt and TBR can be incorporated. For the loss function $\mathcal{L}(\cdot)$ (Line 8 of Algorithm 1), studies usually employ the entropy minimization loss: $\mathcal{L}(p_b) = -\sum_{k=1}^{K} p_b[k] \log p_b[k]$ or the cross-entropy loss with pseudo labels: $\mathcal{L}(p_b) = -\mathbb{I}_{p_b[k^*_b] \geq \tau} \cdot \log p_b[k^*_b]$ (commonly, only samples with high prediction confidence are utilized, $\tau$ is a pre-defined threshold). Similarly, for entropy minimization, Ent-W (Niu et al., 2022) also discards the high-entropy samples and emphasizes the low-entropy ones: $\mathcal{L}(p_b) = -\mathbb{I}_{H_b < \tau} \cdot e^{\tau - H_b} \cdot \sum_{k=1}^{K} p_b[k] \log p_b[k]$, where $H_b$ is the entropy of sample $x_b$.

## 4 EXPERIMENTS

**Datasets and models.** We conduct experiments on common datasets CIFAR100-C, ImageNet-C (Hendrycks & Dietterich, 2019), ImageNet-R (Hendrycks et al., 2021), and a newly introduced video (segments) dataset: the subset of YouTube-BoundingBoxes (YTBB-sub) (Real et al., 2017). CIFAR100-C / ImageNet-C contains 15 corruption types, each with 5 severity levels; we use the

highest level unless otherwise specified. ImageNet-R contains various styles (*e.g.*, paintings) of ImageNet categories. Following Wang et al. (2022a); Niu et al. (2022), for evaluations on CIFAR100-C, we adopt the trained ResNeXt-29 (Xie et al., 2017) model from Hendrycks et al. (2020) as $f_{\{\theta_0, a_0\}}$; for ImageNet-C / -R, we use the trained ResNet-50 model from Torchvision. The models are trained on the corresponding original training data. For YTBB-sub, we use a ResNet-18 trained on the related images of COCO. Details of the tasks, datasets and examples are provided in Appendix A.1.

**Metrics.** Unless otherwise specified, we report the mean accuracy over classes (Acc, %) (Liu et al., 2019); results are averaged over 15 different corruption types for CIFAR100-C and ImageNet-C in the main text, please see detailed performance on each corruption type in Appendix A.5, A.6.

**Implementation.** The configurations are mainly followed previous work Wang et al. (2021; 2022a); Niu et al. (2022) for comparison, details are listed in Appendix A.3. Code is available online.

**Baselines.** We adopt the following SOTA methods as baselines: pseudo label (PL) (Lee et al., 2013), test-time augmentation (TTA) (Ashukha et al., 2020), BN adaptation (BN adapt) (Schneider et al., 2020; Nado et al., 2020), test-time entropy minimization (TENT) (Wang et al., 2021), marginal entropy minimization with one test point (MEMO) (Zhang et al., 2021), efficient test-time adaptation (ETA) (Niu et al., 2022), entropy-based weighting (Ent-W) (Niu et al., 2022), laplacian adjusted maximum-likelihood estimation (LAME) (Boudiaf et al., 2022), continual test-time adaptation (CoTTA/CoTTA*: w/wo resetting) (Wang et al., 2022a). We combine DELTA with PL, TENT, and Ent-W in this work.

Table 4: Acc in IS+CB scenario.

| Method | CIFAR100-C | ImageNet-C |
|---|---|---|
| Source | $53.5_{\pm 0.00}$ | $18.0_{\pm 0.00}$ |
| TTA | – | 17.7 |
| BN adapt | $64.6_{\pm 0.03}$ | $31.5_{\pm 0.02}$ |
| MEMO | – | 23.9 |
| ETA | $69.3_{\pm 0.14}$ | $48.0_{\pm 0.06}$ |
| LAME | $50.8_{\pm 0.06}$ | $17.2_{\pm 0.01}$ |
| CoTTA | $65.5_{\pm 0.04}$ | $34.4_{\pm 0.11}$ |
| CoTTA* | $67.3_{\pm 0.13}$ | $34.8_{\pm 0.53}$ |
| PL | $68.0_{\pm 0.13}$ | $40.2_{\pm 0.11}$ |
| +DELTA | $68.7_{\pm 0.12}$ | $41.8_{\pm 0.03}$ |
| | +0.7 | +1.6 |
| TENT | $68.7_{\pm 0.16}$ | $42.7_{\pm 0.03}$ |
| +DELTA | $69.5_{\pm 0.03}$ | $45.1_{\pm 0.03}$ |
| | +0.8 | +2.4 |
| Ent-W | $69.3_{\pm 0.15}$ | $44.3_{\pm 0.41}$ |
| +DELTA | $\mathbf{70.1}_{\pm 0.05}$ | $\mathbf{49.9}_{\pm 0.05}$ |
| | +0.8 | +5.6 |

**Evaluation in IS+CB scenario.** The results on CIFAR100-C are reported in Table 4. As can be seen, the proposed DELTA consistently improves the previous adaptation approaches PL (gain 0.7%), TENT (gain 0.8%), and Ent-W (gain 0.8%), achieving new state-of-the-art performance. The results also indicate that current test-time adaptation methods indeed suffer from the discussed drawbacks, and the proposed methods can help them obtain superior performance. Then we evaluate the methods on the more challenging dataset ImageNet-C. Consistent with the results on CIFAR100-C, DELTA remarkably improves the existing methods. As the adaptation batch size (64) is too small compared to the class number (1,000) on ImageNet-C, the previous methods undergo more severe damage than on CIFAR100-C. Consequently, DELTA achieves greater gains on ImageNet-C: 1.6% gain over PL, 2.4% gain over TENT, and 5.6% gain over Ent-W.

**Evaluation in DS+CB scenario.** To simulate dependent streams, following Yurochkin et al. (2019), we arrange the samples via the Dirichlet distribution with a concentration factor $\rho > 0$ (the smaller $\rho$ is, the more concentrated the same-class samples will be, which is detailed in Appendix A.1). We test models with $\rho \in \{1.0, 0.5, 0.1\}$. The experimental results are provided in Table 5 (we provide the results of more extreme cases with $\rho = 0.01$ in Appendix A.4). The representative test-time adaptation methods suffer from performance degradation in the dependent scenario, especially on data sampled with small $\rho$. DELTA successfully helps

Table 5: Acc in DS+CB scenario with varying $\rho$.

| Method | CIFAR100-C | | | ImageNet-C | | |
|---|---|---|---|---|---|---|
| | 1.0 | 0.5 | 0.1 | 1.0 | 0.5 | 0.1 |
| Source | $53.5_{\pm 0.00}$ | $53.5_{\pm 0.00}$ | $53.5_{\pm 0.00}$ | $18.0_{\pm 0.00}$ | $18.0_{\pm 0.00}$ | $18.0_{\pm 0.00}$ |
| BN adapt | $53.0_{\pm 0.48}$ | $49.0_{\pm 0.32}$ | $35.2_{\pm 0.64}$ | $21.8_{\pm 0.12}$ | $19.2_{\pm 0.09}$ | $12.1_{\pm 0.13}$ |
| ETA | $55.4_{\pm 0.63}$ | $50.5_{\pm 0.34}$ | $34.5_{\pm 0.83}$ | $27.6_{\pm 0.31}$ | $22.4_{\pm 0.20}$ | $9.7_{\pm 0.24}$ |
| LAME | $60.3_{\pm 0.25}$ | $61.8_{\pm 0.26}$ | $65.4_{\pm 0.41}$ | $21.9_{\pm 0.03}$ | $22.7_{\pm 0.05}$ | $24.7_{\pm 0.03}$ |
| CoTTA | $53.8_{\pm 0.51}$ | $50.0_{\pm 0.23}$ | $36.3_{\pm 0.63}$ | $23.4_{\pm 0.15}$ | $20.5_{\pm 0.05}$ | $12.6_{\pm 0.15}$ |
| CoTTA* | $54.1_{\pm 0.65}$ | $50.2_{\pm 0.23}$ | $36.1_{\pm 0.71}$ | $23.5_{\pm 0.27}$ | $20.3_{\pm 0.55}$ | $12.8_{\pm 0.26}$ |
| PL | $54.9_{\pm 0.54}$ | $50.1_{\pm 0.29}$ | $34.8_{\pm 0.76}$ | $25.9_{\pm 0.18}$ | $22.5_{\pm 0.14}$ | $13.0_{\pm 0.09}$ |
| +DELTA | $68.0_{\pm 0.25}$ | $67.5_{\pm 0.30}$ | $66.0_{\pm 0.45}$ | $40.5_{\pm 0.05}$ | $39.9_{\pm 0.07}$ | $37.3_{\pm 0.10}$ |
| | +13.1 | +17.4 | +31.2 | +14.6 | +17.4 | +24.3 |
| TENT | $54.6_{\pm 0.52}$ | $49.7_{\pm 0.40}$ | $33.7_{\pm 0.70}$ | $26.0_{\pm 0.20}$ | $22.1_{\pm 0.12}$ | $12.1_{\pm 0.10}$ |
| +DELTA | $68.9_{\pm 0.20}$ | $68.5_{\pm 0.40}$ | $\mathbf{67.1}_{\pm 0.47}$ | $43.7_{\pm 0.06}$ | $43.1_{\pm 0.07}$ | $40.3_{\pm 0.06}$ |
| | +14.3 | +18.8 | +33.4 | +17.7 | +21.0 | +28.2 |
| Ent-W | $55.4_{\pm 0.63}$ | $50.5_{\pm 0.35}$ | $34.5_{\pm 0.83}$ | $17.4_{\pm 0.40}$ | $13.0_{\pm 0.22}$ | $4.1_{\pm 0.22}$ |
| +DELTA | $\mathbf{69.4}_{\pm 0.22}$ | $\mathbf{68.8}_{\pm 0.35}$ | $\mathbf{67.1}_{\pm 0.45}$ | $\mathbf{48.3}_{\pm 0.12}$ | $\mathbf{47.4}_{\pm 0.04}$ | $\mathbf{43.2}_{\pm 0.11}$ |
| | +14.0 | +18.3 | +32.6 | +30.9 | +34.4 | +39.1 |

models adapt to environments across different concentration factors. It is worth noting that DELTA's DS+CB results are close to the IS+CB results, *e.g.*, TENT+DELTA achieves 69.5% and 68.5% accuracy on IS+CB and DS+CB ($\rho = 0.5$) test streams from CIFAR100-C.

**Evaluation in IS+CI and DS+CI scenarios.** Following Cui et al. (2019), we resample the test samples with an imbalance factor $\pi$ (the smaller $\pi$ is, the more imbalanced the test data will be,

Table 6: Mean acc in IS+CI, DS+CI scenarios with different $\pi$.

| Method | IS+CI | | | | DS+CI ($\rho = 0.5$) | | | |
|---|---|---|---|---|---|---|---|---|
| | CIFAR100-C | | ImageNet-C | | CIFAR100-C | | ImageNet-C | |
| | 0.1 | 0.05 | 0.1 | 0.05 | 0.1 | 0.05 | 0.1 | 0.05 |
| Source | $53.3_{\pm0.00}$ | $53.3_{\pm0.00}$ | $17.9_{\pm0.00}$ | $17.9_{\pm0.00}$ | $53.3_{\pm0.00}$ | $53.3_{\pm0.00}$ | $17.9_{\pm0.00}$ | $17.9_{\pm0.00}$ |
| BN adapt | $64.3_{\pm0.16}$ | $64.2_{\pm0.48}$ | $31.5_{\pm0.24}$ | $31.4_{\pm0.19}$ | $49.8_{\pm0.47}$ | $49.9_{\pm0.63}$ | $20.0_{\pm0.22}$ | $20.5_{\pm0.22}$ |
| ETA | $68.2_{\pm0.24}$ | $68.2_{\pm0.59}$ | $47.4_{\pm0.23}$ | $47.1_{\pm0.18}$ | $51.1_{\pm0.45}$ | $51.0_{\pm0.54}$ | $21.7_{\pm0.52}$ | $21.0_{\pm0.40}$ |
| LAME | $50.6_{\pm0.18}$ | $50.8_{\pm0.39}$ | $17.2_{\pm0.10}$ | $17.2_{\pm0.07}$ | $60.4_{\pm0.34}$ | $59.6_{\pm0.43}$ | $21.8_{\pm0.12}$ | $21.5_{\pm0.07}$ |
| CoTTA | $65.1_{\pm0.13}$ | $65.1_{\pm0.58}$ | $34.2_{\pm0.26}$ | $34.2_{\pm0.16}$ | $50.5_{\pm0.47}$ | $50.5_{\pm0.60}$ | $21.4_{\pm0.21}$ | $22.0_{\pm0.26}$ |
| CoTTA* | $67.0_{\pm0.17}$ | $66.9_{\pm0.66}$ | $34.6_{\pm0.78}$ | $34.3_{\pm0.51}$ | $50.7_{\pm0.52}$ | $50.6_{\pm0.63}$ | $21.6_{\pm0.56}$ | $22.1_{\pm0.24}$ |
| PL | $67.2_{\pm0.21}$ | $67.3_{\pm0.57}$ | $39.4_{\pm0.21}$ | $39.3_{\pm0.18}$ | $50.7_{\pm0.41}$ | $50.6_{\pm0.53}$ | $22.8_{\pm0.35}$ | $23.1_{\pm0.25}$ |
| +DELTA | $67.6_{\pm0.36}$ | $67.6_{\pm0.46}$ | $40.9_{\pm0.26}$ | $40.7_{\pm0.22}$ | $66.6_{\pm0.39}$ | $66.3_{\pm0.57}$ | $38.8_{\pm0.27}$ | $38.5_{\pm0.21}$ |
| | +0.4 | +0.3 | +1.5 | +1.4 | +15.9 | +15.7 | +16.0 | +15.4 |
| TENT | $67.7_{\pm0.29}$ | $67.7_{\pm0.58}$ | $42.2_{\pm0.26}$ | $42.0_{\pm0.21}$ | $50.3_{\pm0.41}$ | $50.2_{\pm0.56}$ | $22.3_{\pm0.25}$ | $22.5_{\pm0.23}$ |
| +DELTA | $68.5_{\pm0.31}$ | $68.6_{\pm0.60}$ | $44.4_{\pm0.25}$ | $44.2_{\pm0.22}$ | $67.7_{\pm0.41}$ | $67.5_{\pm0.70}$ | $42.1_{\pm0.28}$ | $41.9_{\pm0.24}$ |
| | +0.8 | +0.9 | +2.2 | +2.2 | +17.4 | +17.3 | +19.8 | +19.4 |
| Ent-W | $68.3_{\pm0.26}$ | $68.2_{\pm0.58}$ | $40.8_{\pm0.76}$ | $39.5_{\pm0.82}$ | $51.1_{\pm0.44}$ | $51.0_{\pm0.53}$ | $11.3_{\pm0.81}$ | $10.8_{\pm0.40}$ |
| +DELTA | $\mathbf{69.1}_{\pm0.25}$ | $\mathbf{69.2}_{\pm0.53}$ | $\mathbf{48.4}_{\pm0.31}$ | $\mathbf{47.7}_{\pm0.21}$ | $\mathbf{68.0}_{\pm0.30}$ | $\mathbf{67.8}_{\pm0.60}$ | $\mathbf{45.4}_{\pm0.53}$ | $\mathbf{44.8}_{\pm0.24}$ |
| | +0.8 | +1.0 | +7.6 | +8.2 | +16.9 | +16.8 | +34.1 | +34.0 |

Table 7: Results on in-distribution test set of CIFAR100.

| Method | Accuracy |
|---|---|
| Source | $78.9_{\pm0.00}$ |
| BN adapt | $76.1_{\pm0.15}$ |
| TENT | $78.5_{\pm0.16}$ |
| +DELTA | $78.9_{\pm0.03}$ (+0.4) |
| Ent-W | $78.6_{\pm0.19}$ |
| +DELTA | $\mathbf{79.1}_{\pm0.09}$ (+0.5) |

Figure 4: Across architecture.

which is detailed in Appendix A.1). We test models with $\pi \in \{0.1, 0.05\}$ (similarly, we show the extreme experiments with $\pi = 0.001$ in Appendix A.4). Table 6 summarizes the results in IS+CI and DS+CI scenarios, with the following observations: **(i)** Under class-imbalanced scenario, the performance degradation is not as severe as under dependent data. This is primarily because the imbalanced test data has relatively little effect on the normalization statistics. DELTA works well on the imbalanced test stream. **(ii)** The hybrid DS+CI scenario can be more difficult than the individual scenarios. DELTA can also boost baselines in the hybrid scenario. **(iii)** Though the low-entropy-emphasized method Ent-W improves TENT in IS+CB scenario (Table 4), it can be inferior to TENT in dependent or class-imbalanced scenarios (the results on ImageNet-C in Table 5,6). The reason is that Ent-W leads to a side effect — amplifying the class bias, which would neutralize or even overwhelm its benefits. DELTA eliminates Ent-W's side effects while retaining its benefits, so Ent-W+DELTA always significantly outperforms TENT+DELTA.

**Evaluation on realistic out-of-distribution datasets ImageNet-R and YTBB-sub.** ImageNet-R is inherently class-imbalanced and consists of mixed variants such as cartoon, art, painting, sketch, toy, *etc.* As shown in Table 8, DELTA also leads to consistent improvement on it. While compared to ImageNet-C, ImageNet-R is collected individually, which consists of more hard cases that are still difficult to recognize for DELTA, the gain is not as great as on ImageNet-C. For YTBB-sub, dependent and class-imbalanced samples are encountered naturally. We see that classical methods suffer from severe degradation, whereas DELTA assists them in achieving good performance.

Table 8: Mean acc on ImageNet-R and YTBB-sub.

| Method | ImageNet-R | YTBB-sub |
|---|---|---|
| Source | $38.4_{\pm0.00}$ | $74.0_{\pm0.00}$ |
| BN adapt | $41.9_{\pm0.15}$ | $51.4_{\pm0.29}$ |
| ETA | $48.3_{\pm0.37}$ | $51.5_{\pm0.32}$ |
| TENT | $44.7_{\pm0.23}$ | $51.7_{\pm0.27}$ |
| +DELTA | $45.3_{\pm0.08}$ | $75.7_{\pm0.21}$ |
| | +0.6 | +24.0 |
| Ent-W | $48.3_{\pm0.26}$ | $51.5_{\pm0.28}$ |
| +DELTA | $\mathbf{49.6}_{\pm0.09}$ | $\mathbf{76.2}_{\pm0.23}$ |
| | +1.3 | +24.7 |

**Evaluation on in-distribution test data.** A qualified FTTA method should be "safe" on in-distribution datasets, *i.e.*, $P^{\text{test}}(x, y) = P^{\text{train}}(x, y)$. According to Table 7, (i) DELTA continues to improve performance, albeit slightly; (ii) most adaptation methods can produce comparable results to Source, and the combination with DELTA even outperforms Source on in-distribution data.

**Evaluation with different architectures.** Figure 4 indicates that DELTA can help improve previous test-time adaptation methods with different model architectures. More analyses (*e.g.*, evaluations with small batch size, different severity levels) are provided in Appendix A.4.

**Contribution of each component of DELTA.** DELTA consists of two tools: TBR and DOT. In Table 9, we analyze their contributions on the basis of TENT with four scenarios and two datasets. Row #1 indicates the results of TENT. Applying either TBR or DOT alone on TENT brings gain in most scenarios and datasets. While, we find that TBR achieves less improvement when the test stream is IS+CB and the batch size is large (*e.g.*, performing adaptation with TBR alone on the IS+CB data of CIFAR100-C with batch size of 200 does not improve TENT). However, when the batch size is relatively small (*e.g.*, ImageNet-C, batch size of 64), the benefits of TBR will become apparent. More importantly, TBR is extremely effective and necessary for dependent samples.

DOT can consistently promote TENT or TENT+TBR in all scenarios, especially when the class number is large. These results demonstrate that both the inaccurate normalization statistics and the biased optimization are detrimental, TBR and DOT can effectively alleviate them.

**Comparing DOT with other techniques for class imbalance.** On the basis of Ent-W+TBR, Table 10 compares DOT against the following strategies for solving class imbalance. *Diversity-based weight (Div-W)* (Niu et al., 2022) computes the cosine similarity between the arrived test samples' prediction and a moving average one like $z$, then only employs the samples with low similarity to update model. Although the method is

Table 9: Ablation on the effectiveness of each component (on top of TENT) measured in various scenarios: IS+CB, DS+CB ($\rho$=0.5), IS+CI ($\pi$=0.1), DS+CI ($\rho$=0.5, $\pi$=0.05).

| # | TBR | DOT | CIFAR100-C | | | | ImageNet-C | | | |
|---|---|---|---|---|---|---|---|---|---|---|
| | | | IS+CB | DS+CB | IS+CI | DS+CI | IS+CB | DS+CB | IS+CI | DS+CI |
| 1 | | | $68.7_{\pm0.16}$ | $49.7_{\pm0.40}$ | $67.7_{\pm0.29}$ | $50.2_{\pm0.56}$ | $42.7_{\pm0.03}$ | $22.1_{\pm0.12}$ | $42.0_{\pm0.21}$ | $22.5_{\pm0.23}$ |
| 2 | ✓ | | $68.9_{\pm0.03}$ | $67.4_{\pm0.41}$ | $67.9_{\pm0.27}$ | $66.6_{\pm0.72}$ | $43.4_{\pm0.05}$ | $40.9_{\pm0.11}$ | $42.8_{\pm0.25}$ | $39.6_{\pm0.24}$ |
| 3 | | ✓ | $69.1_{\pm0.07}$ | $50.6_{\pm0.37}$ | $68.1_{\pm0.27}$ | $51.0_{\pm0.60}$ | $44.3_{\pm0.02}$ | $23.7_{\pm0.17}$ | $43.9_{\pm0.25}$ | $24.8_{\pm0.26}$ |
| 4 | ✓ | ✓ | $\textbf{69.5}_{\pm0.03}$ | $\textbf{68.5}_{\pm0.40}$ | $\textbf{68.5}_{\pm0.31}$ | $\textbf{67.5}_{\pm0.70}$ | $\textbf{45.1}_{\pm0.03}$ | $\textbf{43.1}_{\pm0.07}$ | $\textbf{44.2}_{\pm0.22}$ | $\textbf{41.9}_{\pm0.24}$ |

Table 10: Ablation on different techniques for class imbalance (on top of Ent-W+TBR) measured in various scenarios (same as in Table 9).

| Method | CIFAR100-C | | | | ImageNet-C | | | |
|---|---|---|---|---|---|---|---|---|
| | IS+CB | DS+CB | IS+CI | DS+CI | IS+CB | DS+CB | IS+CI | DS+CI |
| Div-W (0.05) | $67.5_{\pm0.12}$ | $68.1_{\pm0.30}$ | $66.8_{\pm0.31}$ | $67.1_{\pm0.59}$ | $48.8_{\pm0.02}$ | $45.1_{\pm0.25}$ | $47.9_{\pm0.25}$ | $41.0_{\pm0.49}$ |
| Div-W (0.1) | $69.3_{\pm0.09}$ | $68.6_{\pm0.34}$ | $68.3_{\pm0.30}$ | $67.6_{\pm0.53}$ | $48.4_{\pm0.08}$ | $43.0_{\pm0.28}$ | $47.7_{\pm0.29}$ | $39.6_{\pm0.56}$ |
| Div-W (0.2) | $69.7_{\pm0.10}$ | $68.2_{\pm0.37}$ | $68.6_{\pm0.28}$ | $67.4_{\pm0.61}$ | $46.4_{\pm0.06}$ | $40.3_{\pm0.18}$ | $46.5_{\pm0.38}$ | $37.5_{\pm0.48}$ |
| Div-W (0.4) | $69.7_{\pm0.08}$ | $68.0_{\pm0.41}$ | $68.4_{\pm0.23}$ | $67.2_{\pm0.63}$ | $43.6_{\pm0.54}$ | $37.5_{\pm0.35}$ | $44.1_{\pm0.47}$ | $35.1_{\pm0.54}$ |
| LA | $70.0_{\pm0.06}$ | $66.9_{\pm0.36}$ | $69.0_{\pm0.27}$ | $66.4_{\pm0.63}$ | $42.2_{\pm0.73}$ | $28.6_{\pm0.57}$ | $43.1_{\pm0.73}$ | $27.5_{\pm0.86}$ |
| KL-div (1e2) | – | – | – | – | $47.6_{\pm1.11}$ | $39.9_{\pm0.94}$ | $46.6_{\pm0.62}$ | $36.5_{\pm1.21}$ |
| KL-div (1e3) | – | – | – | – | $48.9_{\pm0.07}$ | $27.7_{\pm0.36}$ | $43.1_{\pm0.30}$ | $22.5_{\pm0.60}$ |
| Sample-drop | $\textbf{70.1}_{\pm0.08}$ | $68.7_{\pm0.34}$ | $69.0_{\pm0.26}$ | $67.5_{\pm0.55}$ | $49.5_{\pm0.06}$ | $46.9_{\pm0.09}$ | $48.2_{\pm0.34}$ | $42.6_{\pm0.28}$ |
| DOT | $\textbf{70.1}_{\pm0.05}$ | $\textbf{68.8}_{\pm0.35}$ | $\textbf{69.1}_{\pm0.25}$ | $\textbf{67.8}_{\pm0.60}$ | $\textbf{49.9}_{\pm0.05}$ | $\textbf{47.4}_{\pm0.04}$ | $\textbf{48.4}_{\pm0.31}$ | $\textbf{44.8}_{\pm0.24}$ |

proposed to reduce redundancy, we find it can resist class imbalance too. The method relies on a predefined similarity threshold to determine whether to use a sample. We report the results of Div-W with varying thresholds (shown in parentheses). We observe that the threshold is very sensitive and the optimal value varies greatly across datasets. *Logit adjustment (LA)* (Menon et al., 2021) shows strong performance when training on imbalanced data. Following Wang et al. (2022b), we can perform LA with the estimated class-frequency vector $z$ in test-time adaptation tasks. While we find that LA does not show satisfactory results here. We speculate that this is because the estimated class distribution is not accurate under the one-pass adaptation and small batch size, while LA requires a high-quality class distribution estimate. *KL divergence regularizer (KL-div)* (Mummadi et al., 2021) augments loss function to encourage the predictions of test samples to be uniform. While, this is not always reasonable for TTA, e.g., for the class-imbalanced test data, forcing the outputs to be uniform will hurt the performance conversely. We examine multiple regularization strength options (shown in parentheses) and report the best two. The results show that KL-div is clearly inferior in dependent or class-imbalanced scenarios. We further propose another strategy called *Sample-drop*. It records the (pseudo) categories of the test samples that have been employed, then Sample-drop will directly discard a newly arrived test sample (*i.e.*, not use the sample to update the model) if its pseudo category belongs to the majority classes among the counts. This simple strategy is valid but inferior to DOT, as it completely drops too many useful samples.

**Impacts of $\alpha$ in TBR and $\lambda$ in DOT.** Similar to most exponential-moving-average-based methods, when the smoothing coefficient $\alpha$ (or $\lambda$) is too small, the adaptation may be unstable; when $\alpha$ (or $\lambda$) is too large, the adaptation would be slow. Figure 5 provides the ablation studies of $\alpha$ (left) and $\lambda$ (right) on the DS+CB ($\rho = 0.5$) samples of CIFAR100-C (from the validation set). We find that TBR and DOT perform reasonably well under a wide range of $\alpha$ and $\lambda$.

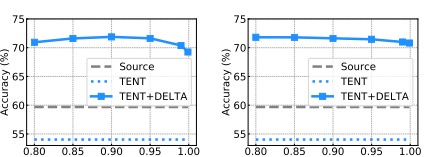

Figure 5: Impacts of $\alpha$ and $\lambda$.

## 5 CONCLUSION

In this paper, we expose the defects in test-time adaptation methods which cause suboptimal or even degraded performance, and propose DELTA to mitigate them. First, the normalization statistics used in BN adapt are heavily influenced by the current test mini-batch, which can be one-sided and highly fluctuant. We introduce TBR to improve it using the (approximate) global statistics. Second, the optimization is highly skewed towards dominant classes, making the model more biased. DOT alleviates this problem by re-balancing the contributions of each class in an online manner. The combination of these two powerful tools results in our plug-in method DELTA, which achieves improvement in different scenarios (IS+CB, DS+CB, IS+CI, and DS+CI) at the same time.

ACKNOWLEDGMENTS

This work is supported in part by the National Natural Science Foundation of China under Grant 62171248, the R&D Program of Shenzhen under Grant JCYJ20220818101012025, the PCNL KEY project (PCL2021A07), and Shenzhen Science and Technology Innovation Commission (Research Center for Computer Network (Shenzhen) Ministry of Education).

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

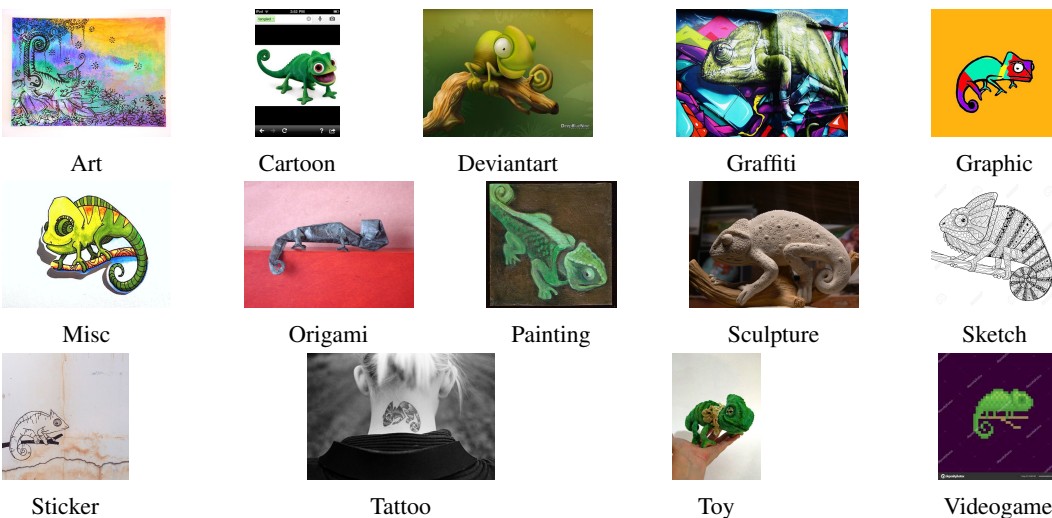

| Art | Cartoon | Deviantart | Graffiti | Graphic |
| --- | --- | --- | --- | --- |
| Misc | Origami | Painting | Sculpture | Sketch |
| Sticker | Tattoo | | Toy | Videogame |

Figure 6: Different renditions of class n01694178 (African chameleon) from ImageNet-R.

# A   APPENDIX

## A.1   DATASETS

Examples of ImageNet-R and ImageNet-C are shown in Figure 6 and Figure 7 respectively. ImageNet-R Hendrycks et al. (2021) holds a variety of renditions (*sketches, graphics, paintings, plastic objects, cartoons, graffiti, origami, patterns, deviantart, plush objects, sculptures, art, tattoos, toys, embroidery, video game*) of 200 ImageNet classes, resulting in 30,000 images. CIFAR100-C and ImageNet-C are established in Hendrycks & Dietterich (2019). CIFAR100-C contains 10,000 images with 15 corruption types: *Gaussian Noise (abbr. Gauss), Shot Noise (Shot), Impulse Noise (Impul), Defocus Blur (Defoc), Frosted Glass Blur (Glass), Motion Blur (Motion), Zoom Blur (Zoom), Snow, Frost, Fog, Brightness (Brit), Contrast (Contr), Elastic, Pixelate (Pixel), JPEG*. There are 50,000 images for each corruption type in ImageNet-C, others are the same as CIFAR100-C.

For the real-word applications with dependent and class-imbalanced test samples, we consider an automatic video content moderation task (e.g., for the short-video platform), which needs to recognize the categories of interest from the extracted frames. It is exactly a natural DS+CI scenario. We collect 1686 test videos from YouTube, which are annotated in YouTube-BoundingBoxes dataset. 49006 video segments are extracted from these videos and form the test stream in this experiment, named YTBB-sub here. We consider 21 categories. For the trained model, we adopt a model (ResNet18) trained on the related images from COCO dataset. Thus, there is a natural difference between the training domain and test domain. The consecutive video segments form the natural dependent samples (an object usually persists over several frames) as shown in Figure 8. Moreover, the test class distribution is also skewed naturally as shown in Figure 8.

To simulate dependent test samples, for each class, we sample $q_k \sim Dir_J(\rho)$, $q_k \in \mathbb{R}^J$ and allocate a $q_{k,j}$ proportion of the $k^{\text{th}}$ class samples to piece $j$, then the $J$ pieces are concatenated to form a test stream in our experiments ($J$ is set to 10 for all experiments); $\rho > 0$ is a concentration factor, when $\rho$ is small, samples belong to the same category will concentrate in test stream.

To simulate class-imbalanced test samples, we re-sample data points with an exponential decay in frequencies across different classes. We control the degree of imbalance through an imbalance factor $\pi$, which is defined as the ratio between sample sizes of the least frequent class and the most frequent class.

For DS+CI scenario, we mimic a class-imbalanced test set first, then the final test samples are dependently sampled from it.

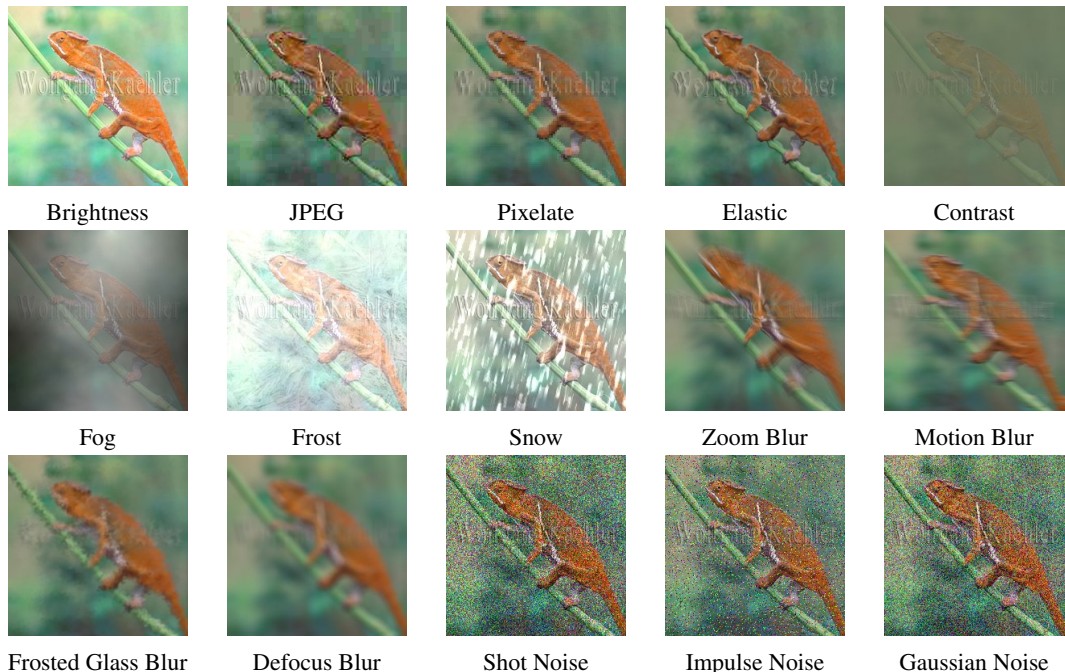

Figure 7: Different corruption types of class n01694178 (African chameleon) from ImageNet-C.

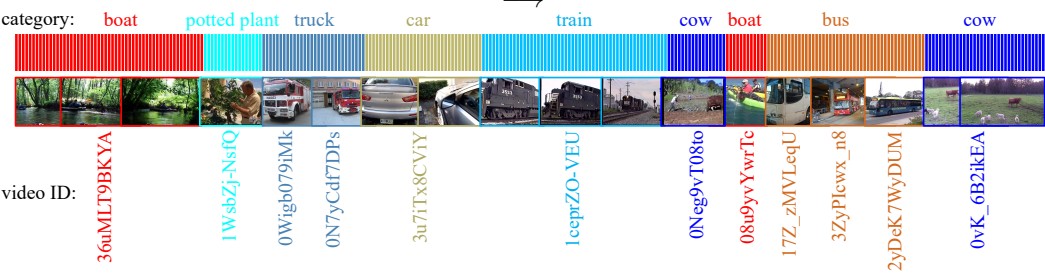

(a) The natural dependent samples in YTBB-sub. Each bar represents a sample, each color represents a category. The videos can be found at "https://www.youtube.com/watch?v={*the above video ID*}".

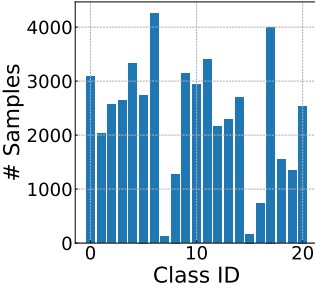

(b) The test class distribution.

Figure 8: Characters of YTBB-sub dataset.

## A.2    THE ALGORITHM DESCRIPTION OF TBR

We present the detailed algorithm description of TBR in Algorithm 2.

---

**Algorithm 2:** **T**est-time **B**atch **R**enormalization (TBR) module

---

**Input:** mini-batch test features $v \in \mathbb{R}^{B \times C \times S \times S'}$ with batch size $B$, $C$ channels, height $S$ and width $S'$; learnable affine parameters $\gamma \in \mathbb{R}^C$, $\beta \in \mathbb{R}^C$; current test-time moving mean $\hat{\mu}^{\text{ema}} \in \mathbb{R}^C$ and standard deviation $\hat{\sigma}^{\text{ema}} \in \mathbb{R}^C$; smoothing coefficient $\alpha$.

1   $\hat{\mu}^{\text{batch}}[c] = \frac{1}{BSS'} \sum_{b,s,s'} v[b,c,s,s']$, $c = 1, 2, \cdots, C$ // `get mean (for each channel)`

2   $\hat{\sigma}^{\text{batch}}[c] = \sqrt{\frac{1}{BSS'} \sum_{b,s,s'} (v[b,c,s,s'] - \hat{\mu}^{\text{batch}}[c])^2 + \epsilon}$, $c = 1, 2, \cdots, C$ // `get standard` `deviation (for each channel)`

3   $r = \frac{sg(\hat{\sigma}^{\text{batch}})}{\hat{\sigma}^{\text{ema}}}$ // `get r`

4   $d = \frac{sg(\hat{\mu}^{\text{batch}}) - \hat{\mu}^{\text{ema}}}{\hat{\sigma}^{\text{ema}}}$ // `get d`

5   $v^* = \frac{v - \hat{\mu}^{\text{batch}}}{\hat{\sigma}^{\text{batch}}} \cdot r + d$ // `normalize`

6   $v^{\star} = \gamma \cdot v^* + \beta$ // `scale and shift`

7   $\hat{\mu}^{\text{ema}} \leftarrow \alpha \cdot \hat{\mu}^{\text{ema}} + (1 - \alpha) \cdot sg(\hat{\mu}^{\text{batch}})$ // `update` $\hat{\mu}^{\text{ema}}$

8   $\hat{\sigma}^{\text{ema}} \leftarrow \alpha \cdot \hat{\sigma}^{\text{ema}} + (1 - \alpha) \cdot sg(\hat{\sigma}^{\text{batch}})$ // `update` $\hat{\sigma}^{\text{ema}}$

**Output:** $v^{\star}, \hat{\mu}^{\text{ema}}, \hat{\sigma}^{\text{ema}}$

---

### A.3   IMPLEMENTATIONS

We use Adam optimizer with learning rate of 1e-3, batch size of 200 for CIFAR100-C; SGD optimizer with learning rate of 2.5e-4, batch size of 64 for ImageNet-C/-R; SGD optimizer with learning rate of 2.5e-4, batch size of 200 for YTBB-sub. For DELTA, the hyper-parameters $\alpha$ and $\lambda$ are roughly selected from {0.9, 0.95, 0.99, 0.999} on validation sets, *e.g.*, the extra sets with corruption types outside the 15 types used in the benchmark. The smoothing coefficient $\alpha$ in TBR is set to 0.95 for CIFAR100-C and ImageNet-C/-R, 0.999 for YTBB-sub, $\lambda$ in DOT is set to 0.95 for ImageNet-C/-R and 0.9 for CIFAR100-C / YTBB-sub.

Then, we summarize the implementation details of the compared methods here, including BN adapt, PL, TENT, LAME, ETA, Ent-W, and CoTTA (CoTTA*). Unless otherwise specified, the optimizer, learning rate, and batch size are the same as those described in the main paper. For BN adapt, we follow the operation in Nado et al. (2020) and the official code of TENT (https://github.com/DequanWang/tent), *i.e.*, using the test-time normalization statistics completely. Though one can introduce a hyper-parameter to adjust the trade-off between current statistics and those inherited from the trained model ($a_0$) (Schneider et al., 2020), we find this strategy does not lead to significant improvement and its effect varies from dataset to dataset. For PL and TENT, besides the normalization statistics, we update the affine parameters in BN modules. The confidence threshold in PL is set to 0.4, which can produce acceptable results in most cases. We adopt/modify the official implementation https://github.com/DequanWang/tent to produce the results of TENT/PL. For LAME, we use the k-NN affinity matrix with 5 nearest neighbors following Boudiaf et al. (2022) and the official implementation https://github.com/fiveai/LAME. For ETA, the entropy constant threshold is set to $0.4 \times \ln K$ ($K$ is the number of task classes), and the similarity threshold is set to 0.4/0.05 for CIFAR/ImageNet experiments following the authors' suggestion and official implementation https://github.com/mr-eggplant/EATA. For Ent-W, the entropy constant threshold is set to 0.4 or 0.5 times $\ln K$. For CoTTA, the used random augmentations include color jitter, random affine, gaussian blur, random horizontal flip, and gaussian noise. 32 augmentations are employed in this method. The learning rate is set to 0.01 for ImageNet experiments following official implementation https://github.com/qinenergy/cotta. The restoration probability is set to 0.01 for CIFAR experiments and 0.001 for ImageNet experiments. The augmentation threshold is set to 0.72 for CIFAR experiments and 0.1 for ImageNet experiments. The exponential-moving-average factor is set to 0.999 for all experiments. CoTTA optimizes all learnable parameters during adaptation.

### A.4   ADDITIONAL ANALYSIS

**Fully test-time adaptation with small (test) batch size.** In the main paper, we report results with the default batch size following previous studies. Here, we study test-time adaptation with a much smaller batch size. The small batch size brings two serious challenges: the normalization statistics can be inaccurate and fluctuate dramatically; the gradient-based optimization can be noisy. Previ-

ous study (Niu et al., 2022) employs a sliding window with $L$ samples in total (including $L - B$ previous samples, assuming $L > B$, $L\%B = 0$ here) to perform adaptation. However, this strategy significantly increases the computational cost: $\frac{L}{B}\times$ forward and backward, *e.g.,* $64\times$ when $B = 1, L = 64$. We employ another strategy, called "fast-inference and slow-update". When the samples arrive, infer them instantly with the current model but do not perform adaptation; the model is updated with the recent $L$ samples every $\frac{L}{B}$ mini-batches. Thus, this strategy only needs $2\times$ forward and $1\times$ backward. Note that the two strategies both need to cache some recent test samples, which may be a bit against the "online adaptation". We evaluate TENT and DELTA on the IS+CB test stream of CIFAR100-C with batch sizes 128, 16, 8, and 1. The results are listed in Table 11. We find that TENT suffers from severe performance degeneration when the batch size is small, which is due to TENT always using the normalization statistics derived from the test mini-batches, thus it is still affected by the small batch size during "fast-inference". With the assistance of DELTA, the performance degradation can be significantly alleviated: it only drops by 0.7% (from 69.8% to 69.1%) when $B = 1$.

Table 11: Results (classification accuracy, %) with different batch sizes on IS+CB test stream of CIFAR100-C.

| Method | 128 | 16 | 8 | 1 |
|---|---|---|---|---|
| Source | 53.5 | 53.5 | 53.5 | 53.5 |
| TENT | 68.7 | 64.9 | 59.9 | 1.6 |
| TENT+DELTA | **69.8** | **69.4** | **69.0** | **69.1** |

**The initialization of TBR's normalization statistics.** As described in Section 3.2, TBR keeps the moving normalization statistics $\hat{\mu}^{\text{ema}}$, $\hat{\sigma}^{\text{ema}}$, we usually have two ways to initialize them: using the statistics $\hat{\mu}_1^{\text{batch}}$, $\hat{\sigma}_1^{\text{batch}}$ derived from the first test mini-batch (First); using the statistics $\mu^{\text{ema}}$, $\sigma^{\text{ema}}$ inherited from the trained model (Inherit). In the main paper, we use the "First" initialization strategy. However, it is worth noting that "First" is not reasonable for too small batch size. We perform TENT+DELTA with the above two initialization strategies and different batch sizes on the IS+CB test stream of CIFAR100-C. Figure 9 summaries the results, we can see that when the batch size is too small, using the inherited normalization statistics as initialization is better; when the batch size is acceptable (just $> 8$ for CIFAR100-C), using the "First" initialization strategy is superior.

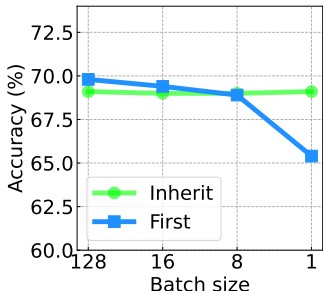

Figure 9: Comparison of two TBR initialization strategies on top of TENT+DELTA in IS+CB scenario on CIFAR100-C.

**Performance under different severity levels on CIFAR100-C and ImageNet-C.** In the main paper, for CIFAR100-C and ImageNet-C, we report the results with the highest severity level 5 following previous studies. Here, we investigate DELTA on top of TENT with different severity levels on CIFAR100-C (IS+CB scenario). Figure 10 presents the results. We observe that (i) as the corruption level increases, the model accuracy decreases; (ii) DELTA works well under all severity levels.

**Performance in extreme cases.** We examine the performance of DELTA with more extreme conditions: DS+CB with $\rho = 0.01$, IS+CI with $\pi = 0.001$. Table 12 shows DELTA can manage the intractable cases.

**Influence of random seeds.** As fully test-time adaptation is established based on a pre-trained model, *i.e.*, does not need random initialization; methods like PL, TENT, Ent-W, and our DELTA

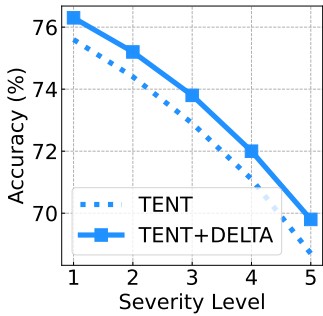

Figure 10: Comparison under different severity levels on CIFAR100-C.

Table 12: Performance in extreme cases.

|  | DS+CB ($\rho = 0.01$) | IS+CI ($\pi = 0.001$) |
|---|---|---|
| Source | 18.0 | 17.9 |
| BN adapt | 6.8 | 31.1 |
| ETA | 3.3 | 44.1 |
| LAME | 26.0 | 17.4 |
| CoTTA | 7.0 | 33.5 |
| CoTTA* | 7.2 | 33.6 |
| PL | 6.6 | 37.9 |
| +DELTA | 34.2 | 38.9 |
| TENT | 6.0 | 39.8 |
| +DELTA | 36.7 | 41.8 |
| Ent-W | 1.4 | 39.9 |
| +DELTA | 36.5 | 45.1 |

also do not bring random initialization. As a result, the adaptation results are always the same on one fixed test stream. However, the random seeds can affect sample order in our experiments. We study the influence of random seeds on Gauss and Shot data (IS+CB scenario) of ImageNet-C with seeds {2020, 2021, 2022, 2023}. The results of TENT and DELTA are summarized in Table 13, from which one can see the methods are not greatly affected by the sample order *within the same scenario*. For fair comparison, all methods are investigated under the same sample order for each specific scenario in our experiments.

Table 13: Influence of random seeds. Classification accuracies (%) are reported on two kinds of corrupted data (IS+CB) of ImageNet-C under four random seeds (2020, 2021, 2022, and 2023).

| Data | TENT | | | | TENT+DELTA | | | |
|---|---|---|---|---|---|---|---|---|
|  | 2020 | 2021 | 2022 | 2023 | 2020 | 2021 | 2022 | 2023 |
| Gauss | 28.672 | 28.434 | 28.774 | 28.796 | 31.186 | 30.916 | 31.270 | 31.208 |
| Shot | 30.536 | 30.496 | 30.370 | 30.458 | 33.146 | 33.140 | 33.124 | 32.994 |

**Ablation on DOT.** We examine the performance of DOT with another way to get the sample weights (Line 5,6 in Algorithm 1). One can discard line 5 and modify line 6 to adopt the original soft probabilities: $\omega_{m_t+b} = \sum_{k=1}^{K} 1/(z_{t-1}[k] + \epsilon) \cdot p_{m_t+b}[k]$. We compare the hard label strategy (Algorithm 1) with the soft one in Table 14 (on the basis of Enw-W+TBR, on ImageNet-C). We find that both strategies work well in all scenarios, demonstrating the effectiveness of the idea of DOT. The performance of the soft strategy is slightly worse than the hard strategy in some scenarios. However, we think it is difficult to say "hard labels are necessarily better than soft labels" or "soft labels are necessarily better than hard labels", for example, the two strategies both exist in recent semi-supervised methods: hard label in FixMatch, soft label in UDA.

Table 14: Ablation on DOT.

| | IS+CB | DS+CB $\rho = 1.0$ | DS+CB $\rho = 0.5$ | DS+CB $\rho = 0.1$ | IS+CI $\pi = 0.1$ | IS+CI $\pi = 0.05$ | DS+CI $\rho = 0.5, \pi = 0.1$ | DS+CI $\rho = 0.5, \pi = 0.05$ |
|---|---|---|---|---|---|---|---|---|
| Hard | 49.9 | 48.3 | 47.4 | 43.2 | 48.4 | 47.7 | 45.4 | 44.8 |
| Soft | 49.7 | 48.0 | 47.3 | 43.0 | 48.3 | 47.5 | 45.1 | 44.5 |

A.5 RESULTS OF EACH CORRUPTION TYPE ON CIFAR100-C.

Table 2 has compared the usages of different normalization statistics, we further provide the detailed results of all corruption types in Table 15.

Table 16 presents the results of all corruption types under different batch sizes and the two initialization strategies for normalization statistics in TBR, the averaged results have been illustrated in Table 11 and Figure 9 respectively.

Table 17 summarises the detailed performance on IS+CB test stream with different severity levels.

Table 18 compares the test-time adaptation methods in IS+CB scenario; Table 19 for DS+CB test stream ($\rho = 1.0$), Table 20 for DS+CB test stream ($\rho = 0.5$), Table 21 for DS+CB test stream ($\rho = 0.1$); Table 22, 23 for IS+CI data with $\pi = 0.1$, $\pi = 0.05$; Table 24 / Table 25 for DS+CI test data with $\rho = 0.5$ and $\pi = 0.1$ / $\pi = 0.05$.

A.6 RESULTS OF EACH CORRUPTION TYPE ON IMAGENET-C.

Table 26 compares the test-time adaptation methods in IS+CB scenario and Table 27 further compares them with different model architectures; Table 28, Table 29, and Table 30 for DS+CB test streams with $\rho = 1.0$, $\rho = 0.5$ and $\rho = 0.1$, respectively; Table 31, 32 for IS+CI data with $\pi = 0.1$, $\pi = 0.05$; Table 33 / Table 34 for DS+CI test data with $\rho = 0.5$ and $\pi = 0.1$ / $\pi = 0.05$. The results in Table 15-Table 34 are obtained with seed 2020.

Table 15: Comparison of the normalization statistics on IS+CB and DS+CB test streams of CIFAR100-C with $B = 128$ in terms of classification accuracy (%).

| Method | Gauss | Shot | Impul | Defoc | Glass | Motion | Zoom | Snow | Frost | Fog | Brit | Contr | Elastic | Pixel | JPEG | Avg |
|---|---|---|---|---|---|---|---|---|---|---|---|---|---|---|---|---|
| Source | 27.0 | 32.0 | 60.6 | 70.7 | 45.9 | 69.2 | 71.2 | 60.5 | 54.2 | 49.7 | 70.5 | 44.9 | 62.8 | 25.3 | 58.8 | 53.5 |
| *IS+CB scenario* | | | | | | | | | | | | | | | | |
| BN adapt | 57.6 | 59.0 | 56.9 | 72.3 | 58.0 | 70.3 | 71.8 | 64.8 | 64.8 | 58.1 | 73.3 | 69.7 | 64.0 | 66.7 | 58.4 | 64.4 |
| BN adapt+TEMA | 58.0 | 59.7 | 57.1 | 72.5 | 58.6 | 70.3 | 72.5 | 65.3 | 65.5 | 58.3 | 74.1 | 70.2 | 64.4 | 67.0 | 59.2 | 64.9 |
| TENT | 62.4 | 64.7 | 67.3 | 74.3 | 62.5 | 72.4 | 74.2 | 69.4 | 67.6 | 66.8 | 75.6 | 71.8 | 66.9 | 71.3 | 62.6 | 68.7 |
| TENT+TEMA | 19.4 | 14.9 | 16.4 | 31.9 | 14.8 | 25.0 | 28.9 | 24.3 | 25.0 | 19.3 | 31.1 | 24.3 | 25.5 | 26.0 | 18.2 | 23.0 |
| TENT+TBR | 62.1 | 64.7 | 67.7 | 74.6 | 62.0 | 72.6 | 74.0 | 69.7 | 67.9 | 67.8 | 76.2 | 71.6 | 67.1 | 71.8 | 63.3 | 68.9 |
| *DS+CB scenario* | | | | | | | | | | | | | | | | |
| BN adapt | 24.1 | 24.7 | 23.4 | 30.2 | 23.2 | 29.9 | 29.8 | 26.6 | 27.2 | 24.2 | 30.0 | 28.6 | 25.7 | 27.8 | 23.8 | 26.6 |
| BN adapt+TEMA | 56.0 | 57.8 | 55.2 | 70.7 | 56.7 | 68.6 | 70.2 | 63.2 | 63.6 | 56.6 | 71.7 | 67.8 | 62.2 | 64.8 | 57.2 | 62.8 |
| TENT | 21.2 | 22.7 | 21.9 | 26.6 | 20.0 | 25.5 | 26.6 | 23.0 | 22.2 | 21.7 | 26.3 | 21.6 | 21.7 | 24.7 | 20.3 | 23.1 |
| TENT+TEMA | 18.0 | 17.3 | 15.2 | 34.2 | 18.6 | 26.3 | 36.6 | 18.9 | 27.2 | 24.6 | 36.2 | 25.8 | 26.5 | 28.6 | 20.4 | 25.0 |
| TENT+TBR | 55.8 | 60.0 | 58.8 | 70.7 | 57.2 | 67.4 | 69.7 | 64.4 | 62.8 | 60.2 | 71.5 | 64.0 | 60.9 | 67.1 | 56.4 | 63.1 |

Table 16: Comparison of different batch sizes and the initialization strategies for TBR's normalization statistics on IS+CB test stream of CIFAR100-C in terms of classification accuracy (%).

| Method | Init | Gauss | Shot | Impul | Defoc | Glass | Motion | Zoom | Snow | Frost | Fog | Brit | Contr | Elastic | Pixel | JPEG | Avg |
|---|---|---|---|---|---|---|---|---|---|---|---|---|---|---|---|---|---|
| Source | – | 27.0 | 32.0 | 60.6 | 70.7 | 45.9 | 69.2 | 71.2 | 60.5 | 54.2 | 49.7 | 70.5 | 44.9 | 62.8 | 25.3 | 58.8 | 53.5 |
| TENT, $B$=128 | – | 62.4 | 64.7 | 67.3 | 74.3 | 62.5 | 72.4 | 74.2 | 69.4 | 67.6 | 66.8 | 75.6 | 71.8 | 66.9 | 71.3 | 62.6 | 68.7 |
| TENT, $B$=16 | – | 58.7 | 61.0 | 63.8 | 70.8 | 58.7 | 68.8 | 70.3 | 65.8 | 64.1 | 63.3 | 72.2 | 66.9 | 62.7 | 67.6 | 59.4 | 64.9 |
| TENT, $B$=8 | – | 54.0 | 56.1 | 58.6 | 65.9 | 53.0 | 64.1 | 65.4 | 61.0 | 58.6 | 57.8 | 67.1 | 62.9 | 58.1 | 62.8 | 53.8 | 59.9 |
| TENT, $B$=1 | – | 1.5 | 1.5 | 1.6 | 1.6 | 1.6 | 1.8 | 1.7 | 1.8 | 1.6 | 1.5 | 1.6 | 1.6 | 1.5 | 1.8 | 1.6 | 1.6 |
| TENT+DELTA, $B$=128 | Inherit | 62.4 | 63.9 | 69.0 | 75.3 | 63.2 | 73.2 | 74.8 | 69.8 | 69.2 | 66.6 | 76.0 | 71.3 | 67.4 | 69.7 | 64.3 | 69.1 |
| TENT+DELTA, $B$=128 | First | 64.0 | 66.0 | 69.1 | 75.3 | 63.3 | 73.0 | 74.6 | 70.3 | 69.4 | 68.1 | 76.7 | 72.9 | 67.6 | 72.3 | 64.6 | 69.8 |
| TENT+DELTA, $B$=16 | Inherit | 62.3 | 64.0 | 69.1 | 75.2 | 63.1 | 73.3 | 74.8 | 69.6 | 69.3 | 66.7 | 76.0 | 70.8 | 67.3 | 69.7 | 64.3 | 69.0 |
| TENT+DELTA, $B$=16 | First | 63.5 | 65.5 | 68.2 | 74.8 | 63.2 | 72.7 | 74.6 | 70.2 | 69.3 | 67.7 | 76.2 | 72.4 | 67.5 | 71.9 | 63.9 | 69.4 |
| TENT+DELTA, $B$=8 | Inherit | 62.4 | 64.0 | 69.0 | 75.2 | 63.1 | 73.3 | 74.8 | 69.7 | 69.4 | 66.6 | 75.9 | 71.2 | 67.3 | 69.6 | 64.2 | 69.0 |
| TENT+DELTA, $B$=8 | First | 63.1 | 65.1 | 67.1 | 74.8 | 62.4 | 72.6 | 74.3 | 69.9 | 69.2 | 67.2 | 75.7 | 71.2 | 67.0 | 71.6 | 63.0 | 68.9 |
| TENT+DELTA, $B$=1 | Inherit | 62.2 | 64.0 | 68.9 | 75.3 | 63.1 | 73.2 | 74.7 | 69.7 | 69.4 | 66.6 | 76.1 | 71.6 | 67.4 | 69.6 | 64.4 | 69.1 |
| TENT+DELTA, $B$=1 | First | 60.0 | 62.0 | 64.4 | 71.4 | 59.5 | 69.0 | 71.4 | 65.6 | 65.7 | 62.9 | 72.6 | 64.0 | 63.6 | 68.6 | 59.8 | 65.4 |

Table 17: Classification accuracy (%) on IS+CB test stream of CIFAR100-C with different severity levels ($B = 128$).

| Method | Level | Gauss | Shot | Impul | Defoc | Glass | Motion | Zoom | Snow | Frost | Fog | Brit | Contr | Elastic | Pixel | JPEG | Avg |
|---|---|---|---|---|---|---|---|---|---|---|---|---|---|---|---|---|---|
| Source | 1 | 64.2 | 70.9 | 77.6 | 78.9 | 54.4 | 77.0 | 76.8 | 76.5 | 74.2 | 78.4 | 78.7 | 78.2 | 74.4 | 76.5 | 70.5 | 73.8 |
|  | 2 | 49.3 | 63.6 | 75.3 | 78.1 | 56.6 | 75.1 | 76.6 | 69.9 | 69.7 | 76.1 | 77.7 | 75.2 | 75.0 | 72.3 | 65.8 | 70.4 |
|  | 3 | 36.5 | 47.2 | 73.1 | 76.8 | 60.6 | 72.3 | 75.4 | 69.6 | 62.1 | 72.3 | 76.6 | 71.9 | 73.7 | 69.1 | 64.1 | 66.8 |
|  | 4 | 31.2 | 40.6 | 68.0 | 75.2 | 39.5 | 72.4 | 74.0 | 65.2 | 61.1 | 65.8 | 74.9 | 65.7 | 68.9 | 52.3 | 62.5 | 61.2 |
|  | 5 | 27.0 | 32.0 | 60.6 | 70.7 | 45.9 | 69.2 | 71.2 | 60.5 | 54.2 | 49.7 | 70.5 | 44.9 | 62.8 | 25.3 | 58.8 | 53.5 |
| TENT | 1 | 73.8 | 75.8 | 77.2 | 77.9 | 69.4 | 76.6 | 77.1 | 76.3 | 75.8 | 78.0 | 77.9 | 77.3 | 73.3 | 76.3 | 71.3 | 75.6 |
|  | 2 | 70.2 | 73.7 | 76.0 | 77.9 | 69.6 | 75.6 | 76.8 | 73.4 | 74.0 | 76.2 | 77.8 | 75.2 | 74.8 | 76.0 | 68.5 | 74.4 |
|  | 3 | 66.7 | 70.0 | 74.5 | 77.4 | 68.7 | 73.7 | 76.4 | 72.1 | 71.4 | 74.9 | 77.3 | 74.4 | 73.9 | 75.6 | 66.3 | 72.9 |
|  | 4 | 64.4 | 68.3 | 70.9 | 76.3 | 62.6 | 74.2 | 75.3 | 70.5 | 70.4 | 72.2 | 76.9 | 74.1 | 70.5 | 74.3 | 65.2 | 71.1 |
|  | 5 | 62.4 | 64.7 | 67.3 | 74.3 | 62.5 | 72.4 | 74.2 | 69.4 | 67.6 | 66.8 | 75.6 | 71.8 | 66.9 | 71.3 | 62.6 | 68.7 |
| TENT+DELTA | 1 | 74.4 | 76.1 | 78.0 | 78.7 | 70.3 | 77.2 | 77.7 | 77.1 | 76.6 | 78.6 | 78.5 | 78.3 | 74.5 | 77.1 | 72.0 | 76.3 |
|  | 2 | 70.9 | 74.7 | 76.4 | 78.4 | 70.3 | 75.8 | 77.4 | 74.5 | 74.8 | 76.9 | 78.4 | 76.9 | 75.4 | 77.0 | 69.8 | 75.2 |
|  | 3 | 67.8 | 70.2 | 75.3 | 77.9 | 69.8 | 74.5 | 76.8 | 73.1 | 72.5 | 75.6 | 78.1 | 76.6 | 74.8 | 76.5 | 67.6 | 73.8 |
|  | 4 | 65.6 | 69.2 | 72.5 | 76.9 | 63.4 | 74.9 | 76.0 | 71.0 | 71.3 | 73.2 | 77.9 | 75.8 | 71.3 | 75.1 | 66.4 | 72.0 |
|  | 5 | 64.0 | 66.0 | 69.1 | 75.3 | 63.3 | 73.0 | 74.6 | 70.3 | 69.4 | 68.1 | 76.7 | 72.9 | 67.6 | 72.3 | 64.6 | 69.8 |

Table 18: Classification accuracy (%) on IS+CB test stream of CIFAR100-C.

| Method | Gauss | Shot | Impul | Defoc | Glass | Motion | Zoom | Snow | Frost | Fog | Brit | Contr | Elastic | Pixel | JPEG | Avg |
|---|---|---|---|---|---|---|---|---|---|---|---|---|---|---|---|---|
| Source | 27.0 | 32.0 | 60.6 | 70.7 | 45.9 | 69.2 | 71.2 | 60.5 | 54.2 | 49.7 | 70.5 | 44.9 | 62.8 | 25.3 | 58.8 | 53.5 |
| BN adapt | 57.9 | 59.3 | 57.3 | 72.4 | 58.2 | 70.3 | 72.1 | 65.1 | 65.0 | 58.5 | 73.5 | 69.7 | 64.3 | 67.1 | 58.8 | 64.6 |
| ETA | 63.2 | 65.3 | 66.9 | 75.1 | 63.2 | 73.1 | 74.9 | 70.0 | 69.7 | 66.9 | 76.5 | 73.6 | 67.7 | 72.0 | 64.0 | 69.5 |
| LAME | 24.1 | 29.0 | 59.2 | 69.0 | 42.8 | 67.0 | 68.9 | 58.3 | 50.7 | 46.5 | 67.6 | 39.2 | 60.3 | 21.4 | 56.7 | 50.7 |
| CoTTA | 60.0 | 61.8 | 60.1 | 72.6 | 60.2 | 70.5 | 72.3 | 64.8 | 65.5 | 56.7 | 73.6 | 69.9 | 64.3 | 68.4 | 62.6 | 65.5 |
| CoTTA* | 60.0 | 62.2 | 60.8 | 73.2 | 62.3 | 71.9 | 73.7 | 67.0 | 67.9 | 59.8 | 75.4 | 72.9 | 67.5 | 72.0 | 66.6 | 67.5 |
| PL | 61.8 | 64.5 | 65.0 | 74.6 | 62.0 | 72.1 | 74.2 | 68.9 | 68.4 | 64.8 | 75.5 | 72.0 | 66.8 | 70.8 | 61.9 | 68.2 |
| PL+DELTA | 62.8 | 64.8 | 66.3 | 74.3 | 62.7 | 72.7 | 74.6 | 69.4 | 68.5 | 65.7 | 75.5 | 72.8 | 66.8 | 71.3 | 62.7 | 68.7 |
| TENT | 62.8 | 65.4 | 66.3 | 74.8 | 62.3 | 72.8 | 74.6 | 69.6 | 68.6 | 66.8 | 76.1 | 72.3 | 67.3 | 71.6 | 63.5 | 69.0 |
| TENT+TBR | 62.5 | 64.9 | 67.0 | 74.8 | 62.1 | 72.9 | 74.3 | 69.8 | 68.3 | 66.8 | 76.6 | 72.0 | 67.1 | 71.9 | 63.0 | 68.9 |
| TENT+DOT | 63.6 | 65.7 | 66.9 | 75.1 | 63.0 | 73.1 | 74.8 | 69.8 | 69.0 | 67.1 | 76.2 | 73.2 | 67.6 | 71.8 | 63.8 | 69.4 |
| TENT+DELTA | 63.5 | 65.7 | 67.8 | 75.1 | 63.3 | 73.1 | 74.7 | 70.3 | 69.3 | 67.4 | 76.8 | 72.8 | 67.8 | 72.3 | 63.6 | 69.6 |
| Ent-W | 63.5 | 65.5 | 67.2 | 75.1 | 63.2 | 73.1 | 74.8 | 70.1 | 69.8 | 67.1 | 76.6 | 73.5 | 67.7 | 72.0 | 64.1 | 69.6 |
| Ent-W+TBR+Div-W(0.05) | 60.3 | 63.5 | 63.8 | 73.5 | 60.8 | 71.8 | 73.7 | 68.6 | 66.2 | 63.8 | 74.9 | 71.8 | 66.7 | 69.9 | 61.7 | 67.4 |
| Ent-W+TBR+Div-W(0.1) | 63.5 | 65.3 | 67.0 | 75.2 | 62.7 | 72.8 | 74.7 | 70.0 | 69.4 | 66.7 | 76.1 | 73.2 | 67.1 | 71.7 | 63.8 | 69.3 |
| Ent-W+TBR+Div-W(0.2) | 63.8 | 65.6 | 68.1 | 75.3 | 63.1 | 73.4 | 75.0 | 70.7 | 70.0 | 67.4 | 77.0 | 73.5 | 67.3 | 72.5 | 64.1 | 69.8 |
| Ent-W+TBR+Div-W(0.4) | 63.6 | 65.4 | 68.2 | 75.3 | 63.1 | 73.3 | 75.0 | 70.8 | 69.9 | 67.3 | 76.9 | 73.6 | 67.1 | 72.6 | 64.0 | 69.7 |
| Ent-W+TBR+LA | 64.0 | 65.9 | 68.4 | 75.4 | 63.5 | 73.6 | 75.1 | 71.0 | 70.2 | 67.6 | 77.0 | 73.8 | 67.6 | 72.8 | 64.5 | 70.0 |
| Ent-W+TBR+Sample-drop | 64.1 | 66.2 | 68.6 | 75.8 | 63.8 | 73.5 | 75.5 | 70.9 | 70.2 | 67.7 | 77.0 | 73.9 | 68.2 | 72.8 | 64.4 | 70.2 |
| Ent-W+DELTA | 64.2 | 66.1 | 68.5 | 75.6 | 63.6 | 73.5 | 75.2 | 71.2 | 70.3 | 68.0 | 77.1 | 74.0 | 68.0 | 72.8 | 64.7 | 70.2 |

Table 19: Classification accuracy (%) on DS+CB ($\rho = 1.0$) test stream of CIFAR100-C.

| Method | Gauss | Shot | Impul | Defoc | Glass | Motion | Zoom | Snow | Frost | Fog | Brit | Contr | Elastic | Pixel | JPEG | Avg |
|---|---|---|---|---|---|---|---|---|---|---|---|---|---|---|---|---|
| Source | 27.0 | 32.0 | 60.6 | 70.7 | 45.9 | 69.2 | 71.2 | 60.5 | 54.2 | 49.7 | 70.5 | 44.9 | 62.8 | 25.3 | 58.8 | 53.5 |
| BN adapt | 47.2 | 47.8 | 46.2 | 59.5 | 47.2 | 57.4 | 58.8 | 52.2 | 53.2 | 46.7 | 59.9 | 57.4 | 51.9 | 54.4 | 46.9 | 52.4 |
| ETA | 50.1 | 51.2 | 52.6 | 60.3 | 49.4 | 58.7 | 60.2 | 55.2 | 54.7 | 51.2 | 61.0 | 58.0 | 53.2 | 56.8 | 50.1 | 54.9 |
| LAME | 28.0 | 34.2 | 68.6 | 80.1 | 52.1 | 78.6 | 80.7 | 70.5 | 61.1 | 57.4 | 79.3 | 49.2 | 73.3 | 26.1 | 68.7 | 60.5 |
| CoTTA | 49.1 | 51.2 | 49.7 | 57.7 | 49.3 | 56.8 | 58.6 | 52.8 | 53.6 | 46.6 | 60.0 | 53.6 | 52.6 | 57.3 | 50.9 | 53.3 |
| CoTTA* | 49.1 | 51.3 | 49.5 | 57.4 | 49.8 | 56.6 | 58.4 | 53.1 | 54.1 | 46.9 | 59.1 | 54.2 | 53.3 | 57.1 | 52.8 | 53.5 |
| PL | 49.9 | 50.5 | 51.5 | 60.0 | 48.3 | 58.2 | 60.4 | 54.2 | 54.6 | 50.4 | 60.7 | 57.4 | 53.0 | 56.5 | 49.2 | 54.3 |
| PL+DELTA | 61.3 | 62.9 | 64.4 | 73.9 | 61.8 | 71.7 | 74.0 | 68.1 | 68.0 | 63.9 | 74.9 | 71.2 | 66.2 | 70.1 | 62.2 | 67.6 |
| TENT | 49.3 | 50.7 | 52.6 | 59.9 | 48.7 | 57.8 | 59.5 | 53.8 | 53.5 | 50.7 | 60.2 | 56.8 | 52.7 | 56.1 | 49.4 | 54.1 |
| TENT+DELTA | 62.3 | 64.4 | 66.7 | 74.5 | 62.6 | 72.0 | 74.3 | 68.9 | 68.5 | 65.8 | 75.6 | 72.0 | 66.8 | 71.4 | 63.4 | 68.6 |
| Ent-W | 50.0 | 51.3 | 52.9 | 60.3 | 49.3 | 58.9 | 60.3 | 54.9 | 54.9 | 51.1 | 61.0 | 57.8 | 53.1 | 56.7 | 50.0 | 54.8 |
| Ent-W+DELTA | 62.7 | 64.9 | 67.4 | 74.6 | 62.7 | 72.6 | 74.4 | 69.6 | 69.2 | 66.1 | 75.7 | 72.4 | 66.8 | 71.7 | 64.2 | 69.0 |

Table 20: Classification accuracy (%) on DS+CB ($\rho = 0.5$) test stream of CIFAR100-C.

| Method | Gauss | Shot | Impul | Defoc | Glass | Motion | Zoom | Snow | Frost | Fog | Brit | Contr | Elastic | Pixel | JPEG | Avg |
|---|---|---|---|---|---|---|---|---|---|---|---|---|---|---|---|---|
| Source | 27.0 | 32.0 | 60.6 | 70.7 | 45.9 | 69.2 | 71.2 | 60.5 | 54.2 | 49.7 | 70.5 | 44.9 | 62.8 | 25.3 | 58.8 | 53.5 |
| BN adapt | 43.8 | 45.2 | 43.9 | 56.2 | 44.5 | 54.7 | 55.5 | 49.1 | 50.0 | 43.9 | 57.0 | 54.2 | 48.7 | 51.2 | 45.0 | 49.5 |
| ETA | 45.8 | 47.5 | 48.9 | 56.4 | 45.3 | 54.5 | 55.8 | 51.2 | 51.2 | 48.1 | 57.4 | 53.8 | 49.4 | 53.1 | 45.9 | 50.9 |
| LAME | 28.5 | 34.8 | 69.7 | 80.8 | 53.5 | 79.6 | 81.8 | 71.9 | 62.7 | 58.6 | 81.1 | 50.6 | 74.5 | 26.9 | 69.5 | 61.6 |
| CoTTA | 46.9 | 48.3 | 46.5 | 55.1 | 46.6 | 54.2 | 55.2 | 49.3 | 50.6 | 43.4 | 56.9 | 50.8 | 49.3 | 54.2 | 48.3 | 50.4 |
| CoTTA* | 46.9 | 48.4 | 46.5 | 54.5 | 47.2 | 53.8 | 54.4 | 50.0 | 51.2 | 43.9 | 56.1 | 51.5 | 50.3 | 53.9 | 49.4 | 50.5 |
| PL | 45.4 | 47.0 | 47.8 | 56.0 | 46.3 | 54.3 | 55.7 | 50.8 | 51.3 | 47.2 | 57.1 | 52.6 | 49.4 | 52.7 | 45.9 | 50.6 |
| PL+DELTA | 61.3 | 62.5 | 63.2 | 73.1 | 61.3 | 70.8 | 73.6 | 68.0 | 67.0 | 63.3 | 74.5 | 70.0 | 65.7 | 69.7 | 61.2 | 67.0 |
| TENT | 44.8 | 46.7 | 48.4 | 55.9 | 45.5 | 54.0 | 55.2 | 50.0 | 50.1 | 47.3 | 56.6 | 52.2 | 48.4 | 52.6 | 45.6 | 50.2 |
| TENT+TBR | 59.7 | 62.4 | 64.6 | 73.3 | 60.7 | 70.7 | 72.9 | 67.3 | 66.6 | 64.2 | 74.2 | 68.9 | 65.0 | 69.5 | 61.0 | 66.7 |
| TENT+DOT | 45.9 | 47.5 | 49.3 | 56.8 | 46.4 | 54.8 | 55.8 | 50.8 | 51.1 | 48.2 | 57.4 | 53.6 | 49.6 | 53.0 | 46.4 | 51.1 |
| TENT+DELTA | 61.3 | 63.5 | 65.5 | 73.9 | 62.2 | 71.5 | 73.8 | 68.3 | 67.5 | 65.6 | 74.8 | 70.8 | 66.1 | 70.4 | 62.0 | 67.8 |
| Ent-W | 45.8 | 47.5 | 49.0 | 56.3 | 45.5 | 54.5 | 55.6 | 51.6 | 51.1 | 48.3 | 57.2 | 53.8 | 49.3 | 53.0 | 45.9 | 51.0 |
| Ent-W+TBR+Div-W(0.05) | 61.5 | 64.0 | 64.1 | 73.8 | 60.7 | 71.7 | 73.5 | 67.6 | 68.2 | 64.2 | 74.8 | 71.0 | 66.4 | 70.3 | 62.2 | 67.6 |
| Ent-W+TBR+Div-W(0.1) | 62.4 | 63.9 | 65.7 | 74.3 | 61.9 | 71.8 | 73.8 | 68.3 | 68.5 | 65.0 | 75.0 | 71.1 | 66.2 | 70.5 | 62.4 | 68.1 |
| Ent-W+TBR+Div-W(0.2) | 61.0 | 63.5 | 65.5 | 73.6 | 60.8 | 71.2 | 72.9 | 68.0 | 67.9 | 65.1 | 74.5 | 70.7 | 65.7 | 70.2 | 62.2 | 67.5 |
| Ent-W+TBR+Div-W(0.4) | 60.5 | 63.4 | 65.2 | 73.4 | 60.3 | 71.2 | 72.9 | 67.6 | 67.9 | 65.0 | 74.4 | 70.6 | 65.3 | 69.9 | 61.8 | 67.3 |
| Ent-W+TBR+LA | 60.0 | 62.8 | 64.2 | 72.3 | 59.5 | 69.9 | 71.7 | 66.8 | 66.8 | 63.9 | 73.3 | 69.4 | 64.7 | 69.1 | 61.0 | 66.4 |
| Ent-W+TBR+Sample-drop | 61.9 | 64.2 | 65.6 | 74.2 | 61.8 | 71.7 | 73.8 | 68.3 | 68.4 | 65.5 | 74.9 | 71.4 | 66.2 | 70.7 | 62.7 | 68.1 |
| Ent-W+DELTA | 61.9 | 64.2 | 66.0 | 74.3 | 61.9 | 71.9 | 73.9 | 68.3 | 68.5 | 65.9 | 74.9 | 71.5 | 66.4 | 70.9 | 62.9 | 68.2 |

Table 21: Classification accuracy (%) on DS+CB ($\rho = 0.1$) test stream of CIFAR100-C.

| Method | Gauss | Shot | Impul | Defoc | Glass | Motion | Zoom | Snow | Frost | Fog | Brit | Contr | Elastic | Pixel | JPEG | Avg |
|---|---|---|---|---|---|---|---|---|---|---|---|---|---|---|---|---|
| Source | 27.0 | 32.0 | 60.6 | 70.7 | 45.9 | 69.2 | 71.2 | 60.5 | 54.2 | 49.7 | 70.5 | 44.9 | 62.8 | 25.3 | 58.8 | 53.5 |
| BN adapt | 31.5 | 32.8 | 31.4 | 40.4 | 31.0 | 39.3 | 40.0 | 35.3 | 35.7 | 31.5 | 40.7 | 38.0 | 34.5 | 36.7 | 31.7 | 35.4 |
| ETA | 31.2 | 32.2 | 32.6 | 38.4 | 30.4 | 37.7 | 38.4 | 34.6 | 34.7 | 32.2 | 39.4 | 36.3 | 33.2 | 36.3 | 31.2 | 34.6 |
| LAME | 30.3 | 36.9 | 73.2 | 84.3 | 57.2 | 83.3 | 85.0 | 76.7 | 66.4 | 63.1 | 84.7 | 54.3 | 79.2 | 28.6 | 73.9 | 65.1 |
| CoTTA | 33.7 | 34.8 | 34.1 | 39.5 | 33.2 | 39.3 | 40.1 | 36.3 | 36.8 | 31.8 | 39.9 | 36.8 | 35.8 | 39.6 | 35.3 | 36.5 |
| CoTTA* | 33.7 | 35.0 | 33.7 | 39.0 | 33.2 | 38.7 | 39.4 | 35.8 | 36.6 | 31.8 | 39.1 | 36.1 | 35.6 | 38.8 | 35.5 | 36.1 |
| PL | 32.2 | 32.0 | 32.5 | 39.2 | 30.7 | 37.8 | 39.2 | 35.0 | 35.0 | 32.1 | 39.5 | 36.8 | 33.5 | 36.9 | 31.2 | 34.9 |
| PL+DELTA | 59.2 | 61.0 | 61.6 | 72.0 | 58.8 | 70.1 | 72.2 | 66.2 | 65.2 | 61.6 | 72.8 | 69.2 | 63.5 | 67.4 | 59.6 | 65.4 |
| TENT | 29.9 | 31.1 | 32.1 | 37.8 | 30.0 | 36.6 | 37.6 | 33.6 | 33.3 | 31.4 | 38.1 | 34.4 | 32.0 | 36.0 | 30.1 | 33.6 |
| TENT+DELTA | 60.3 | 62.7 | 63.1 | 72.7 | 60.2 | 70.7 | 72.1 | 66.7 | 65.9 | 63.4 | 73.6 | 69.8 | 64.5 | 68.5 | 60.2 | 66.3 |
| Ent-W | 31.0 | 32.1 | 32.7 | 38.3 | 30.1 | 37.7 | 38.5 | 34.5 | 34.5 | 32.0 | 39.3 | 36.0 | 32.9 | 36.2 | 30.6 | 34.4 |
| Ent-W+DELTA | 60.2 | 62.3 | 63.5 | 72.3 | 59.6 | 70.0 | 72.3 | 67.3 | 66.3 | 63.2 | 73.7 | 70.3 | 64.2 | 69.2 | 60.5 | 66.3 |

Table 22: Classification accuracy (%) on IS+CI ($\pi = 0.1$) test stream of CIFAR100-C.

| Method | Gauss | Shot | Impul | Defoc | Glass | Motion | Zoom | Snow | Frost | Fog | Brit | Contr | Elastic | Pixel | JPEG | Avg |
|---|---|---|---|---|---|---|---|---|---|---|---|---|---|---|---|---|
| Source | 26.2 | 31.7 | 60.1 | 70.3 | 45.7 | 69.5 | 71.2 | 60.1 | 53.9 | 49.7 | 69.7 | 45.1 | 62.5 | 25.6 | 58.9 | 53.3 |
| BN adapt | 58.0 | 58.8 | 56.7 | 71.6 | 58.3 | 69.6 | 71.5 | 64.9 | 65.1 | 58.6 | 72.9 | 68.7 | 64.4 | 66.3 | 58.5 | 64.3 |
| ETA | 62.6 | 63.7 | 65.2 | 73.6 | 62.9 | 71.6 | 73.8 | 68.6 | 68.9 | 65.6 | 75.2 | 72.1 | 65.9 | 70.7 | 62.7 | 68.2 |
| LAME | 23.5 | 28.6 | 59.4 | 68.8 | 43.3 | 67.1 | 68.8 | 58.2 | 50.9 | 46.6 | 67.1 | 39.4 | 60.4 | 21.6 | 56.7 | 50.7 |
| CoTTA | 59.8 | 61.3 | 59.7 | 71.8 | 59.8 | 69.6 | 71.6 | 64.4 | 65.3 | 56.5 | 73.1 | 68.5 | 64.2 | 68.2 | 62.5 | 65.1 |
| CoTTA* | 59.8 | 61.9 | 60.1 | 72.0 | 61.7 | 70.9 | 72.6 | 66.2 | 67.4 | 59.1 | 74.5 | 71.3 | 67.3 | 71.5 | 66.3 | 66.8 |
| PL | 61.7 | 62.3 | 62.8 | 73.1 | 61.7 | 71.1 | 73.6 | 67.2 | 68.1 | 63.7 | 74.3 | 71.3 | 65.5 | 69.7 | 61.3 | 67.2 |
| PL+DELTA | 62.4 | 63.0 | 63.2 | 73.4 | 61.3 | 71.9 | 73.5 | 67.2 | 68.3 | 64.0 | 75.0 | 71.5 | 65.6 | 70.1 | 62.2 | 67.5 |
| TENT | 61.7 | 63.3 | 63.9 | 73.0 | 62.3 | 71.4 | 73.1 | 67.6 | 68.1 | 65.1 | 74.9 | 71.4 | 65.5 | 70.7 | 62.5 | 67.6 |
| TENT+TBR | 61.6 | 63.8 | 64.4 | 73.3 | 62.2 | 71.5 | 73.6 | 68.0 | 68.0 | 64.9 | 74.8 | 71.4 | 65.5 | 71.0 | 63.0 | 67.8 |
| TENT+DOT | 62.4 | 63.6 | 64.7 | 73.1 | 62.6 | 71.6 | 73.7 | 68.0 | 68.6 | 65.3 | 74.7 | 71.8 | 66.1 | 70.7 | 63.0 | 68.0 |
| TENT+DELTA | 62.5 | 64.3 | 65.3 | 73.8 | 62.4 | 71.3 | 73.6 | 68.3 | 69.0 | 66.1 | 75.1 | 71.6 | 66.2 | 71.1 | 63.9 | 68.3 |
| Ent-W | 62.5 | 63.8 | 65.2 | 73.6 | 62.9 | 71.7 | 73.7 | 68.5 | 68.9 | 65.5 | 75.3 | 72.0 | 66.3 | 70.7 | 62.9 | 68.2 |
| Ent-W+TBR+Div-W(0.05) | 61.1 | 62.0 | 62.6 | 73.0 | 60.8 | 71.1 | 73.1 | 66.9 | 66.9 | 63.4 | 74.1 | 70.2 | 65.6 | 68.6 | 60.5 | 66.7 |
| Ent-W+TBR+Div-W(0.1) | 62.5 | 63.5 | 64.8 | 73.7 | 62.8 | 72.0 | 74.2 | 68.5 | 68.7 | 65.5 | 75.2 | 71.7 | 66.7 | 70.7 | 62.4 | 68.2 |
| Ent-W+TBR+Div-W(0.2) | 63.3 | 64.1 | 66.2 | 73.9 | 63.2 | 72.0 | 73.8 | 68.9 | 69.5 | 65.8 | 75.7 | 72.5 | 66.8 | 71.2 | 62.9 | 68.7 |
| Ent-W+TBR+Div-W(0.4) | 62.7 | 63.7 | 65.7 | 73.5 | 62.9 | 71.8 | 74.2 | 68.3 | 69.5 | 65.5 | 75.6 | 73.1 | 66.5 | 70.9 | 62.9 | 68.5 |
| Ent-W+TBR+LA | 63.6 | 64.6 | 66.4 | 74.2 | 63.7 | 72.1 | 74.2 | 69.0 | 70.1 | 66.0 | 76.0 | 73.3 | 67.2 | 71.8 | 63.4 | 69.0 |
| Ent-W+TBR+Sample-drop | 63.3 | 64.6 | 65.8 | 73.8 | 63.6 | 72.2 | 74.0 | 69.5 | 69.7 | 66.4 | 75.6 | 72.5 | 67.0 | 71.5 | 63.1 | 68.8 |
| Ent-W+DELTA | 63.9 | 64.8 | 66.4 | 74.1 | 63.7 | 72.2 | 74.4 | 69.2 | 70.5 | 66.2 | 75.6 | 73.3 | 67.0 | 71.6 | 63.3 | 69.1 |

Table 23: Classification accuracy (%) on IS+CI ($\pi = 0.05$) test stream of CIFAR100-C.

| Method | Gauss | Shot | Impul | Defoc | Glass | Motion | Zoom | Snow | Frost | Fog | Brit | Contr | Elastic | Pixel | JPEG | Avg |
|---|---|---|---|---|---|---|---|---|---|---|---|---|---|---|---|---|
| Source | 26.2 | 31.8 | 60.5 | 70.5 | 46.4 | 68.9 | 70.6 | 59.8 | 53.7 | 50.3 | 70.4 | 44.9 | 61.8 | 24.7 | 58.2 | 53.3 |
| BN adapt | 56.7 | 58.0 | 55.5 | 71.4 | 57.5 | 69.5 | 71.1 | 64.7 | 64.1 | 57.5 | 72.5 | 69.0 | 63.1 | 66.2 | 58.0 | 63.6 |
| ETA | 61.3 | 63.2 | 64.6 | 73.6 | 61.5 | 72.2 | 73.3 | 68.1 | 67.7 | 65.0 | 74.4 | 71.4 | 65.6 | 70.2 | 62.8 | 67.7 |
| LAME | 23.2 | 28.9 | 59.0 | 67.9 | 43.8 | 66.7 | 67.8 | 58.2 | 50.5 | 47.1 | 67.7 | 39.8 | 59.7 | 20.6 | 56.9 | 50.5 |
| CoTTA | 58.4 | 60.6 | 58.8 | 71.6 | 58.2 | 69.4 | 71.2 | 63.5 | 64.2 | 55.6 | 72.5 | 68.6 | 62.4 | 67.9 | 61.0 | 64.3 |
| CoTTA* | 58.4 | 60.9 | 59.1 | 72.0 | 59.9 | 70.9 | 71.8 | 65.2 | 66.5 | 58.6 | 73.9 | 71.0 | 65.7 | 70.5 | 65.2 | 66.0 |
| PL | 60.4 | 62.1 | 62.9 | 72.8 | 60.8 | 71.4 | 72.7 | 67.7 | 67.1 | 62.6 | 73.5 | 71.3 | 65.4 | 69.4 | 61.4 | 66.8 |
| PL+DELTA | 61.0 | 63.1 | 62.8 | 73.2 | 61.8 | 71.6 | 73.2 | 67.9 | 67.6 | 63.5 | 74.2 | 71.4 | 65.3 | 69.6 | 62.0 | 67.2 |
| TENT | 61.0 | 63.4 | 64.0 | 73.3 | 60.6 | 71.7 | 73.2 | 68.7 | 66.9 | 64.9 | 73.9 | 71.0 | 65.1 | 70.0 | 62.0 | 67.3 |
| TENT+DELTA | 61.7 | 64.8 | 65.6 | 73.5 | 62.1 | 71.2 | 73.4 | 69.0 | 68.6 | 65.4 | 74.6 | 71.1 | 66.1 | 70.6 | 63.1 | 68.1 |
| Ent-W | 61.4 | 63.2 | 64.7 | 73.7 | 61.5 | 72.1 | 73.2 | 68.4 | 67.8 | 64.9 | 74.4 | 71.3 | 65.6 | 70.1 | 62.7 | 67.7 |
| Ent-W+DELTA | 62.8 | 64.4 | 65.6 | 74.4 | 62.5 | 72.3 | 74.1 | 69.1 | 68.9 | 66.2 | 75.5 | 73.0 | 66.1 | 71.7 | 63.0 | 68.6 |

Table 24: Classification accuracy (%) on DS+CI ($\rho = 0.5$, $\pi = 0.1$) test stream of CIFAR100-C.

| Method | Gauss | Shot | Impul | Defoc | Glass | Motion | Zoom | Snow | Frost | Fog | Brit | Contr | Elastic | Pixel | JPEG | Avg |
|---|---|---|---|---|---|---|---|---|---|---|---|---|---|---|---|---|
| Source | 26.2 | 31.7 | 60.1 | 70.3 | 45.7 | 69.5 | 71.2 | 60.1 | 53.9 | 49.7 | 69.7 | 45.1 | 62.5 | 25.6 | 58.9 | 53.3 |
| BN adapt | 44.9 | 45.6 | 44.7 | 56.1 | 44.8 | 54.4 | 56.4 | 49.4 | 50.8 | 44.3 | 56.5 | 53.4 | 49.2 | 51.7 | 46.0 | 49.9 |
| ETA | 46.5 | 47.0 | 48.6 | 56.2 | 46.1 | 55.1 | 56.5 | 51.4 | 51.9 | 47.1 | 56.7 | 53.5 | 49.3 | 53.4 | 46.7 | 51.1 |
| LAME | 27.2 | 34.0 | 68.5 | 80.0 | 52.5 | 78.9 | 80.5 | 70.3 | 60.5 | 56.6 | 78.2 | 49.9 | 72.7 | 26.4 | 68.6 | 60.3 |
| CoTTA | 47.0 | 48.3 | 47.5 | 55.0 | 47.0 | 54.6 | 55.5 | 49.7 | 51.6 | 44.0 | 55.9 | 50.3 | 50.5 | 55.2 | 49.0 | 50.7 |
| CoTTA* | 47.0 | 48.3 | 47.6 | 54.7 | 48.0 | 54.1 | 54.7 | 49.7 | 51.7 | 45.1 | 55.2 | 50.1 | 50.6 | 54.7 | 50.6 | 50.8 |
| PL | 46.0 | 45.9 | 47.7 | 56.0 | 45.8 | 55.4 | 56.4 | 50.7 | 50.7 | 46.4 | 56.3 | 53.5 | 48.8 | 53.2 | 46.8 | 50.6 |
| PL+DELTA | 60.3 | 62.1 | 62.9 | 72.6 | 60.9 | 70.9 | 72.4 | 66.6 | 67.4 | 62.2 | 73.5 | 69.9 | 65.6 | 69.3 | 62.5 | 66.6 |
| TENT | 46.3 | 46.1 | 47.6 | 55.8 | 45.2 | 54.7 | 55.6 | 49.8 | 50.5 | 47.4 | 56.7 | 51.3 | 48.6 | 52.4 | 45.2 | 50.2 |
| TENT+DELTA | 62.5 | 63.7 | 64.9 | 73.5 | 62.2 | 70.8 | 72.1 | 67.6 | 68.0 | 65.7 | 75.0 | 70.5 | 66.6 | 69.9 | 63.4 | 67.8 |
| Ent-W | 46.7 | 46.9 | 48.7 | 56.1 | 46.1 | 55.0 | 56.3 | 51.2 | 51.9 | 47.7 | 57.1 | 53.5 | 49.2 | 53.2 | 46.6 | 51.1 |
| Ent-W+DELTA | 62.4 | 63.9 | 65.0 | 73.5 | 61.9 | 71.4 | 73.5 | 68.1 | 68.8 | 65.4 | 74.7 | 70.7 | 66.2 | 70.4 | 63.3 | 67.9 |

Table 25: Classification accuracy (%) on DS+CI ($\rho = 0.5$, $\pi = 0.05$) test stream of CIFAR100-C.

| Method | Gauss | Shot | Impul | Defoc | Glass | Motion | Zoom | Snow | Frost | Fog | Brit | Contr | Elastic | Pixel | JPEG | Avg |
|---|---|---|---|---|---|---|---|---|---|---|---|---|---|---|---|---|
| Source | 26.2 | 31.8 | 60.5 | 70.5 | 46.4 | 68.9 | 70.6 | 59.8 | 53.7 | 50.3 | 70.4 | 44.9 | 61.8 | 24.7 | 58.2 | 53.3 |
| BN adapt | 43.0 | 45.0 | 42.3 | 55.4 | 44.0 | 54.2 | 54.9 | 49.1 | 49.4 | 43.8 | 56.2 | 53.1 | 48.5 | 51.3 | 44.3 | 49.0 |
| ETA | 45.4 | 46.4 | 46.8 | 56.2 | 45.3 | 54.7 | 54.8 | 50.7 | 50.1 | 46.5 | 56.4 | 52.2 | 48.8 | 53.0 | 45.5 | 50.2 |
| LAME | 27.1 | 33.3 | 67.6 | 78.7 | 51.7 | 77.1 | 78.9 | 68.5 | 59.7 | 56.0 | 77.5 | 49.3 | 70.1 | 25.1 | 66.6 | 59.2 |
| CoTTA | 46.5 | 47.3 | 45.3 | 54.5 | 45.5 | 53.7 | 55.0 | 48.6 | 49.9 | 42.4 | 56.0 | 49.0 | 49.1 | 53.5 | 47.2 | 49.6 |
| CoTTA* | 46.5 | 47.8 | 45.5 | 54.1 | 46.2 | 53.4 | 54.2 | 48.8 | 50.6 | 43.5 | 54.2 | 49.4 | 49.6 | 52.8 | 48.7 | 49.7 |
| PL | 44.3 | 45.8 | 46.4 | 55.8 | 45.2 | 54.2 | 54.8 | 50.7 | 49.3 | 45.8 | 56.5 | 52.4 | 49.1 | 52.0 | 45.5 | 49.9 |
| PL+DELTA | 59.3 | 61.1 | 62.2 | 71.6 | 59.4 | 70.3 | 70.8 | 66.3 | 65.5 | 61.4 | 74.0 | 69.0 | 64.5 | 67.5 | 59.8 | 65.5 |
| TENT | 44.7 | 46.7 | 45.7 | 55.3 | 44.6 | 53.8 | 53.7 | 50.0 | 48.6 | 46.1 | 55.5 | 50.0 | 48.9 | 52.0 | 44.6 | 49.3 |
| TENT+TBR | 58.8 | 61.6 | 62.5 | 72.2 | 58.6 | 70.3 | 70.9 | 67.0 | 64.8 | 62.5 | 73.5 | 68.1 | 63.4 | 68.5 | 59.4 | 65.5 |
| TENT+DOT | 45.2 | 47.1 | 46.7 | 55.6 | 45.4 | 54.3 | 54.3 | 50.9 | 49.7 | 47.3 | 56.1 | 51.7 | 49.2 | 52.9 | 45.6 | 50.1 |
| TENT+DELTA | 60.3 | 62.3 | 63.7 | 72.9 | 60.3 | 70.3 | 71.3 | 67.8 | 66.2 | 64.1 | 74.2 | 68.7 | 64.3 | 69.1 | 60.7 | 66.4 |
| Ent-W | 45.6 | 46.4 | 47.0 | 56.0 | 45.4 | 54.9 | 54.9 | 50.7 | 50.1 | 46.8 | 56.3 | 52.2 | 48.6 | 53.1 | 45.1 | 50.2 |
| Ent-W+TBR+Div-W(0.05) | 60.9 | 62.3 | 62.9 | 73.0 | 59.5 | 70.7 | 72.0 | 67.0 | 66.2 | 62.4 | 74.5 | 69.8 | 64.9 | 69.0 | 60.7 | 66.4 |
| Ent-W+TBR+Div-W(0.1) | 61.2 | 62.8 | 64.5 | 73.5 | 59.9 | 71.3 | 71.8 | 67.4 | 66.4 | 63.7 | 74.5 | 70.6 | 65.2 | 69.5 | 61.0 | 66.9 |
| Ent-W+TBR+Div-W(0.2) | 60.4 | 62.4 | 63.6 | 73.5 | 59.4 | 70.7 | 72.0 | 67.1 | 65.8 | 63.4 | 74.4 | 70.1 | 64.5 | 69.5 | 60.4 | 66.5 |
| Ent-W+TBR+Div-W(0.4) | 59.7 | 62.3 | 63.3 | 72.9 | 59.4 | 70.6 | 71.9 | 67.0 | 65.8 | 63.1 | 74.3 | 69.6 | 63.9 | 69.4 | 60.3 | 66.2 |
| Ent-W+TBR+LA | 59.2 | 61.6 | 62.3 | 71.9 | 58.9 | 69.5 | 71.3 | 65.7 | 65.1 | 62.8 | 73.2 | 69.0 | 63.3 | 68.2 | 60.0 | 65.5 |
| Ent-W+TBR+Sample-drop | 60.9 | 62.6 | 63.7 | 73.2 | 60.0 | 70.6 | 72.0 | 66.9 | 66.6 | 64.1 | 74.9 | 69.4 | 64.6 | 69.9 | 61.1 | 66.7 |
| Ent-W+DELTA | 61.2 | 62.9 | 64.0 | 73.7 | 60.4 | 71.1 | 72.3 | 67.4 | 67.0 | 64.2 | 74.7 | 70.2 | 64.7 | 69.8 | 61.0 | 67.0 |

Table 26: Classification accuracy (%) on IS+CB test stream of ImageNet-C.

| Method | Gauss | Shot | Impul | Defoc | Glass | Motion | Zoom | Snow | Frost | Fog | Brit | Contr | Elastic | Pixel | JPEG | Avg |
|---|---|---|---|---|---|---|---|---|---|---|---|---|---|---|---|---|
| Source | 2.2 | 2.9 | 1.9 | 17.9 | 9.8 | 14.8 | 22.5 | 16.9 | 23.3 | 24.4 | 58.9 | 5.4 | 17.0 | 20.6 | 31.6 | 18.0 |
| TTA | 4.1 | 4.9 | 4.5 | 12.5 | 8.2 | 12.9 | 25.8 | 14.0 | 19.1 | 21.3 | 53.0 | 12.4 | 14.6 | 24.6 | 33.6 | 17.7 |
| BN adapt | 15.2 | 15.8 | 15.8 | 15.0 | 15.3 | 26.4 | 38.8 | 34.3 | 33.1 | 47.8 | 65.3 | 16.8 | 43.9 | 48.9 | 39.7 | 31.5 |
| MEMO | 7.5 | 8.7 | 9.0 | 19.7 | 13.0 | 20.7 | 27.6 | 25.3 | 28.8 | 32.1 | 61.0 | 11.0 | 23.8 | 33.0 | 37.5 | 23.9 |
| ETA | 35.6 | 37.5 | 36.2 | 33.7 | 33.1 | 47.7 | 52.5 | 51.9 | 45.8 | 60.0 | 67.8 | 44.7 | 57.8 | 60.9 | 55.2 | 48.0 |
| LAME | 1.6 | 2.4 | 1.3 | 17.6 | 9.1 | 13.9 | 21.9 | 15.6 | 22.5 | 22.8 | 58.6 | 5.2 | 15.2 | 19.9 | 31.1 | 17.2 |
| CoTTA | 17.6 | 18.0 | 17.4 | 15.6 | 18.2 | 31.2 | 43.6 | 36.6 | 35.1 | 53.0 | 66.5 | 19.5 | 46.3 | 54.9 | 42.6 | 34.4 |
| CoTTA* | 17.6 | 22.1 | 24.3 | 19.8 | 22.7 | 29.7 | 38.1 | 36.0 | 37.2 | 45.2 | 60.1 | 26.4 | 46.6 | 53.4 | 46.8 | 35.1 |
| PL | 26.2 | 26.2 | 27.0 | 25.2 | 24.3 | 37.2 | 46.5 | 43.3 | 39.5 | 55.0 | 66.7 | 30.2 | 51.2 | 55.7 | 49.1 | 40.2 |
| PL+DELTA | 27.7 | 29.4 | 28.5 | 27.0 | 26.1 | 38.1 | 47.9 | 44.1 | 40.7 | 55.9 | 67.4 | 34.1 | 52.9 | 56.6 | 50.3 | 41.8 |
| TENT | 28.7 | 30.5 | 30.1 | 28.0 | 27.2 | 41.4 | 49.4 | 47.2 | 41.2 | 57.4 | 67.4 | 26.5 | 54.6 | 58.5 | 52.5 | 42.7 |
| TENT+TBR | 29.5 | 31.4 | 30.9 | 28.8 | 28.0 | 41.9 | 50.3 | 47.7 | 41.8 | 58.3 | 68.1 | 26.9 | 55.4 | 59.3 | 53.3 | 43.5 |
| TENT+DOT | 30.5 | 32.3 | 31.6 | 29.6 | 29.3 | 42.5 | 49.9 | 47.8 | 42.2 | 57.5 | 67.5 | 37.5 | 55.4 | 58.8 | 52.9 | 44.4 |
| TENT+DELTA | 31.2 | 33.1 | 32.1 | 30.5 | 30.2 | 42.9 | 50.9 | 48.2 | 43.0 | 58.5 | 68.1 | 37.9 | 56.2 | 59.5 | 53.6 | 45.1 |
| Ent-W | 34.5 | 29.0 | 33.1 | 29.6 | 26.3 | 47.4 | 52.2 | 51.9 | 45.6 | 59.9 | 67.8 | 17.8 | 57.8 | 60.9 | 55.0 | 44.6 |
| Ent-W+TBR+Div-W(0.05) | 36.1 | 37.9 | 37.8 | 34.4 | 33.5 | 49.1 | 53.3 | 53.2 | 46.7 | 60.9 | 68.5 | 45.1 | 58.9 | 61.7 | 56.0 | 48.9 |
| Ent-W+TBR+Div-W(0.1) | 35.3 | 37.3 | 36.3 | 33.6 | 32.2 | 49.1 | 53.4 | 53.1 | 46.6 | 61.0 | 68.4 | 43.1 | 58.7 | 61.7 | 55.9 | 48.4 |
| Ent-W+TBR+Div-W(0.2) | 32.5 | 35.4 | 33.5 | 26.7 | 25.8 | 48.9 | 53.0 | 52.9 | 46.2 | 60.9 | 68.4 | 31.1 | 58.7 | 61.7 | 56.0 | 46.1 |
| Ent-W+TBR+Div-W(0.4) | 28.7 | 32.8 | 31.7 | 20.3 | 19.3 | 48.9 | 53.0 | 52.7 | 46.2 | 60.8 | 68.4 | 13.9 | 58.7 | 61.7 | 56.0 | 43.5 |
| Ent-W+TBR+LA | 26.7 | 22.4 | 29.6 | 20.3 | 20.0 | 49.2 | 53.4 | 52.9 | 46.7 | 60.7 | 68.0 | 10.1 | 58.8 | 61.5 | 56.0 | 42.4 |
| Ent-W+TBR+Sample-drop | 37.0 | 38.9 | 38.2 | 35.8 | 35.4 | 49.6 | 53.8 | 53.3 | 47.4 | 61.0 | 68.5 | 46.4 | 59.1 | 62.0 | 56.4 | 49.5 |
| Ent-W+DELTA | 38.1 | 39.6 | 39.0 | 36.3 | 36.5 | 49.9 | 54.0 | 53.5 | 47.6 | 61.1 | 68.4 | 46.9 | 59.2 | 61.9 | 56.6 | 49.9 |

Table 27: Classification accuracy (%) on IS+CB test stream of ImageNet-C with different architectures.

| Method | Gauss | Shot | Impul | Defoc | Glass | Motion | Zoom | Snow | Frost | Fog | Brit | Contr | Elastic | Pixel | JPEG | Avg |
|---|---|---|---|---|---|---|---|---|---|---|---|---|---|---|---|---|
| *ResNet18* | | | | | | | | | | | | | | | | |
| Source | 1.2 | 1.8 | 1.0 | 11.4 | 8.7 | 11.2 | 17.6 | 10.9 | 16.5 | 14.3 | 51.3 | 3.4 | 16.8 | 23.1 | 29.6 | 14.6 |
| TENT | 22.3 | 24.7 | 22.2 | 20.3 | 21.1 | 32.2 | 41.1 | 37.8 | 33.7 | 49.0 | 59.2 | 19.5 | 46.9 | 50.6 | 45.8 | 35.1 |
| TENT+DELTA | 24.5 | 26.8 | 24.4 | 22.6 | 23.7 | 34.0 | 42.7 | 38.9 | 35.4 | 50.2 | 60.3 | 27.5 | 48.5 | 51.9 | 47.0 | 37.2 |
| Ent-W | 27.1 | 30.7 | 24.3 | 22.3 | 17.5 | 37.6 | 44.2 | 42.5 | 37.8 | 51.5 | 59.9 | 5.5 | 49.5 | 52.9 | 48.5 | 36.8 |
| Ent-W+DELTA | 31.7 | 33.8 | 32.0 | 29.0 | 30.3 | 40.2 | 46.1 | 44.2 | 39.7 | 53.1 | 60.9 | 36.9 | 51.5 | 54.7 | 49.8 | 42.3 |
| *ResNet50* | | | | | | | | | | | | | | | | |
| Source | 2.2 | 2.9 | 1.9 | 17.9 | 9.8 | 14.8 | 22.5 | 16.9 | 23.3 | 24.4 | 58.9 | 5.4 | 17.0 | 20.6 | 31.6 | 18.0 |
| TENT | 28.7 | 30.5 | 30.1 | 28.0 | 27.2 | 41.4 | 49.4 | 47.2 | 41.2 | 57.3 | 67.4 | 26.7 | 54.6 | 58.5 | 52.5 | 42.7 |
| TENT+DELTA | 31.2 | 33.1 | 32.1 | 30.5 | 30.2 | 42.9 | 50.9 | 48.2 | 43.0 | 58.5 | 68.1 | 37.9 | 56.2 | 59.5 | 53.6 | 45.1 |
| Ent-W | 34.5 | 29.0 | 33.1 | 29.6 | 26.3 | 47.4 | 52.2 | 51.9 | 45.6 | 59.9 | 67.8 | 17.8 | 57.8 | 60.9 | 55.0 | 44.6 |
| Ent-W+DELTA | 38.1 | 39.6 | 39.0 | 36.3 | 36.5 | 49.9 | 54.0 | 53.5 | 47.6 | 61.1 | 68.4 | 46.9 | 59.2 | 61.9 | 56.6 | 49.9 |
| *ResNet101* | | | | | | | | | | | | | | | | |
| Source | 3.5 | 4.3 | 3.5 | 21.9 | 13.1 | 19.2 | 26.5 | 21.0 | 26.7 | 28.1 | 61.4 | 7.2 | 24.3 | 35.0 | 42.3 | 22.5 |
| TENT | 32.6 | 34.0 | 33.2 | 32.2 | 32.4 | 45.1 | 53.0 | 50.8 | 45.0 | 59.6 | 69.1 | 33.8 | 58.6 | 61.1 | 55.8 | 46.4 |
| TENT+DELTA | 35.1 | 37.4 | 35.6 | 34.9 | 35.1 | 46.8 | 54.6 | 51.8 | 46.7 | 60.7 | 69.9 | 42.6 | 60.1 | 62.3 | 57.2 | 48.7 |
| Ent-W | 36.1 | 20.8 | 37.3 | 33.6 | 31.7 | 50.3 | 55.6 | 54.9 | 46.8 | 62.4 | 69.8 | 19.7 | 61.1 | 63.2 | 58.2 | 46.8 |
| Ent-W+DELTA | 40.9 | 43.0 | 41.9 | 39.8 | 40.1 | 53.1 | 57.4 | 56.5 | 50.8 | 63.4 | 70.2 | 50.6 | 62.3 | 64.2 | 59.8 | 53.0 |
| *ResNet152* | | | | | | | | | | | | | | | | |
| Source | 3.6 | 4.4 | 3.3 | 22.1 | 11.9 | 24.8 | 25.5 | 22.1 | 28.9 | 27.7 | 63.1 | 5.2 | 24.9 | 27.1 | 42.2 | 22.5 |
| TENT | 34.0 | 36.8 | 35.3 | 34.1 | 34.0 | 46.9 | 54.0 | 52.4 | 47.0 | 61.3 | 70.7 | 35.5 | 59.9 | 62.4 | 57.2 | 48.1 |
| TENT+DELTA | 36.6 | 39.2 | 37.7 | 36.7 | 36.3 | 48.7 | 55.6 | 54.0 | 48.4 | 62.4 | 71.2 | 44.0 | 61.3 | 63.3 | 58.4 | 50.2 |
| Ent-W | 38.7 | 33.4 | 34.6 | 36.6 | 33.2 | 52.9 | 57.4 | 56.9 | 46.5 | 64.2 | 71.0 | 29.3 | 62.7 | 64.8 | 60.0 | 49.5 |
| Ent-W+DELTA | 42.6 | 45.4 | 44.5 | 42.0 | 42.2 | 55.5 | 58.9 | 58.5 | 52.7 | 65.5 | 71.4 | 51.9 | 63.7 | 65.8 | 61.2 | 54.8 |
| *WideResNet50* | | | | | | | | | | | | | | | | |
| TENT | 34.5 | 37.2 | 34.7 | 30.6 | 31.6 | 45.2 | 52.0 | 51.1 | 45.8 | 60.5 | 69.9 | 38.4 | 58.3 | 61.7 | 54.9 | 47.1 |
| TENT+DELTA | 36.7 | 39.6 | 37.2 | 33.5 | 34.6 | 47.4 | 54.5 | 53.0 | 47.6 | 62.2 | 71.2 | 44.1 | 60.3 | 63.4 | 56.9 | 49.5 |
| Ent-W | 34.0 | 37.1 | 33.6 | 25.0 | 27.7 | 51.0 | 54.7 | 55.5 | 49.9 | 62.8 | 70.4 | 24.9 | 60.7 | 63.9 | 57.6 | 47.3 |
| Ent-W+DELTA | 41.1 | 44.9 | 42.9 | 38.6 | 39.3 | 53.4 | 57.3 | 57.6 | 51.8 | 64.7 | 71.4 | 52.0 | 62.4 | 65.7 | 59.8 | 53.5 |
| *ResNeXt50* | | | | | | | | | | | | | | | | |
| TENT | 33.3 | 36.2 | 34.2 | 32.3 | 30.9 | 45.5 | 52.2 | 51.1 | 45.9 | 59.6 | 69.3 | 39.0 | 57.1 | 61.5 | 53.8 | 46.8 |
| TENT+DELTA | 35.3 | 38.5 | 36.1 | 34.5 | 33.5 | 46.6 | 53.7 | 52.1 | 47.0 | 60.5 | 69.9 | 43.9 | 58.4 | 62.4 | 55.0 | 48.5 |
| Ent-W | 31.4 | 37.5 | 34.7 | 34.0 | 25.2 | 51.0 | 54.6 | 55.1 | 49.1 | 62.2 | 70.0 | 49.1 | 60.3 | 64.3 | 57.1 | 49.0 |
| Ent-W+DELTA | 40.7 | 43.6 | 42.0 | 39.5 | 39.1 | 53.1 | 56.7 | 56.6 | 51.1 | 63.2 | 70.4 | 50.7 | 61.5 | 64.9 | 58.2 | 52.8 |

Table 28: Classification accuracy (%) on DS+CB ($\rho = 1.0$) test stream of ImageNet-C.

| Method | Gauss | Shot | Impul | Defoc | Glass | Motion | Zoom | Snow | Frost | Fog | Brit | Contr | Elastic | Pixel | JPEG | Avg |
|---|---|---|---|---|---|---|---|---|---|---|---|---|---|---|---|---|
| Source | 2.2 | 2.9 | 1.9 | 17.9 | 9.8 | 14.8 | 22.5 | 16.9 | 23.3 | 24.4 | 58.9 | 5.4 | 17.0 | 20.6 | 31.6 | 18.0 |
| BN adapt | 10.6 | 10.9 | 10.9 | 10.2 | 10.3 | 17.5 | 25.8 | 23.5 | 23.1 | 33.0 | 46.5 | 11.3 | 30.2 | 33.3 | 27.0 | 21.6 |
| ETA | 17.0 | 19.2 | 18.2 | 14.1 | 12.0 | 25.9 | 31.1 | 30.9 | 26.8 | 38.6 | 46.1 | 18.9 | 36.0 | 38.7 | 33.5 | 27.1 |
| LAME | 1.8 | 2.7 | 1.5 | 22.4 | 11.3 | 17.2 | 28.4 | 19.8 | 28.4 | 29.8 | 74.4 | 5.9 | 20.0 | 25.6 | 40.4 | 22.0 |
| CoTTA | 12.2 | 12.5 | 12.8 | 9.5 | 11.2 | 19.7 | 28.4 | 24.7 | 23.9 | 35.9 | 47.4 | 12.8 | 31.1 | 37.0 | 28.4 | 23.2 |
| CoTTA* | 12.2 | 14.9 | 16.2 | 12.3 | 14.2 | 18.9 | 24.2 | 24.4 | 25.2 | 30.0 | 41.5 | 15.3 | 30.8 | 35.5 | 31.2 | 23.1 |
| PL | 15.9 | 15.6 | 16.4 | 14.4 | 13.9 | 23.1 | 29.8 | 28.1 | 26.2 | 37.3 | 47.2 | 12.8 | 34.2 | 37.2 | 32.2 | 25.6 |
| PL+DELTA | 26.3 | 27.4 | 27.1 | 25.5 | 25.1 | 37.4 | 46.5 | 43.0 | 39.8 | 54.8 | 66.6 | 32.7 | 51.4 | 55.6 | 48.6 | 40.5 |
| TENT | 16.1 | 16.8 | 16.8 | 15.1 | 14.1 | 23.3 | 30.2 | 28.8 | 24.9 | 37.5 | 46.7 | 9.3 | 34.9 | 37.8 | 33.0 | 25.7 |
| TENT+DELTA | 29.6 | 31.7 | 30.4 | 29.1 | 28.6 | 41.5 | 49.8 | 47.0 | 42.1 | 57.6 | 67.5 | 35.7 | 54.9 | 58.5 | 52.0 | 43.7 |
| Ent-W | 4.2 | 2.8 | 3.1 | 2.9 | 3.6 | 11.3 | 20.2 | 20.0 | 12.5 | 34.4 | 44.7 | 1.7 | 32.0 | 37.1 | 21.5 | 16.8 |
| Ent-W+DELTA | 35.6 | 37.9 | 36.0 | 34.4 | 34.4 | 47.9 | 52.8 | 51.9 | 46.5 | 60.1 | 67.8 | 44.2 | 57.9 | 60.8 | 55.4 | 48.3 |

Table 29: Classification accuracy (%) on DS+CB ($\rho = 0.5$) test stream of ImageNet-C.

| Method | Gauss | Shot | Impul | Defoc | Glass | Motion | Zoom | Snow | Frost | Fog | Brit | Contr | Elastic | Pixel | JPEG | Avg |
|---|---|---|---|---|---|---|---|---|---|---|---|---|---|---|---|---|
| Source | 2.2 | 2.9 | 1.9 | 17.9 | 9.8 | 14.8 | 22.5 | 16.9 | 23.3 | 24.4 | 58.9 | 5.4 | 17.0 | 20.6 | 31.6 | 18.0 |
| BN adapt | 9.6 | 9.9 | 9.8 | 8.8 | 9.1 | 15.8 | 22.8 | 21.0 | 20.8 | 29.5 | 41.9 | 10.3 | 26.7 | 29.5 | 24.2 | 19.3 |
| ETA | 13.9 | 15.5 | 13.3 | 11.1 | 10.3 | 21.4 | 26.2 | 26.1 | 22.9 | 33.4 | 40.5 | 13.4 | 30.9 | 33.2 | 29.3 | 22.8 |
| LAME | 1.9 | 2.8 | 1.6 | 23.6 | 11.7 | 17.8 | 29.4 | 20.4 | 29.4 | 30.5 | 76.1 | 6.2 | 20.8 | 26.4 | 41.5 | 22.7 |
| CoTTA | 10.8 | 11.0 | 11.0 | 7.8 | 10.4 | 17.4 | 25.0 | 22.0 | 21.6 | 32.0 | 42.6 | 9.9 | 27.9 | 32.5 | 25.8 | 20.5 |
| CoTTA* | 10.8 | 13.3 | 14.3 | 11.0 | 12.9 | 17.1 | 22.0 | 21.8 | 22.8 | 27.6 | 38.0 | 14.8 | 27.8 | 32.6 | 28.0 | 21.0 |
| PL | 14.2 | 13.4 | 14.5 | 12.5 | 11.5 | 20.0 | 26.3 | 24.9 | 23.4 | 33.2 | 42.4 | 11.1 | 30.5 | 33.1 | 28.5 | 22.6 |
| PL+DELTA | 25.6 | 27.3 | 26.2 | 25.2 | 24.6 | 36.0 | 45.7 | 42.9 | 39.3 | 54.4 | 66.5 | 31.0 | 50.6 | 55.0 | 47.9 | 39.9 |
| TENT | 13.9 | 14.6 | 14.5 | 12.6 | 11.7 | 19.0 | 26.1 | 25.2 | 21.5 | 33.2 | 41.6 | 6.5 | 30.5 | 33.1 | 28.9 | 22.2 |
| TENT+TBR | 27.3 | 28.5 | 28.2 | 26.0 | 25.4 | 38.9 | 48.5 | 46.0 | 39.6 | 57.1 | 67.3 | 18.5 | 53.6 | 57.6 | 51.2 | 40.9 |
| TENT+DOT | 15.5 | 16.5 | 15.9 | 14.2 | 14.0 | 20.9 | 27.1 | 25.9 | 23.5 | 33.7 | 41.8 | 15.2 | 31.4 | 33.5 | 29.5 | 23.9 |
| TENT+DELTA | 29.1 | 30.9 | 29.7 | 28.2 | 27.8 | 40.3 | 49.0 | 46.7 | 41.5 | 57.3 | 67.3 | 33.9 | 54.4 | 58.1 | 51.6 | 43.1 |
| Ent-W | 2.9 | 2.5 | 3.5 | 1.4 | 1.0 | 7.1 | 11.9 | 15.1 | 8.5 | 27.7 | 37.0 | 1.3 | 22.2 | 31.1 | 20.1 | 12.9 |
| Ent-W+TBR+Div-W(0.05) | 32.4 | 34.6 | 33.3 | 27.2 | 28.3 | 45.2 | 51.3 | 50.5 | 44.3 | 59.3 | 67.4 | 36.0 | 57.0 | 60.1 | 54.3 | 45.4 |
| Ent-W+TBR+Div-W(0.1) | 30.1 | 33.4 | 31.1 | 25.5 | 21.7 | 44.8 | 51.1 | 50.4 | 43.4 | 59.3 | 67.4 | 16.3 | 56.9 | 60.2 | 54.3 | 43.1 |
| Ent-W+TBR+Div-W(0.2) | 23.7 | 30.5 | 26.5 | 19.7 | 12.2 | 44.3 | 51.1 | 50.5 | 41.1 | 59.4 | 67.4 | 7.0 | 56.8 | 60.2 | 54.2 | 40.3 |
| Ent-W+TBR+Div-W(0.4) | 17.1 | 15.3 | 22.2 | 11.2 | 4.8 | 43.7 | 51.1 | 50.2 | 36.5 | 59.5 | 67.4 | 6.1 | 56.7 | 60.3 | 54.2 | 37.1 |
| Ent-W+TBR+LA | 10.9 | 7.2 | 14.3 | 5.1 | 5.0 | 35.0 | 39.9 | 39.3 | 23.2 | 46.8 | 53.6 | 4.1 | 44.6 | 47.2 | 42.4 | 27.9 |
| Ent-W+TBR+Sample-drop | 33.7 | 36.4 | 35.1 | 31.9 | 30.8 | 46.7 | 52.2 | 51.2 | 45.6 | 60.0 | 67.6 | 40.4 | 57.3 | 60.6 | 54.7 | 46.9 |
| Ent-W+DELTA | 34.9 | 37.5 | 35.8 | 32.7 | 32.3 | 46.7 | 52.3 | 51.5 | 46.0 | 59.7 | 67.3 | 42.8 | 57.3 | 60.4 | 54.9 | 47.5 |

Table 30: Classification accuracy (%) on DS+CB ($\rho = 0.1$) test stream of ImageNet-C.

| Method | Gauss | Shot | Impul | Defoc | Glass | Motion | Zoom | Snow | Frost | Fog | Brit | Contr | Elastic | Pixel | JPEG | Avg |
|---|---|---|---|---|---|---|---|---|---|---|---|---|---|---|---|---|
| Source | 2.2 | 2.9 | 1.9 | 17.9 | 9.8 | 14.8 | 22.5 | 16.9 | 23.3 | 24.4 | 58.9 | 5.4 | 17.0 | 20.6 | 31.6 | 18.0 |
| BN adapt | 6.3 | 6.4 | 6.2 | 5.6 | 5.6 | 9.8 | 13.9 | 13.6 | 13.4 | 18.4 | 26.1 | 6.4 | 16.5 | 18.1 | 15.0 | 12.1 |
| ETA | 4.6 | 5.2 | 5.1 | 2.3 | 2.8 | 7.0 | 12.3 | 11.6 | 10.9 | 17.5 | 22.5 | 2.3 | 14.5 | 17.1 | 14.4 | 10.0 |
| LAME | 1.9 | 2.9 | 1.6 | 26.2 | 12.8 | 19.8 | 32.7 | 22.8 | 32.5 | 33.8 | 80.0 | 6.6 | 22.5 | 29.0 | 45.5 | 24.7 |
| CoTTA | 7.1 | 7.0 | 7.1 | 5.0 | 6.2 | 10.5 | 15.0 | 14.1 | 13.7 | 19.3 | 26.5 | 6.4 | 17.1 | 19.6 | 15.8 | 12.7 |
| CoTTA* | 7.1 | 8.2 | 8.7 | 6.3 | 7.3 | 10.3 | 13.4 | 14.3 | 14.5 | 17.9 | 23.7 | 7.9 | 16.9 | 19.4 | 17.3 | 12.9 |
| PL | 7.7 | 7.6 | 8.3 | 6.4 | 6.1 | 10.8 | 15.4 | 15.0 | 14.0 | 20.0 | 25.9 | 5.0 | 17.5 | 19.4 | 17.0 | 13.1 |
| PL+DELTA | 23.4 | 24.6 | 24.0 | 22.0 | 21.5 | 33.3 | 43.4 | 40.0 | 37.3 | 52.2 | 65.0 | 26.1 | 47.8 | 52.5 | 45.4 | 37.2 |
| TENT | 7.4 | 7.8 | 7.8 | 6.2 | 5.9 | 8.9 | 14.7 | 12.5 | 11.6 | 19.0 | 24.5 | 3.0 | 16.8 | 18.5 | 16.5 | 12.1 |
| TENT+DELTA | 26.7 | 28.2 | 27.3 | 25.0 | 24.8 | 37.1 | 46.6 | 43.6 | 39.6 | 55.1 | 65.7 | 27.2 | 51.6 | 55.6 | 49.0 | 40.2 |
| Ent-W | 1.5 | 0.6 | 1.4 | 1.1 | 0.8 | 2.3 | 4.6 | 4.4 | 3.1 | 8.4 | 15.5 | 0.5 | 7.0 | 9.7 | 5.7 | 4.4 |
| Ent-W+DELTA | 30.4 | 33.1 | 31.4 | 26.8 | 28.1 | 42.2 | 48.9 | 48.2 | 42.6 | 56.9 | 65.4 | 31.5 | 54.4 | 57.8 | 51.5 | 43.3 |

Table 31: Classification accuracy (%) on IS+CI ($\pi = 0.1$) test stream of ImageNet-C.

| Method | Gauss | Shot | Impul | Defoc | Glass | Motion | Zoom | Snow | Frost | Fog | Brit | Contr | Elastic | Pixel | JPEG | Avg |
|---|---|---|---|---|---|---|---|---|---|---|---|---|---|---|---|---|
| Source | 2.4 | 3.0 | 1.9 | 17.8 | 9.7 | 14.7 | 22.4 | 16.5 | 23.1 | 24.2 | 58.9 | 5.5 | 16.9 | 20.4 | 31.5 | 17.9 |
| BN adapt | 15.0 | 15.8 | 15.4 | 14.7 | 15.1 | 25.6 | 39.1 | 34.4 | 33.2 | 47.8 | 65.1 | 17.5 | 44.4 | 48.8 | 39.8 | 31.5 |
| ETA | 34.6 | 36.7 | 35.7 | 33.1 | 32.5 | 46.5 | 51.9 | 51.3 | 45.4 | 59.5 | 67.6 | 44.8 | 57.3 | 60.9 | 55.1 | 47.5 |
| LAME | 1.8 | 2.5 | 1.5 | 17.5 | 9.0 | 13.9 | 21.8 | 15.1 | 22.3 | 22.6 | 58.5 | 5.3 | 14.9 | 19.8 | 30.9 | 17.2 |
| CoTTA | 17.1 | 17.8 | 17.5 | 15.9 | 16.7 | 30.2 | 43.2 | 36.8 | 35.7 | 51.9 | 66.4 | 17.7 | 47.1 | 54.0 | 42.8 | 34.1 |
| CoTTA* | 17.1 | 22.0 | 24.1 | 19.0 | 22.2 | 28.0 | 35.7 | 35.2 | 35.8 | 42.8 | 57.8 | 22.9 | 44.8 | 50.4 | 45.3 | 33.5 |
| PL | 24.9 | 24.8 | 25.9 | 24.3 | 23.4 | 36.2 | 45.7 | 42.3 | 39.5 | 54.6 | 66.5 | 28.6 | 49.9 | 55.5 | 48.5 | 39.4 |
| PL+DELTA | 26.4 | 27.7 | 27.0 | 26.3 | 24.9 | 37.5 | 46.9 | 43.3 | 40.2 | 55.3 | 66.8 | 33.3 | 52.1 | 56.5 | 49.8 | 40.9 |
| TENT | 27.8 | 29.3 | 29.2 | 28.1 | 26.6 | 40.8 | 48.7 | 46.5 | 41.0 | 57.2 | 67.3 | 25.7 | 53.6 | 58.2 | 51.9 | 42.1 |
| TENT+TBR | 28.5 | 30.1 | 29.7 | 28.7 | 27.3 | 41.3 | 49.9 | 47.0 | 41.7 | 57.6 | 67.9 | 25.1 | 54.5 | 59.0 | 52.9 | 42.7 |
| TENT+DOT | 29.8 | 31.6 | 30.9 | 29.4 | 28.8 | 41.7 | 49.4 | 47.0 | 42.1 | 57.3 | 67.3 | 36.8 | 54.9 | 58.6 | 52.4 | 43.9 |
| TENT+DELTA | 30.7 | 32.5 | 31.3 | 30.3 | 29.3 | 42.0 | 50.5 | 47.5 | 42.9 | 57.8 | 67.7 | 36.4 | 55.7 | 59.2 | 53.1 | 44.4 |
| Ent-W | 23.2 | 21.7 | 29.4 | 19.1 | 19.6 | 46.7 | 51.7 | 51.0 | 39.0 | 58.9 | 67.5 | 10.1 | 57.2 | 60.5 | 54.9 | 40.7 |
| Ent-W+TBR+Div-W(0.05) | 34.1 | 37.4 | 36.4 | 32.5 | 32.9 | 47.7 | 52.9 | 52.1 | 45.7 | 60.0 | 67.9 | 42.6 | 57.8 | 61.7 | 55.7 | 47.8 |
| Ent-W+TBR+Div-W(0.1) | 34.5 | 36.1 | 35.9 | 32.4 | 32.0 | 48.0 | 52.9 | 52.1 | 45.8 | 59.8 | 68.0 | 40.2 | 57.9 | 61.5 | 55.7 | 47.5 |
| Ent-W+TBR+Div-W(0.2) | 32.5 | 34.1 | 35.3 | 30.0 | 29.7 | 47.6 | 52.7 | 51.9 | 45.5 | 59.7 | 68.0 | 30.2 | 57.9 | 61.5 | 55.7 | 46.1 |
| Ent-W+TBR+Div-W(0.4) | 29.2 | 27.5 | 34.3 | 27.4 | 25.1 | 47.8 | 52.8 | 51.8 | 44.7 | 59.5 | 68.0 | 6.1 | 58.0 | 61.4 | 55.8 | 43.3 |
| Ent-W+TBR+LA | 24.8 | 23.5 | 34.6 | 25.1 | 20.4 | 48.2 | 52.9 | 52.2 | 45.0 | 59.7 | 67.3 | 4.2 | 58.0 | 61.3 | 55.8 | 42.2 |
| Ent-W+TBR+Sample-drop | 36.1 | 37.8 | 37.3 | 33.7 | 33.2 | 47.3 | 52.9 | 52.1 | 46.0 | 59.7 | 68.0 | 43.7 | 57.9 | 61.5 | 55.5 | 48.2 |
| Ent-W+DELTA | 36.6 | 38.6 | 37.8 | 34.9 | 34.4 | 47.7 | 52.6 | 51.9 | 46.1 | 59.5 | 67.4 | 44.6 | 57.9 | 60.9 | 55.4 | 48.4 |

Table 32: Classification accuracy (%) on IS+CI ($\pi = 0.05$) test stream of ImageNet-C.

| Method | Gauss | Shot | Impul | Defoc | Glass | Motion | Zoom | Snow | Frost | Fog | Brit | Contr | Elastic | Pixel | JPEG | Avg |
|---|---|---|---|---|---|---|---|---|---|---|---|---|---|---|---|---|
| Source | 2.2 | 2.9 | 1.9 | 18.0 | 10.0 | 14.6 | 22.5 | 16.6 | 23.1 | 24.4 | 58.4 | 5.5 | 16.9 | 20.6 | 31.5 | 17.9 |
| BN adapt | 15.1 | 15.2 | 15.6 | 14.9 | 15.8 | 25.6 | 39.0 | 34.6 | 33.2 | 47.8 | 64.7 | 17.2 | 44.1 | 48.2 | 39.9 | 31.4 |
| ETA | 34.2 | 36.1 | 35.0 | 32.0 | 32.0 | 46.1 | 52.0 | 50.6 | 45.0 | 59.4 | 67.3 | 43.4 | 57.0 | 60.3 | 54.5 | 47.0 |
| LAME | 1.6 | 2.4 | 1.4 | 17.7 | 9.1 | 13.9 | 21.9 | 15.4 | 22.3 | 22.7 | 58.0 | 5.2 | 15.1 | 19.8 | 30.9 | 17.2 |
| CoTTA | 17.3 | 17.4 | 17.8 | 15.4 | 17.1 | 29.8 | 43.1 | 37.5 | 35.4 | 51.9 | 65.8 | 19.3 | 46.8 | 53.3 | 42.5 | 34.0 |
| CoTTA* | 17.3 | 21.6 | 23.8 | 19.9 | 22.9 | 29.3 | 37.4 | 35.7 | 36.6 | 44.5 | 59.0 | 24.2 | 45.5 | 51.8 | 45.9 | 34.4 |
| PL | 24.2 | 24.6 | 25.8 | 24.7 | 23.5 | 36.2 | 45.8 | 42.7 | 38.9 | 54.3 | 65.9 | 27.0 | 49.0 | 55.0 | 48.0 | 39.0 |
| PL+DELTA | 26.1 | 27.3 | 27.1 | 25.8 | 25.3 | 36.2 | 46.8 | 43.2 | 39.9 | 54.8 | 66.4 | 32.6 | 51.1 | 55.4 | 48.8 | 40.5 |
| TENT | 27.1 | 29.0 | 28.8 | 27.7 | 27.1 | 40.3 | 49.1 | 46.4 | 40.7 | 57.1 | 66.6 | 24.8 | 53.1 | 57.8 | 51.3 | 41.8 |
| TENT+DELTA | 30.1 | 32.3 | 31.2 | 29.6 | 29.6 | 41.4 | 50.0 | 47.4 | 42.4 | 57.6 | 67.2 | 35.3 | 55.1 | 58.5 | 52.6 | 44.0 |
| Ent-W | 17.2 | 13.4 | 25.6 | 15.8 | 12.1 | 45.9 | 51.0 | 50.4 | 44.6 | 59.3 | 66.9 | 10.0 | 56.5 | 60.0 | 54.1 | 38.9 |
| Ent-W+DELTA | 35.7 | 38.2 | 37.1 | 34.1 | 33.8 | 46.5 | 51.7 | 51.1 | 45.6 | 58.4 | 66.0 | 43.5 | 57.0 | 59.3 | 54.5 | 47.5 |

Table 33: Classification accuracy (%) on DS+CI ($\rho = 0.5, \pi = 0.1$) test stream of ImageNet-C.

| Method | Gauss | Shot | Impul | Defoc | Glass | Motion | Zoom | Snow | Frost | Fog | Brit | Contr | Elastic | Pixel | JPEG | Avg |
|---|---|---|---|---|---|---|---|---|---|---|---|---|---|---|---|---|
| Source | 2.4 | 3.0 | 1.9 | 17.8 | 9.7 | 14.7 | 22.4 | 16.5 | 23.1 | 24.2 | 58.9 | 5.5 | 16.9 | 20.4 | 31.5 | 17.9 |
| BN adapt | 9.7 | 10.1 | 10.1 | 9.3 | 9.5 | 16.1 | 24.2 | 21.7 | 21.4 | 30.8 | 43.5 | 11.2 | 27.0 | 30.9 | 25.7 | 20.1 |
| ETA | 11.9 | 12.7 | 12.7 | 8.6 | 7.3 | 18.8 | 27.1 | 25.9 | 22.8 | 34.0 | 41.9 | 7.9 | 30.7 | 34.9 | 30.3 | 21.8 |
| LAME | 2.0 | 2.8 | 1.5 | 22.4 | 11.4 | 17.1 | 27.5 | 19.4 | 28.0 | 29.5 | 73.1 | 6.0 | 19.8 | 24.9 | 40.0 | 21.7 |
| CoTTA | 11.4 | 11.6 | 11.7 | 9.8 | 10.4 | 17.9 | 26.4 | 22.9 | 22.6 | 33.3 | 44.4 | 11.6 | 28.6 | 33.8 | 27.3 | 21.6 |
| CoTTA* | 11.4 | 13.9 | 14.9 | 11.7 | 13.3 | 17.9 | 22.8 | 22.7 | 23.5 | 29.2 | 39.4 | 14.6 | 28.2 | 33.1 | 29.6 | 21.7 |
| PL | 14.4 | 12.5 | 14.0 | 12.6 | 11.8 | 20.2 | 27.2 | 25.3 | 24.1 | 34.1 | 43.9 | 10.7 | 29.8 | 34.2 | 29.8 | 23.0 |
| PL+DELTA | 24.8 | 25.5 | 25.4 | 23.6 | 23.0 | 34.9 | 44.8 | 41.0 | 38.8 | 53.2 | 65.9 | 29.6 | 49.7 | 54.1 | 47.3 | 38.8 |
| TENT | 12.9 | 13.9 | 14.3 | 12.8 | 11.7 | 18.5 | 27.0 | 25.0 | 21.7 | 34.1 | 42.9 | 6.6 | 30.1 | 34.5 | 30.1 | 22.4 |
| TENT+DELTA | 28.3 | 30.1 | 29.1 | 27.5 | 27.2 | 39.3 | 48.3 | 45.2 | 41.2 | 56.3 | 66.8 | 31.0 | 53.6 | 57.2 | 51.2 | 42.2 |
| Ent-W | 1.6 | 1.6 | 2.4 | 2.3 | 1.3 | 5.6 | 12.9 | 13.5 | 11.1 | 16.7 | 40.4 | 1.1 | 16.8 | 17.4 | 16.6 | 10.8 |
| Ent-W+DELTA | 32.2 | 35.0 | 34.1 | 30.5 | 29.4 | 44.8 | 50.7 | 49.5 | 44.5 | 58.1 | 66.6 | 36.6 | 55.7 | 58.4 | 53.7 | 45.3 |

Table 34: Classification accuracy (%) on DS+CI ($\rho = 0.5, \pi = 0.05$) test stream of ImageNet-C.

| Method | Gauss | Shot | Impul | Defoc | Glass | Motion | Zoom | Snow | Frost | Fog | Brit | Contr | Elastic | Pixel | JPEG | Avg |
|---|---|---|---|---|---|---|---|---|---|---|---|---|---|---|---|---|
| Source | 2.2 | 2.9 | 1.9 | 18.0 | 10.0 | 14.6 | 22.5 | 16.6 | 23.1 | 24.4 | 58.4 | 5.5 | 16.9 | 20.6 | 31.5 | 17.9 |
| BN adapt | 9.8 | 9.9 | 10.5 | 9.6 | 9.5 | 15.8 | 23.8 | 22.1 | 21.8 | 31.4 | 43.9 | 11.2 | 26.9 | 30.6 | 25.8 | 20.2 |
| ETA | 9.1 | 11.2 | 12.5 | 3.9 | 7.8 | 18.2 | 26.3 | 25.6 | 22.4 | 34.4 | 41.9 | 6.4 | 30.7 | 33.7 | 29.7 | 20.9 |
| LAME | 1.8 | 2.7 | 1.6 | 21.9 | 11.5 | 16.9 | 27.3 | 19.3 | 28.3 | 29.3 | 71.4 | 5.8 | 20.2 | 24.6 | 39.1 | 21.5 |
| CoTTA | 11.3 | 11.4 | 12.1 | 8.9 | 10.0 | 17.9 | 26.3 | 23.4 | 23.0 | 33.9 | 44.6 | 10.1 | 29.0 | 34.1 | 27.6 | 21.6 |
| CoTTA* | 11.3 | 13.6 | 14.9 | 12.1 | 13.4 | 17.8 | 23.1 | 23.3 | 23.5 | 29.1 | 40.0 | 13.9 | 28.4 | 33.1 | 29.8 | 21.8 |
| PL | 13.3 | 11.2 | 14.7 | 12.4 | 12.4 | 19.6 | 27.1 | 25.7 | 24.2 | 34.7 | 44.4 | 8.4 | 29.7 | 34.0 | 30.0 | 22.8 |
| PL+DELTA | 23.8 | 25.0 | 25.1 | 23.4 | 22.6 | 33.6 | 44.3 | 41.2 | 38.7 | 53.1 | 65.4 | 27.9 | 48.9 | 53.6 | 46.6 | 38.2 |
| TENT | 12.6 | 13.6 | 14.3 | 12.6 | 11.4 | 17.2 | 26.6 | 25.2 | 21.7 | 34.6 | 43.0 | 6.0 | 29.6 | 34.2 | 30.3 | 22.2 |
| TENT+TBR | 24.7 | 26.6 | 26.9 | 24.8 | 24.7 | 37.1 | 47.0 | 44.4 | 39.0 | 55.5 | 66.2 | 15.5 | 51.0 | 56.3 | 50.0 | 39.3 |
| TENT+DOT | 15.4 | 16.6 | 16.4 | 14.6 | 14.5 | 20.1 | 27.7 | 26.5 | 24.1 | 35.4 | 43.3 | 13.6 | 31.4 | 35.2 | 31.3 | 24.4 |
| TENT+DELTA | 27.5 | 29.4 | 28.9 | 26.3 | 27.2 | 38.4 | 47.7 | 45.3 | 40.8 | 56.0 | 66.4 | 29.1 | 52.7 | 56.8 | 50.5 | 41.5 |
| Ent-W | 0.9 | 1.5 | 3.6 | 0.8 | 1.4 | 5.9 | 11.7 | 10.8 | 8.9 | 23.0 | 36.2 | 0.5 | 18.0 | 23.5 | 13.9 | 10.7 |
| Ent-W+TBR+Div-W(0.05) | 27.0 | 28.5 | 29.4 | 21.3 | 23.3 | 40.1 | 48.5 | 48.1 | 42.1 | 57.3 | 66.1 | 13.4 | 54.4 | 58.3 | 52.6 | 40.7 |
| Ent-W+TBR+Div-W(0.1) | 24.3 | 28.8 | 28.8 | 16.5 | 22.0 | 40.0 | 48.6 | 48.1 | 41.5 | 57.1 | 66.2 | 6.9 | 54.7 | 58.5 | 52.5 | 39.6 |
| Ent-W+TBR+Div-W(0.2) | 20.6 | 22.6 | 24.4 | 9.4 | 15.0 | 39.9 | 49.2 | 48.5 | 42.4 | 57.2 | 66.4 | 3.2 | 54.7 | 58.7 | 52.5 | 37.6 |
| Ent-W+TBR+Div-W(0.4) | 12.5 | 10.7 | 15.0 | 7.4 | 13.4 | 41.0 | 49.3 | 48.5 | 37.9 | 57.1 | 66.4 | 2.1 | 54.9 | 58.6 | 52.7 | 35.2 |
| Ent-W+TBR+LA | 7.5 | 7.6 | 13.0 | 3.6 | 6.4 | 33.8 | 39.7 | 39.3 | 30.4 | 46.3 | 54.1 | 1.7 | 44.3 | 47.3 | 42.7 | 27.8 |
| Ent-W+TBR+Sample-drop | 27.9 | 32.2 | 30.9 | 24.3 | 27.0 | 40.8 | 48.9 | 48.2 | 41.9 | 56.4 | 65.8 | 29.3 | 54.0 | 58.0 | 52.2 | 42.5 |
| Ent-W+DELTA | 30.8 | 34.4 | 33.0 | 28.7 | 29.3 | 42.8 | 49.7 | 49.1 | 43.9 | 57.2 | 65.3 | 36.7 | 54.9 | 58.6 | 52.7 | 44.5 |

