# OpenReview forum: "DELTA: DEGRADATION-FREE FULLY TEST-TIME ADAPTATION"
_ICLR.cc/2023/Conference — ICLR 2023 poster_

### Official Review · Reviewer_MFwz · 2022-10-14

**Confidence:** 4
**Correctness:** 3
**Technical Novelty And Significance:** 3
**Empirical Novelty And Significance:** 3
**Recommendation:** 6

**Clarity, Quality, Novelty And Reproducibility:**

The overall structure and exposition of the material in the paper is good. There is, however, a confusing misuse of terminology like "bias" or "IID" (as discussed above) that is even contained in the title and reduces the overall clarity of the paper. The first main contribution (using BRN for test-time adaptation) is novel and well supported. The second contribution DOT has some questionable design choices that require clarification (see above).

Minor points:
 * The authors should clarify how hyperparameters like $\alpha$ and $\lambda$ have been selected.
 * Figure 3 is confusing: sometimes methods correspond to lines and different settings go into different plots (a and b) and sometimes methods go to different plots and settings are different lines (c and d). Could this be presented more consistently?
 * Table 3: Standard deviation could be presented instead of variance because this is essential linear in the range and not quadratic.
 * Could the authors clarify why SHOT's applicability is limited? (as of the related works)
 * In Figure 1, the CI setting is still relatively balanced. For clarity of exposition, it could be helpful to have a more imbalanced class distribution here.
 * The sentence 'LAME (Boudiaf et al., 2022) is a relatively “conservative” method, which does not rectify the model’s parameters but only the model’s output. ' does not well explain what LAME does.

**Strength And Weaknesses:**

Strength:
 * Introduction and Related Works are well written.
 * The presentation and motivation of the methods is nicely structured (I like the split into diagnosis and treatment)
 * Using Batch Renormalization (BRN) instead of Batch Normalization is a simple but very good idea
 * The experimental evaluation is extensive and supports the proposed methods

Weaknesses:
 * Confusing use of terminology:
    + Using IID to refer to a setting with covariate shift ($p^{train}(x|y) \neq p^{test}(x|y)$) because samples are drawn independently is misleading since IID typically refers to situation where the test data is drawn IID from the training distribution. A better term might be "Independent Samples + Covariate Shift" (IS-CS) vs "Dependent Samples + Covariate Shift" (DS +CS)
    + Similarly, denoting class distribution shift ($P^{train}(y) \neq P^{test}(y)$) as "Class Imbalance" is confusing. This indicates that the author assume that $P^{train}(y)$ is uniform over classes (and $P^{test}(y)$ is not) - the experiments are also conducted in this setting but it is a more special setting than dealing with $P^{train}(y) \neq P^{test}(y)$. Thus, the paper should either make more clear that $P^{train}(y)$ is assumed as the uniform distribution over classes or the term "Class Shift" should be used instead of "Class Imbalance".
    + Section 3.2 confuses the "bias" and the "variance" of an estimator: judging from Figure 2, "BN adapt" has an empirical mean very close to "Global" and thus provides a (nearly) unbiased estimate of the global mean/standard deviation. However, as the authors not, "BN adapt" has high fluctuations and thus provides a high _variance_ estimate of the global mean/standard deviation. Accordingly, Batch Renormalization is a variance reduction technique here and not a debiasing technique.
   + Also, the usage of the term "bias" in Section 3.3 is unfortunate: TENT etc. do not have an intrinsic _bias_ towards certain classes and it is also no a priori necessary to "overweight" rarely predicted classes; only if one assumes that  $P^{test}(y)$ is uniform, an intervention like overweighting is recommended (and such an intervention actually _biases_ the method to predicting classes uniformly in expectation).
   + In summary, the methods discussed in this paper are not "debiasing" but rather a variance reduction technique and a useful inductive biases (uniform class frequency). It is unfortunate that the term "debiased" even made it into the title of the paper and the name of the method.
 * Doubts about the method DOT:
   * The authors motivate DOT with "[the] common constraint of making the output distribution uniform Liang et al. (2020) is no longer reasonable.". However, it seems that DOT is doing exactly that: assigning weight to samples inversely proportional to the frequency of their predicted label encourages the output distribution to be uniform. Can the authors comment on that?
   * Using a "hard label" in DOT (line 5 in Algorithm 1) seems suboptimal. Why not use the soft labels?
   * The weight in line 6 can in principle become arbitrary large; it seems more appropriate to define it as $1 / (z_t[k^*_{m_t + b}] + \epsilon)$ for some small $\epsilon$
   * Comparison to other approaches for controlling the output distribution are missing. For instance, Mummadi et al. (https://arxiv.org/abs/2106.14999) proposed a KL-based diversity regularizer for matching the empirical output distribution to a prior output distribution.

**Summary Of The Paper:**

The paper diagnosis two shortcomings of existing approaches to fully test-time adaption and proposes two methodological extensions to overcome these shortcomings. The first shortcoming is that estimates of means and variances in batch normalization (BN) have relatively high error on a per-batch level; this is addressed by replacing BN by batch renormalization (Ioffe, 2017). The second shortcoming identified is that self-learning in test-time adaptation can reinforce imbalanced class distributions; this is addressed by a dynamic online re-weighting of test samples.

**Summary Of The Review:**

The paper contains novel contributions that are well motivated and have sufficient empirical support. However, the paper in its current form, lacks clarity since some basic terminologies are used inconsistently with their definitions in statistics/machine learning. Moreover, there remain some doubts regarding DOT that need to be addressed. While the paper has sufficiently novel contributions, is needs a major revision before it can be accepted in my opinion.

## Update after author response ##
The authors have largely revised the manuscript and addressed my main concerns. I am no leaning towards recommending acceptance of this paper and have improved my score accordingly.

---

> ### Author Response · Authors · 2022-11-13
> **Response to Reviewer MFwz [1/2]**
>
> Thanks for your recognition of our motivation, presentation, and experiments. Moreover, we thank you very much for the valuable, detailed comments and suggestions. Here are our responses to your concerns, which are also clarified or fixed in the updated manuscript.
>
> **Q1 (confusing use of terminology I): IID/non-IID, class imbalance.**
>
> Thanks for the very useful comments. We have revised these terms in the revision.
>
> For "IID" and "non-IID", we use "independent" and "dependent" to describe the difference instead.
>
> For "class-imbalanced", as pointed out by the reviewer, our expression here is confusable and redundant. To be precise, "class-imbalanced" is used to describe $P^{test}(y)$: the essential issue is that "the test data is class-imbalanced", so some TTA methods may fail in such a scenario. As long as the test data is imbalanced, the problem may arise regardless of whether the trained model was learned with balanced or imbalanced training data. Certainly, $P^{train}(y)$ can affect TTA performance (via the trained model), it would be interesting to explore the TTA performance (under different $P^{test}(y)$) with models obtained with different $P^{train}(y)$; moreover, other factors in the training stage, such as data augmentations, may also affect TTA performance. The study of trained models may be beyond the scope of this work, so we focus on models learned on balanced training data using benchmark experiments from previous studies. We have clarified it in the revision.
>
> Besides, when $P^{train}(y)$ is skewed, it is common to take some techniques (e.g., resampling, logit adjustment) to bring the model closer to the one trained on balanced data. In our additional video experiments (Q3 of Reviewer Hnwy), for example, the trained model is learned on skewed training data with logit adjustment. In such a case, we also see good performance.
>
> Finally, we summarize the modified terms and their meanings in the table below (for brevity, we omit the mark of domain shift, as most of the discussions are based on the out-of-distribution data, except Table 7 which is based on in-distribution data).
>
> |     term     | meaning  |
> |:----------:|----|
> |  IS+CB   |  the test stream is independently sampled from a class-balanced distribution|
> |  DS+CB    | the test stream is dependently sampled from a class-balanced distribution|
> | IS+CI   | the test stream is independently sampled from a class-imbalanced distribution|
> | DS+CI   | the test stream is dependently sampled from a class-imbalanced distribution|
>
>
>
> **Q2 (confusing use of terminology II): The term "bias" in Section 3.2 and Section 3.3.**
>
> Thanks for the comment.
>
> For "bias" in Section 3.2, we intended to say that during per batch (in TTA, the purpose is not to output the averaged statistics after seeing all test samples, but to use these estimates online), BN adapt is strongly dominated by the current test mini-batch. Thanks for your reminder. We realized that can be ambiguous, so we have made adjustments in the revision.
>
> For "bias" in Section 3.3, we agree that "only if one assumes that $P^{test}(y)$ is uniform, an intervention like overweighting is recommended". However, we think that this does not necessarily mean that the realistic $P^{test} (y)$ is uniform. The misclassification error, $P_{x,y} (y\neq argmax_{k\in [1,K]} f(x)[k])$, is not always a suitable measure of performance, for instance, in the class-imbalanced scenario. A natural alternative is the balanced error, which averages each of the per-class error: $\frac{1}{K} \sum_{y \in [1,K]} P_{x|y}(y \neq argmax_{k\in [1,K]} f(x)[k])$. It can be seen as implicitly using a balanced class-probability function $P(y|x) \propto \frac{1}{K} \cdot P(x|y)$ v.s. the $P(y|x) \propto P(y) \cdot P(x|y)$ in the misclassification error measure (Menon et al., 2021). Thus, in accordance with research in the field of long-tailed/class-imbalanced learning, we consider a model that clearly favors some categories to be biased from the perspective of the balanced error measure.
>
> The labels in TTA are produced by the model itself, we find that the predictions of the trained model are skewed (even for the model trained on balanced training data and tested on balanced test data, as shown in Figure 3). While, the entropy-minimization loss tends to focus more on low-entropy samples, making the model more biased towards those dominant classes. We agree with the reviewer that TENT does not have an intrinsic class bias (the bias originates from the trained model), our original expression can be imprecise. We thank the reviewer for the detailed suggestions and we have revised them in the manuscript.
>
> ```
> Menon et al., Long-tail learning via logit adjustment, ICLR 2021.
> ```

---

> ### Author Response · Authors · 2022-11-13
> **Response to Reviewer MFwz [2/2]**
>
>
> **Q3 (about DOT I): The authors motivate DOT with "[the] common constraint of making the output distribution uniform Liang et al. (2020) is no longer reasonable.". However, it seems that DOT is doing exactly that: assigning weight to samples inversely proportional to the frequency of their predicted label encourages the output distribution to be uniform. Can the authors comment on that?**
>
> SHOT (also the work listed in Q6) uses a KL-divergence penalty to enforce the output distribution to be uniform. However, this is not always reasonable for TTA (TTA is similar to transductive learning in that the data used for learning is exactly the final test data). For example, for class-imbalanced test data, forcing the output to be uniform will harm performance.
>
> An unbiased model (from the perspective of balanced error measure) is built on healthy training data (regarding the number of samples per class, data quality, intra-class diversity, inter-class relationship, etc.), such a model will not output uniform results on the class-imbalanced test samples. Inspired by this, we designed DOT, which attempts to update the model with class-balanced data (via reweighting) in TTA. DOT does not force the model's output to be uniform.
>
> In the response to Q6, we added the experiment to compare DOT with KL-div. It can be seen that KL-div is clearly inferior in DS+CB, IS+CI, and DS+CI scenarios, supporting the preceding opinion.
>
>
> **Q4 (about DOT II): Using a "hard label" in DOT (line 5 in Algorithm 1) seems suboptimal. Why not use the soft labels?**
>
> Line 5 is to get a class index, which needs to be an integer, then we can obtain the sample weight based on the moving class-frequency vector $z$ (Line 6). In fact, in Line 10, the class-frequency vector $z$ is updated with the soft labels (the original output probabilities).
>
>
> **Q5 (about DOT III): The weight in line 6 can in principle become arbitrary large; it seems more appropriate to define it as** $1 / (z_t[k^*_{m_t + b}] + \epsilon)$  **for some small** $\epsilon$.
>
> Thanks for pointing this out. $1 / (z_{t-1}[k^*_{m_t + b}] + \epsilon)$ is indeed more rigorous. We have revised it (after checking, it does not affect our experimental results).
>
>
> **Q6 (about DOT IV): Comparison to other approaches for controlling the output distribution are missing. For instance, Mummadi et al. ([https://arxiv.org/abs/2106.14999](https://arxiv.org/abs/2106.14999)) proposed a KL-based diversity regularizer for matching the empirical output distribution to a prior output distribution.**
>
> Thank you for bringing this to our attention. The experimental results are shown in the table below. Due to a hyper-parameter being required to control the regularization strength, we examine multiple choices from 0.1 to 10000.0 in IS+CB scenario and only perform experiments with the promising strength in other scenarios. The results demonstrate DOT's superiority and also support the responses to Q3.
>
>
> |          | IS+CB  | DS+CB | IS+CI   | DS+CI |
> |----------|:----:|:-------:|:----:|:----------:|
> | +KL0.1     | 41.3 | -       | -    | -          |
> | +KL1.0     | 41.3 | -       | -    | -          |
> | +KL10.0    | 42.0 | -       | -    | -          |
> | +KL100.0   | 47.6 | 39.9    | 46.6 | 36.5       |
> | +KL1000.0  | 48.9 | 27.7    | 43.1 | 22.5       |
> | +KL10000.0 | 1.3  | -       | -    | -          |
> | +DOT       | 49.9 | 47.5    | 48.4 | 44.5       |
>
>
> **Q7 (minor points I): The authors should clarify how hyperparameters like $\alpha$ and $\lambda$ have been selected.**
>
> Thanks for the comments. The hyper-parameters $\alpha$ and $\lambda$ are roughly selected from {0.9, 0.95, 0.99, 0.999} on the validation set (e.g., out of the 15 corruption types on the benchmark). Besides, we also find that the method works under a wide range of $\alpha$ and $\lambda$ (Figure 5).
>
>
> **Q8 (minor points II): Could the authors clarify why SHOT's applicability is limited? (as of the related works)**
>
> SHOT modifies the model architecture and the training loss, which are not very common. For source (training data) free setting, the methods are expected to be based on the most common models. Certainly, the main ideas of SHOT still can be used even if the model is not trained with such modifications, though the performance will be lower than the reported one. We have clarified it in the revision.
>
>
> **Q9 (minor points III): Figure 3 is confusing. Could this be presented more consistently?
> Table 3: Standard deviation could be presented instead of variance because this is essential linear in the range and not quadratic.
> In Figure 1, the CI setting is still relatively balanced. For clarity of exposition, it could be helpful to have a more imbalanced class distribution here.
> The sentence 'LAME (Boudiaf et al., 2022) is ...' does not well explain what LAME does.**
>
> Thanks for the detailed suggestions. We have improved them in the revision.

---

> > ### Author Response · Authors · 2022-11-15
> > **Additional discussion on Q4, the "hard label" in DOT (line 5 in Algorithm 1)**
> >
> > One can discard line 5 and modify line 6 to adopt the original soft probabilities: $\omega_{m_t+b} = \sum_{k=1}^K 1 / ( z_{t-1}[k] + \epsilon) * p_{m_t+b}[k]$. We compare the hard label strategy (Algorithm 1) with the soft one in the table below (on the basis of Enw-W+TBR, on ImageNet-C). We find that both strategies work well in all scenarios, demonstrating the effectiveness of the idea of DOT. The performance of the soft strategy is slightly worse than the hard strategy in some scenarios. However, we think it is difficult to say "hard labels are necessarily better than soft labels" or "soft labels are necessarily better than hard labels", for example, the two strategies both exist in recent semi-supervised methods: hard label in FixMatch, soft label in UDA. We will also include this discussion in the appendix.
> >
> > | | IS+CB | DS+CB ($\rho=1.0$) | DS+CB ($\rho=0.5$) | DS+CB ($\rho=0.1$) | IS+CI ($\pi=0.1$) | IS+CI ($\pi=0.05$) | DS+CI ($\rho=0.5, \pi=0.1$) | DS+CI ($\rho=0.5, \pi=0.05$) |
> > |-------|-------|------------------|------------------|------------------|-----------------|------------------|---------------------------|----------------------------|
> > | Hard | 49.9 | 48.3             | 47.4             | 43.2             | 48.4            | 47.7             | 45.4                      | 44.8                       |
> > | Soft | 49.7 | 48.0   | 47.3        | 43.0      | 48.3    | 47.5        | 45.1                     | 44.5                   |
> >
> > ```
> > FixMatch: Simplifying Semi-Supervised Learning with Consistency and Confidence, NeurIPS 2020
> > Unsupervised Data Augmentation for Consistency Training, NeurIPS 2020
> > ```

---

> > > ### Comment · Reviewer_MFwz · 2022-11-16
> > > **Feedback to response**
> > >
> > > I would like to thank the authors for their response/revision. The paper has undergone a major revision and the response addresses many of my concerns. Also the most recent comparison of soft vs. hard labelling was what I was referring to.
> > > One remaining concern is the statement "The hyper-parameters and are roughly selected from {0.9, 0.95, 0.99, 0.999} on the validation set (e.g., out of the 15 corruption types on the benchmark)". In no case should hyper-parameters be tuned on the test corruptions - that is why Imagenet-C comes with 4 extra corruptions for model validation. Figure 5 indicates that the choice of hyper-parameters is not critical but still I think the method for tuning hyperparameters might be problematic. Can the authors comment on this issue?

---

> > > > ### Author Response · Authors · 2022-11-16
> > > > **Response to the remaining concern**
> > > >
> > > > Thank you for your feedback. We are glad to hear that most of your concerns were addressed.
> > > >
> > > > For the remaining concern, we agree with the reviewer that "in no case should hyper-parameters be tuned on the test corruptions". The validation sets used in our experiments are exactly the 4 extra corruptions: 'gaussian blur', 'saturate', 'spatter', 'speckle noise' (we mean "outside the 15 test corruption types" in our original response).
> > > >
> > > >
> > > > For example, the used values of $\alpha$ and $\lambda$ for CIFAR100-C (0.95 and 0.9) are selected according to the tables below, which are the averaged accuracies on the 4 extra corrupted sets. Note that the selection is based on the IS+CB scenario and the selected values are shared across different scenarios and different basic methods (TENT, PL, Ent-W), which is also normative because we cannot know the actual scenarios in advance. Besides, the values in the candidate set {0.9, 0.95, 0.99, 0.999} are the most commonly used values in momentum update.
> > > >
> > > > | $\alpha$ = 0.9   | $\alpha$ = 0.95  | $\alpha$ = 0.99  | $\alpha$ = 0.999 |
> > > > |-------|-------|-------|-------|
> > > > | 73.28 | 73.31 | 72.59 | 71.84 |
> > > >
> > > > | $\lambda$ = 0.9   | $\lambda$ = 0.95  | $\lambda$ = 0.99  | $\lambda$ = 0.999 |
> > > > |-------|-------|-------|-------|
> > > > | 73.31 | 73.10 | 72.84 | 72.73 |
> > > >
> > > >
> > > > Figure 5 is indeed the performance of the test sets, while it is just used to reflect the impacts of $\alpha$ and $\lambda$ (actually, from Figure 5, the results with other choices are even better here, e.g., $\alpha=0.9$, $\lambda=0.85$). The figure is easy to cause misunderstanding, replacing Figure 5 with the results on the validation set might be better, we will do this soon.

---

> > > > > ### Comment · Reviewer_MFwz · 2022-11-17
> > > > > **Review updated**
> > > > >
> > > > > Thanks for the clarification. I have increased the score in the review to reflect that my main concerns have been addressed sufficiently.

---

> > > > > > ### Author Response · Authors · 2022-11-17
> > > > > > **Thanks for the positive feedback on our response**
> > > > > >
> > > > > > Thanks for your feedback and support. We think the manuscript has been significantly improved with the help of the comments and suggestions, thank you for your efforts. Besides, to avoid misunderstanding, Figure 5 has been replaced and the sentences have been clarified in the manuscript.

---

### Official Review · Reviewer_BBcu · 2022-10-23

**Confidence:** 4
**Correctness:** 3
**Technical Novelty And Significance:** 3
**Empirical Novelty And Significance:** 3
**Recommendation:** 6

**Clarity, Quality, Novelty And Reproducibility:**

The paper is well-writen and novel. The proposed TBW is not described clearly and I can not reproduce it.

**Strength And Weaknesses:**

**Positive Points:**

 - The authors point out the normalization statistics may be biased with the current test mini-batch and the optimization gradient may be biased with the dominant class.

 - The proposed methods including TBR and DOT are simple and effective.

 **Negative Points:**

 - In Section 3.2, I suggest the authors can provide more descriptions of stopping gradient operation. In my opinion, the word "stopping gradient" is confusing and I can not understand the meaning of this. Some readers may be unfamiliar with this and have to check for the orginal batch renormalization paper (Ioffe, 2017). Beside, the batch renormalization seems to be different from the original one. Thus I strongly recommend the authors can provide a detailed Algorithm of TBR like Algorithm 1 (DOT) of the paper.

 - About TBR, some details are missing. What is the output of batch renormalization? $v^*$ or $\gamma v^* + \beta$? And do $\hat{\sigma}^{ema}$ and $\hat{\mu}^{ema}$ need to be updated?

- I noitice the authors mentioned "TBR discards the warm-up and truncation operation". Thus, in this case, $v^{*} = (v - \hat{\mu}^{batch}) / \hat{\sigma}^{batch} \cdot r + d = (v - \hat{\mu}^{batch}) / \hat{\sigma}^{batch} \cdot (\hat{\sigma}^{batch} / \hat{\mu}^{ema}) + (\hat{\mu}^{batch} - \hat{\mu}^{ema}) / \hat{\sigma}^{ema} = (v- \hat{\mu}^{ema}) / \hat{\sigma}^{ema}$. I found that it is the same as the original BN because the authors remove truncation operation, right? The differences between the proposed TBR and the regular BN are unclear.

 - In the analysis of the results in Table 2, the authors state "Different from the TEMA, TBR is well compatible with gradient-based adaptation methods". Could the authors give more explanations?

 - In Table 3, what are variance and range? Do the authors calculate them in some features of a layer?

 - In Table 3, what are the differences between the baselines efficient test-time adaptation (ETA) and entropy-based weighting (Ent-W)?

  - Would the authors release the code upon acceptance?

**Summary Of The Paper:**

**Summary:**

This paper presents a new Test-time adaptation (TTA) method called DELTA for debiased fully TTA. To be specific, the authors 1) introduce batch renormalization to alleviate the bias in normalization statistics, 2) propose dynamic online re-weighting (DOT) to address the class bias within optimization.

**Summary Of The Review:**

The proposed method alleviates the bias resulted from the small mini-batch and dominant class, which is simple and effective. Thus I vote for accept. But some importance details in TBR are missing or unclear. I hope the authors can address my concerns.

---

> ### Author Response · Authors · 2022-11-13
> **Response to Reviewer BBcu [1/2]**
>
> Thanks for your recognition of our motivation and methods. We think most of your concerns are misled by our confusable presentation which is fixed in the updated manuscript. Here are our responses to your concerns.
>
> **Q1: In Section 3.2, I suggest the authors can provide more descriptions of stopping gradient operation. I strongly recommend the authors can provide a detailed Algorithm of TBR like Algorithm 1 (DOT) of the paper.**
>
> Thanks for the suggestions. "stopping-gradient" means that the tensor will not be taken into account for computing gradients ($\hat{\sigma}^{batch}$ and $\hat{\mu}^{batch}$ can contribute gradients originally). It is exactly the `Tensor.detach()` function in PyTorch and the `tf.stop_gradient()` function in TensorFlow. We have clarified it in the revision and provided an Algorithm description for TBR in the appendix.
>
>
> **Q2: What is the output of batch renormalization?** $v^*$ **or** $\gamma v^*+\beta$. **And do** $\hat{\sigma}^{ema}$ **and** $\hat{\mu}^{ema}$ **need to be updated?**
>
> $\gamma v^*+\beta$, the affine parameters are needed. $\hat{\sigma}^{ema}$ and $\hat{\mu}^{ema}$ are updated via moving average during test. We have clarified them in the text and the Algorithm description.
>
>
> **Q3: I noitice the authors mentioned "TBR discards the warm-up and truncation operation". Thus, in this case,  $v^{*} = (v - \hat{\mu}^{batch}) / \hat{\sigma}^{batch} \cdot r + d = (v - \hat{\mu}^{batch}) / \hat{\sigma}^{batch} \cdot (\hat{\sigma}^{batch} / \hat{\mu}^{ema}) + (\hat{\mu}^{batch} - \hat{\mu}^{ema}) / \hat{\sigma}^{ema} = (v- \hat{\mu}^{ema}) / \hat{\sigma}^{ema}$. I found that it is the same as the original BN because the authors remove truncation operation, right? The differences between the proposed TBR and the regular BN are unclear.**
>
> They are not the same.
>
> "truncation" refers to clamping $r$ into $[1/r_{max}, r_{max}], r_{max} \geq 1$ and clamping $d$ into the range $[-d_{max}, d_{max}], d_{max} \geq 0$. "warm up" refers to use $r_{max}=1$ and $d_{max}=0$ in the first several iterations and gradually relax them to reach the predefined values, e.g., $r_{max}=3$ and $d_{max}=5$. These two operations are used to stabilize training in Ioffe (2017). However, in TTA, as we start with a trained model, we find that such operations are insignificant here. To avoid introducing too many hyper-parameters, we discard the two operations in our design.
>
> The difference between the derivation in Q3 and Equation (1) in our manuscript is the $sg(\cdot)$ operation. In Equation (1), $v, \hat{\mu}^{batch}, \hat{\sigma}^{batch}, v^*$ can have attachments with the current gradients (for methods like TENT which involve gradient-based optimization); $r, d$ have no attachments with gradients even for methods like TENT (the test-time moving averages $\hat{\mu}^{ema}, \hat{\sigma}^{ema}$ also do not involve gradients, as it is unlikely to track all the historical tensors in the computational graph in practice).
>
> If $r=\hat{\sigma}^{batch} / \hat{\sigma}^{ema}, d=(\hat{\mu}^{batch} - \hat{\mu}^{ema}) / \hat{\sigma}^{ema}$, i.e., without $sg(\cdot)$, TBR will become the method TEMA as the reviewer's derivation. However, as mentioned in the manuscript, for methods like TENT, TEMA can destroy the model (e.g., Table 2) as the gradient descent optimization does not consider the normalization operation, leading to unlimited growth of model parameters. While, in TBR, the $\hat{\mu}^{batch}, \hat{\sigma}^{batch}$ make gradient optimization take into account the normalization operation. In summary, though Equation (1) appears to have the same forward output as in TEMA, they result in completely different gradients and parameter update.
>
> Besides, if $r \equiv 1, d \equiv 0$, TBR will become BN adapt, which only uses the current test mini-batch, resulting in inaccurate statistics as shown in Diagnosis I. While, in TBR, $r, d$ can rectify the normalized features. We summarize the three test-time normalization methods below.
>
> BN adapt: $v^* = (v-\hat{\mu}^{batch}) / \hat{\sigma}^{batch}$
>
> TEMA: $v^* = (v-\hat{\mu}^{ema}) / \hat{\sigma}^{ema}$
>
> TBR: $v^* = (v-\hat{\mu}^{batch}) / \hat{\sigma}^{batch} \cdot sg(\hat{\sigma}^{batch}) / \hat{\sigma}^{ema}  + (sg(\hat{\mu}^{batch}) - \hat{\mu}^{ema}) / \hat{\sigma}^{ema}$

---

> ### Author Response · Authors · 2022-11-13
> **Response to Reviewer BBcu [2/2]**
>
>
> **Q4: In the analysis of the results in Table 2, the authors state "Different from the TEMA, TBR is well compatible with gradient-based adaptation methods". Could the authors give more explanations?**
>
> The response to Q3 may also help explain this question. As mentioned in the above response, for the gradient-related adaptation methods like TENT, TEMA can destroy the model (e.g., Table 2) as the gradient descent optimization does not consider the normalization operation, leading to unlimited growth of model parameters. TBR, on the other hand, makes gradient optimization take into account the normalization operation, resulting in good performance. As a result, we conclude that TBR is well compatible with TENT-like methods.
>
>
> **Q5: In Table 3, what are variance and range? Do the authors calculate them in some features of a layer?**
>
> They are the variance and range (maximum-minimum) of the number of predictions for each category. For example, assuming there are 90 samples in the test stream (belonging to 3 classes, 30 samples per class actually); the model predicts 50 samples as the 1st class, 30 samples as the 2nd class, 10 samples as the 3rd class, then we have:
> $Variance  = [(50-30)^2 + (30-30)^2 + (10-30)^2]/3$,
> $Range = 50 - 10$.
> The Variance and Range indicate that the trained model is biased toward some classes (if not, the model should give uniform predictions on the balanced test set; this is not to say that every sample must be correctly predicted).
>
> Following the suggestion of Reviewer MFwz, we modified the "Variance" to "Standard Deviation".
>
>
> **Q6: In Table 3, what are the differences between the baselines ETA and Ent-W?**
>
> (We guess there is a typo in Q6: Table 3 -> Table 4)
>
> ETA consists of two components: Ent-W and Div-W. We find that Div-W is not a very robust component, so we add DELTA to Ent-W individually. We also construct an ablation study on Div-W as a competitor to DOT as shown in Table 10 (though Div-W is not designed for combating class imbalance).
>
>
> **Q7: Would the authors release the code upon acceptance?**
>
> Yes. The code will be available online.

---

> > ### Comment · Reviewer_BBcu · 2022-11-17
> > **Response to authors**
> >
> > Thanks for the response from the authors. The rebuttal has addressed most of my concerns. After reading the comments of Reviewer Hnwy and MFwz,  I would keep my scoring *weak accept*.

---

> > > ### Author Response · Authors · 2022-11-17
> > > **Thanks for the feedback and support**
> > >
> > > Thanks for your feedback and support. We are happy that we could address your concerns.

---

> > > ### Author Response · Authors · 2022-11-25
> > > **Response to Reviewer BBcu after receiving the feedback from the other reviewers**
> > >
> > >
> > > Dear reviewer BBcu:
> > >
> > > We are delighted to share with you that the concerns of Reviewer Hnwy and MFwz have been addressed now.
> > >
> > > Best,
> > >
> > > Authors

---

> > ### Public Comment · ~J_W2 · 2023-02-22
> > **Request for the code repo**
> >
> > "Q7: Would the authors release the code upon acceptance?
> >
> > Yes. The code will be available online."
> >
> > Hello, I’m wondering if you could kindly point me to where I can access the code. Thank you~

---

> > > ### Author Response · Authors · 2023-02-22
> > > **Response**
> > >
> > > Thank you for your interest in our work. The code can be found at https://github.com/bwbwzhao/DELTA

---

### Official Review · Reviewer_Hnwy · 2022-10-24

**Confidence:** 4
**Correctness:** 4
**Technical Novelty And Significance:** 2
**Empirical Novelty And Significance:** 3
**Recommendation:** 6

**Clarity, Quality, Novelty And Reproducibility:**

Quality
---

Overall, the paper is of relatively high quality. There are two main ways to improve the paper's quality, both of which are important. First, without some notion of error bars in the experimental results, it is not always straightforward to tell whether a result really is significant. The most obvious example of this is Table 7, in which the proposed method is claimed to improve performance on the in distribution CIFAR-100 test set by 0.1%, but this seems within the range of noise. Certainly some aspects of the evaluation pipeline are random, e.g., the order in which test points are presented, and all randomized aspects should be run with multiple seeds and the standard error of the final results should be reported in all tables.

Second, the paper would benefit greatly from having experiments on a real world application exhibiting non-IID-ness and class imbalance. Video prediction datasets would be a natural fit here. Right now, it is up to the reader to decide whether the studied setting is even of practical relevance, and while personally I believe it is, it is best to not leave this up to the reader. Furthermore, experiments in such a setting would simply serve to determine whether the proposed method truly works as advertised, on naturally occurring problems rather than synthetically created ones.

Clarity
---

Overall, the paper is well written. My only major concern is that I do not believe "bias" is the correct term to describe the fact that the BN statistics "are far from the ideal global values and fluctuate dramatically during adaptation" (page 4). Is this not variance?

Originality
---

As stated, the problem setting considered by this paper is relatively understudied in the test time adaptation literature and of potential importance. My main concern related to originality is still that there is no clear example of a real world application exhibiting the properties of the proposed problem setting.

**Strength And Weaknesses:**

Strengths
---

- The paper provides a number of experiments and ablations which demonstrate the effectiveness of the proposed approach.
- The paper is reasonably well written and well motivated.
- The paper studies a potentially important scenario that has received less attention in prior papers studying test time adaptation to distribution shift.

Weaknesses
---

- There are no error bars in the experimental results.
- Use of the term "bias" seems incorrect -- see next section for more detail.
- There are no experiments studying a real world scenario in which the problems of non-IID-ness and class imbalance occur naturally.

**Summary Of The Paper:**

This paper proposes two improvements to test time adaptation in the streaming data setting: one to address the possibility of non IID test data, and one to address the possibility of class imbalance in the test distribution. These improvements are relatively plug-and-play with existing popular adaptation approaches, e.g., adaptation via batch normalization and entropy minimization. Taken together, these improvements seem to consistently improve adaptation performance on CIFAR and ImageNet distribution shift test sets in which non-IID-ness and class imbalance are artificially introduced.

**Summary Of The Review:**

In summary, the paper is reasonably well written and the experiments are sound, but there are several important improvements that can be made as laid out above. I am recommending weak reject and am happy to discuss further.

Edit after author response
---

I apologize for my late response. I believe that the authors have adequately addressed my concerns and I am updating my recommendation to weak accept.

---

> ### Author Response · Authors · 2022-11-13
> **Response to Reviewer Hnwy [1/2]**
>
> Thank you for the positive affirmations about the studied problem, our writing, motivation, experiments, and ablation studies. We also thank you for the valuable suggestions. Here are our responses to your concerns, which are also addressed in the updated manuscript.
>
> **Q1: There are no error bars in the experimental results (the most obvious example of this is Table 7, in which the proposed method is claimed to improve performance on the in distribution CIFAR-100 test set by 0.1%, but this seems within the range of noise).**
>
> Thanks for pointing this out. In the revision, we have updated the averaged results with error bars (with 5 runs) in Table 4, Table 6 (left part), Table7, and Table8, showing the results are reliable. For others, e.g., Table 5, and Table 6 (right part), the improvements are very significant and will not affect the judgment, we will update them soon in the next few days.
>
> As discussed in the appendix (paragraph 'Influence of random seeds'), for test-time adaptation methods like TENT, which do not introduce additional randomness, the results are always the same on the same test stream (same sample order), as the model is updated from the trained parameters, and the random seed can only affect sample order. For the other methods like CoTTA, the random seed can also affect model update.
>
> In particular, after averaging the multiple runs in Table 7, we find that the improvements increase to 0.4% and 0.5% on the basis of TENT and Ent-W, respectively. We provide the separate results in the table below, showing that DELTA achieves consistent improvement. Besides, the improvement in Table.7 is relatively marginal, as the TTA methods are primarily used for out-of-distribution data, and there may be not much room for improvement on in-distribution data for TTA methods.
>
> |       | seed=2020 | seed=2021 | seed=2022 | seed=2023 | seed=2024 | mean | std  |
> |-----------|:--------:|:--------:|:--------:|:--------:|:--------:|:----:|:----:|
> | TENT        | 78.7     | 78.4     | 78.6     | 78.3     | 78.4     | 78.5 | 0.16 |
> | TENT+DELTA  | 78.8 (+0.1)   | 78.9 (+0.5)    | 78.9 (+0.3)     | 78.9 (+0.6)     | 78.9   (+0.5)   | 78.9 (+0.4)| 0.03 |
> | Ent-W       | 78.8     | 78.6    | 78.3    | 78.4   | 78.7    | 78.6 | 0.19 |
> | Ent-W+DELTA | 79.0 (+0.2)   | 79.0 (+0.4)    | 79.2  (+0.9)    | 79.2 (+0.8)    | 79.1 (+0.4)    | 79.1 (+0.5)  | 0.09 |
>
>
> **Q2: Use of the term "bias" seems incorrect.**
>
> Thank you for bringing this to our attention. We intended to say that BN adapt is strongly dominated by the current test mini-batch per batch (in TTA, the goal is not to output the averaged statistics after seeing all test samples, but to use these estimates online). Thank you for the reminder. We recognize that it may be ambiguous (Reviewer MFwz suggests we revise the term too), and we have made changes in the revision (we describe that the normalization statistics are inaccurate within each test mini-batch in Section 3.2 of the updated manuscript to avoid confusion and we also revise other relevant descriptions).

---

> > ### Author Response · Authors · 2022-11-14
> > **Error Bars**
> >
> > We have added error bars to the main reported results (Table 4 - Table 8) in the updated manuscript, including Table 5 and Table 6 (right part).

---

> > ### Author Response · Authors · 2022-11-17
> > **Error Bars #2**
> >
> > We have added error bars to Table 2, Table 3, Table 9, and Table 10 in the updated manuscript. By now, all results are reported with error bars. Thank you for the reminder, which helped us improve the manuscript.

---

> ### Author Response · Authors · 2022-11-14
> **Response to Reviewer Hnwy [2/2]**
>
> **Q3: There are no experiments studying a real world scenario in which the problems of non-IID-ness and class imbalance occur naturally.**
>
> Thanks for the constructive comment and suggestion. We construct a video-related experiment to redress this deficiency. We consider an automatic video content moderation task (e.g., for the short-video platform), which needs to recognize the categories of interest from the extracted frames. In such a scenario, the problems of non-IID-ness and class imbalance can be encountered.
>
> We collected 1686 test videos from YouTube which are annotated in the YouTube-BoundingBoxes dataset. In this experiment, 49006 video segments are extracted from these videos to form the test stream, which contains 21 categories. A ResNet18 trained on the related images from COCO dataset is employed as the trained model. As a result, there is a natural difference between the training domain and the test domain. The consecutive video segments form the natural non-IID-ness scenario (an object usually persists over several frames), please see the visualization in the appendix (Figure 8 in the updated manuscript). Moreover, the class distribution is naturally skewed, we also provide a visualization in the appendix.
>
> The results are listed in the table below. The classical methods, like the results on the synthetic datasets, suffer from non-IID-ness and imbalance. DELTA, on the other hand, assists them in achieving good performance. The experiment is also included in the revision. Thank you once more for your constructive comments, which helped us improve the manuscript.
>
> | Method      | Acc  |
> |-----------|:----:|
> | Source      | 74.0$_{\pm 0.00}$   |
> | BN adapt    | 51.4$_{\pm 0.29}$ |
> | ETA    | 51.5$_{\pm 0.32}$ |
> | PL          | 51.7$_{\pm 0.31}$ |
> | PL+DELTA    | 75.5$_{\pm 0.22}$ |
> | TENT        | 51.7$_{\pm 0.27}$ |
> | TENT+DELTA  | 75.7$_{\pm 0.21}$ |
> | Ent-W       | 51.5$_{\pm 0.28}$ |
> | Ent-W+DELTA | 76.2$_{\pm 0.23}$ |

---

> ### Author Response · Authors · 2022-11-21
> **Response to Reviewer Hnwy [3/-]**
>
> Dear reviewer Hnwy:
>
> Thank you again for your constructive comments and suggestions, which have helped us improve the manuscript significantly. We would be happy to know whether our responses and the updated manuscript have addressed your concerns. We are happy to provide other clarification and responses.
>
> Looking forward to hearing from you!
>
> Best,
>
> Authors

---

> ### Author Response · Authors · 2022-11-25
> **Thanks for your positive feedback with score updating**
>
> Thanks for your feedback and support. We are happy that we could address your concerns.

---

### Official Review · Reviewer_E7Gz · 2022-10-25

**Confidence:** 4
**Correctness:** 3
**Technical Novelty And Significance:** 3
**Empirical Novelty And Significance:** 3
**Recommendation:** 6

**Clarity, Quality, Novelty And Reproducibility:**

The class imbalance settings of TTA and the perspective of addressing the issue of TTA are novel.

**Strength And Weaknesses:**

Strength:

1.	This paper introduces a new setting, namely the class imbalance issue during the test-time process, in the test-time adaptation.

2.	The authors dig out the issues of existing test-time adaptation methods (i.e., the bias issue) from a new perspective.

3.	Extensive experiments on the ImageNet-C, ImageNet-R, and CIFAR 100-C demonstrate the effectiveness of the proposed method.

Weakness:

1.	The technical contribution is not very significant. For example, the TBR just adopts the batch renormalization technique from the initial Batch Normalization paper, and the re-weighting technique is derived from class-wise re-weighting.

2.	As referred to the Treatment II, some components of DOT have multiple options. However, the ablation studies about the components of DOT and the selection of the components of DOT may be missing.

3.	In Table 8, when conducting experiments on the real-world out-of-distribution dataset ImageNet-R, the improvement of DELTA is marginal compared with that on the ImageNet-C dataset, more discussions are required.

4.	In Table 5, how about the performance of DELTA with \rho<0.1 (e.g., 0.01?), i.e., totally concentrate on one same class during a period. Similarly, could the authors provide more results regarding smaller \pi<0.05 (e.g., 0.001)? I am curious about these results since any value of \pi and \rho may appear in practice.

5.	In Table 10, could the authors further provide the results of Div-W+Fisher regularization (namely EATA, this is the full version of ETA in Niu et al, 2022)?

6.	Figure 1 is somewhat confusing. It is hard to distinguish the difference between IID and CI, non-IID and CI & non-IID.

7.	In Section 3.1, it would be better to detail describe each scenario and the difference between the previous TTA setting.

8.	In Treatment II of Section 3.3, it would be better to extend the “L x” to “Line x of Algorithm 1” to improve the readability of the paper.


**Summary Of The Paper:**

This paper presents a method named Debiased Fully Test-time Adaptation (DELTA) to address the biased issue in test-time adaptation. Specifically, the authors conduct experiments to verify the claims that 1) the normalization statistics tend to fit the current test mini-batch, and 2) the test-time adaptation optimization would bias to some dominant classes. The authors first adopt the renormalization technique to alleviate the biases in the normalization statistics of batch normalization, and then devise a re-weighting module to assign different weights for test samples with different pseudo labels to address the optimization issue. Extensive experiments on the ImageNet-C, ImageNet-R, and CIFAR 100-C demonstrate the effectiveness of the proposed method. However, I have some concerns about this paper. My detailed comments are as follows.

**Summary Of The Review:**

This paper proposes a new setting of test-time adaptation, namely class imbalance in the mini-batch, and proposes to address this problem from the de-bias perspective, which is interesting. Though the pure technical contribution is not very significant, it is new in the area of TTA. Some results are still missing to convince me regarding its effectiveness.

---

> ### Author Response · Authors · 2022-11-13
> **Response to Reviewer E7Gz [1/2]**
>
> Thank you for acknowledging the work's insight. Also, thank you for your insightful comments. The responses to your concerns are as follows, and they are also clarified or corrected in the revised version of our manuscript.
>
>
> **Q1: The technical contribution is not very significant.**
>
> We agree that the underlying tools we used are not entirely novel. However, as you and the other reviewers mentioned, our main contribution is to expose and elegantly solve the hidden problems in the existing TTA methods. We believe the work will help with stronger test-time adaptation.
>
> Furthermore, the components TBR and DOT are improved and specifically designed for TTA rather than being directly borrowed from other fields. As compared in the paper, naive "A+B" yields suboptimal results.
>
> **Q2: As referred to the Treatment II, some components of DOT have multiple options. However, the ablation studies about the components of DOT and the selection of the components of DOT may be missing.**
>
> The expression  `Some components in Algorithm 1 have multiple options` in our manuscript may be ambiguous. It is primarily intended to demonstrate how DOT can be used to improve a variety of TTA techniques; the 'components' do not refer to DOT's own components.
>
> For the "Forward (·)" function (Line 3 of Algorithm 1), in Table 9, #3 indicates "forward with BN adapt",  #4 indicates "forward with TBR". "With TBR" is better.
>
> For the loss function $\mathcal{L}(\cdot)$ (Line 8 of Algorithm 1), in Table 4-8, PL is based on the loss
>
> $\mathcal{L}(p_b) = - \mathbb{I}_{p_b[k^*_b] \geq \tau} \cdot \log p_b[k^*_b]$,
>
> TENT is based on the loss
>
> $\mathcal{L}(p_b) = -\sum_{k=1}^K p_b[k] \log p_b[k]$,
>
> Ent-W is based on the loss
>
> $\mathcal{L}(p_b) = - \mathbb{I}_{H_b < \tau} \cdot e^{\tau - H_b}$
>
> $\cdot \sum_{k=1}^K p_b[k] \log p_b[k]$.
>
>
> Ent-W is usually better. We have clarified it in the revision.
>
> **Q3: In Table 8, when conducting experiments on the real-world out-of-distribution dataset ImageNet-R, the improvement of DELTA is marginal compared with that on the ImageNet-C dataset, more discussions are required.**
>
> Thank you for the suggestion. ImageNet-C is constructed based on the original test set of ImageNet, while ImageNet-R is collected individually. ImageNet-R consists of more hard cases which are still difficult to recognize for methods with DELTA. On the other hand, the improvement of DELTA also varies on the 15 different corruption types of ImageNet-C as shown in the appendix. We have added the discussion in the revision.
>
> Besides, in the revision, we update the results of Ent-W+DELTA in Table 8: the improvement of DELTA is 1.3% actually, though it is still smaller than the average gain on ImageNet-C.
>
> (to provide the error bar suggested by Reviewer Hnwy, we rerun the experiments and find a bug in "Ent-W+DELTA" of Table 8: we incorrectly rearranged the samples in the experiment. The results of other methods in Table 8 and in other tables are rechecked and they are not affected.)
>
> **Q4: In Table 5, how about the performance of DELTA with $\rho<0.1$ (e.g., 0.01?). Similarly, could the authors provide more results regarding smaller $\pi<0.05$ (e.g., 0.001)?**
>
> Thanks for the constructive comment. We agree that it is an interesting point for discussion. We add the experiments on ImageNet-C. The results are provided in the table below, showing DELTA can manage the intractable cases. We also added these results in the appendix.
>
> |    | $\rho = 0.01$ | $\pi = 0.001$ |
> |----------|:---------------:|:---------------:|
> | Source   | 18.0$_{\pm 0.00}$         | 17.9$_{\pm 0.00}$          |
> | BN adapt | 6.8$_{\pm 0.10}$           | 31.1$_{\pm 0.43}$          |
> | ETA      | 3.3$_{\pm 0.11}$           | 44.1$_{\pm 0.38}$          |
> | LAME     | 26.0$_{\pm 0.05}$          | 17.4$_{\pm 0.26}$          |
> | CoTTA    | 7.0$_{\pm 0.10}$           | 33.5$_{\pm 0.44}$          |
> | CoTTA*   | 7.2$_{\pm 0.11}$           | 33.6$_{\pm 0.62}$          |
> | PL       | 6.6$_{\pm 0.11}$           | 37.9$_{\pm 0.45}$          |
> | +DELTA   | 34.2$_{\pm 0.05}$          | 38.9$_{\pm 0.42}$          |
> | TENT     | 6.0$_{\pm 0.10}$           | 39.8$_{\pm 0.45}$          |
> | +DELTA   | 36.7$_{\pm 0.04}$          | 41.8$_{\pm 0.58}$          |
> | Ent-W    | 1.4$_{\pm 0.06}$           | 39.9$_{\pm 0.79}$          |
> | +DELTA   | 36.5$_{\pm 0.21}$          | 45.1$_{\pm 0.67}$          |

---

> ### Author Response · Authors · 2022-11-13
> **Response to Reviewer E7Gz [2/2]**
>
>
> **Q5: In Table 10, could the authors further provide the results of Div-W+Fisher regularization (namely EATA, this is the full version of ETA in Niu et al, 2022)?**
>
> We add the experiments of EATA on ImageNet-C. As the results in Table 10 are based on Ent-W+TBR, we further provide the performance of EATA+TBR. The results are presented in the table below, demonstrating the superiority of TBR and DOT.
>
> |                    | IID  | non-IID | CI   | CI&non-IID |
> |-----------------------|:----:|:-------:|:----:|:----------:|
> | EATA (Ent-W+Div-W+Fisher) | 48.4 | 22.2    | 47.8 | 20.0       |
> | EATA+TBR (Ent-W+Div-W+Fisher+TBR)                  | 49.2 | 46.0    | 48.4 | 42.7       |
> | Ent-W+TBR+DOT             | 49.9 | 47.5    | 48.4 | 44.5       |
>
>
>
> **Q6: Figure 1 is somewhat confusing. It is hard to distinguish the difference between IID and CI, non-IID and CI & non-IID.**
>
> Thanks for pointing this out. We have redrawn Figure 1 in the revision.
>
> **Q7: In Section 3.1, it would be better to detail describe each scenario and the difference between the previous TTA setting.**
>
> Thanks for the suggestion. We have modified this part in the revision. Besides, following the suggestions from Reviewer MFwz, to avoid confusion, we also update the related terminologies.
>
> **Q8: In Treatment II of Section 3.3, it would be better to extend the “L x” to “Line x of Algorithm 1” to improve the readability of the paper.**
>
> Thanks for the suggestion. We have modified them in the revision.

---

> ### Author Response · Authors · 2022-11-21
> **Response to Reviewer E7Gz [3/-]**
>
> Dear reviewer E7Gz:
>
> Thank you again for your constructive comments and suggestions, which have helped us improve the manuscript significantly. We would be happy to know whether our responses and the updated manuscript have addressed your concerns. We are happy to provide other clarification and responses.
>
> Looking forward to hearing from you!
>
> Best,
>
> Authors

---

### Author Response · Authors · 2022-11-13
**General Comment and Updated Manuscript**

We thank all the reviewers for their valuable comments and suggestions. We have put in a major effort to address the concerns. We believe the paper's quality has been significantly improved with the help of the reviewers.

All major modifications in the pdf file have been highlighted in blue to ease the reading. We summarize our main revision of the manuscript below.

- We modified some terms to avoid confusion as suggested by the reviewers, including the names of different scenarios (IID $\rightarrow$ IS+CB, non-IID $\rightarrow$ DS+CB, CI $\rightarrow$ IS+CI, CI&non-IID $\rightarrow$ DS+CI), and the "bias" term in Section 3.2.
- Correspondingly, we modified the name of our method (also title) as suggested by the reviewers. The new one emphasizes the effect of the proposed approaches in multiple scenarios compared with previous ways as shown in Table 1.
- We provided more experimental results, including the error bars (Table 4-8), the comparison on the newly introduced real-world dataset YTBB-sub (Table 8), the KL-div's results (Table 10), and the results under smaller $\rho$ and $\pi$ (Table 12), etc.
- We provided more detailed discussions for the experimental results.
- We improved some figures, e.g., Figure 1 and Figure 3.
- We provided an algorithm description for TBR.
- We reformulated some sentences for clarity.

---

> ### Comment · Reviewer_E7Gz · 2022-11-28
> **further comments**
>
> Dear All,
>
> The authors have addressed most of my concerns and I would like to keep my score as "wc".
>
> Best,

---

> ### Author Response · Authors · 2022-11-28
> **Thanks for the feedback and discussion**
>
> We are pleased that all reviewers' main concerns are addressed or clarified now. We thank all reviewers for the feedback and discussion over these days!

---

### Decision · Program_Chairs · 2023-01-20

**Decision:**

Accept: poster

**Justification For Why Not Higher Score:**

The paper proposed some interesting ideas and has a reasonable experimental evaluation. The reviewers were in general supportive but none of the reviewers found it to be stand-out paper either as the paper did have its share of weaknesses. Given the scores from all the reviewers and the discussion, I believe Accept (poster) is a reasonable rating for the paper.

**Justification For Why Not Lower Score:**

All reviewer voted in favor of acceptance.

**Metareview: Summary, Strengths And Weaknesses:**

This paper studies some of the recently proposed techniques to solve the test-time adaptation (TTA) problem. In particular, the paper shows that methods that update the batch-norm statistics using the test samples are biased towards more recent samples. Another issue pointed out by the paper is the effect of class-imbalance which can hamper the performance of TTA methods. The paper proposed techniques to mitigate both these issues.

The paper initially received borderline scores and the reviewers raised some concerns. The authors provided a detailed response as well as a major revision of the paper. After the discussions, the reviewers are largely satisfied by the response and the revised version of the paper, and all of the reviewers are now leaning towards acceptance.

TTA is an important problem and the paper is addressing some of the key issues faced by existing TTA methods. All reviewers have appreciated the contributions and, from my own reading too, I have found the paper to be technically strong and the experimental evaluation to be reasonably thorough. Therefore, and following the recommendations from all the reviewers, I recommend the paper for acceptance.

**Note From Pc:**

if the above contains the word "oral" or "spotlight" please see: "oral" presentation means -> notable-top-5% and "spotlight" means -> notable-top-25%. As stated in our emails, we are disassociating presentation type from AC recommendations